# Adaptive Hopfield Network: Rethinking Similarities in Associative Memory

**Shurong Wang**[1,2]**, Yuqi Pan**[2]**, Zhuoyang Shen**[1]**, Meng Zhang**[1]**, Hongwei Wang**[1*] **& Guoqi Li**[2*]
[1]Zhejiang University  [2]Institute of Automation, Chinese Academy of Sciences
shurong.22@intl.zju.edu.cn, guoqi.li@ia.ac.cn

## Abstract

Associative memory models are content-addressable memory systems fundamental to biological intelligence and are notable for their high interpretability. However, existing models evaluate the quality of retrieval based on proximity, which cannot guarantee that the retrieved pattern has the strongest association with the query, failing correctness. We reframe this problem by proposing that a query is a generative variant of a stored memory pattern, and define a variant distribution to model this subtle context-dependent generative process. Consequently, correct retrieval should return the memory pattern with the maximum a posteriori probability of being the query's origin. This perspective reveals that an ideal similarity measure should approximate the likelihood of each stored pattern generating the query in accordance with variant distribution, which is impossible for fixed and pre-defined similarities used by existing associative memories. To this end, we develop adaptive similarity, a novel mechanism that learns to approximate this insightful but unknown likelihood from samples drawn from context, aiming for correct retrieval. We theoretically prove that our proposed adaptive similarity achieves optimal correct retrieval under three canonical and widely applicable types of variants: noisy, masked, and biased. We integrate this mechanism into a novel adaptive Hopfield network (`A-Hop`), and empirical results show that it achieves state-of-the-art performance across diverse tasks, including memory retrieval and various classification tasks. Our code is publicly available here.

## 1 Introduction

Associative memory represents a fundamental paradigm in information storage and retrieval, functioning as a content-addressable memory system that serves as a cornerstone of biological intelligence (Miyashita, 1988; Pearce & Bouton, 2001), particularly in the hippocampus and neocortex (Wang et al., 2014). Unlike conventional computer memory, which retrieves data based on a specific address, associative memory retrieves stored patterns by using a partial or noisy variant of the pattern itself as a cue. This memory paradigm enables robust pattern completion, error correction, and fault-tolerant information processing, making it a compelling model for both understanding biological cognition and developing artificial intelligence systems.

The computational modeling of associative memory has evolved dramatically since its inception. Hopfield (1982) pioneered this field by introducing a recurrent neural network, dubbed Hopfield network, capable of storing and retrieving patterns through energy minimization. Subsequent work (Krotov & Hopfield, 2016; Demircigil et al., 2017) extended memory capacity using a steeper energy function. A pivotal breakthrough came with the establishment of a profound connection between the modern Hopfield network and the attention mechanism (Vaswani et al., 2017), achieved by using the $\text{softmax}(\cdot)$ function to further separate memories (Ramsauer et al., 2021). This insight not only unified two previously disparate fields but also inspired further refinements that strengthened associative memory's performance from different perspectives (Millidge et al., 2022; Hu et al., 2023; Wu et al., 2024a), and broadened applications to tasks like clustering (Saha et al., 2023), time series prediction (Wu et al., 2024b), and more (Krotov et al., 2025).

Despite these significant advances, a critical and unaddressed limitation pervades the literature: the absence of a rigorous framework for assessing retrieval accuracy. Current evaluations typically rely

on proximity-based criteria, such as $\epsilon$-retrieval (Ramsauer et al., 2021; Hu et al., 2023; Wu et al., 2024a; Hu et al., 2024; 2025), which deem retrieval successful if the retrieved pattern is sufficiently close to a certain stored pattern. However, proximity does not establish correctness; ensuring the retrieval is a valid memory provides no guarantee that it is the *correct* one, that is, the one that has the strongest association with the query. This oversight leads to a universal reliance on fixed, pre-defined similarity measures (e.g., inner product or Euclidean distance between two memory patterns). Such one-size-fits-all metrics fail to capture the nuanced, context-dependent *association*, or *similarity*, between the query and the stored memory patterns. For instance, the word *click* is semantically similar to *tap*, phonetically similar to *clique*, and orthographically to *clock* — illustrating that an appropriate notion of similarity is context and task dependent while fixed metrics cannot adapt to such context, nor can they certify correctness.

Our central premise is that correctness is inherently generative: a query $\mathbf{x}$ emerges as a *variant* of an unknown stored pattern $\boldsymbol{\xi}_k$. So, to properly define and achieve correct retrieval, we should model the generative process that transforms a stored pattern $\boldsymbol{\xi}_k$ into a query $\mathbf{x}$. To this end, we encapsulate the context-dependent and application-related subtleness into a probabilistic framework centered on the concept of *variant distribution* $\mathcal{V}(\boldsymbol{\xi}_{1\cdots N})$, a joint distribution over stored patterns $\boldsymbol{\xi}_{1\cdots N}$ and memory variants $\mathbf{x}$, where the likelihood $p_{\mathcal{V}}(\boldsymbol{\xi}_k, \mathbf{x})$ captures how probable that we observe $\boldsymbol{\xi}_k$ and it coincidentally generates $\mathbf{x}$ for $(\boldsymbol{\xi}_k, \mathbf{x}) \sim \mathcal{V}(\boldsymbol{\xi}_{1\cdots N})$. Under this view, a *correct retrieval* returns the memory pattern $\boldsymbol{\xi}_k$ maximizing the posterior $p_{\mathcal{V}}(\boldsymbol{\xi}_k|\mathbf{x})$, that is, the likelihood of $\mathbf{x}$ originates from $\boldsymbol{\xi}_k$ when observed $\mathbf{x}$ as query. With further decomposition, maximizing $p_{\mathcal{V}}(\boldsymbol{\xi}_k|\mathbf{x})$ is equivalent to maximizing $p_{\mathcal{V}}(\mathbf{x}|\boldsymbol{\xi}_k)$, i.e., given $\boldsymbol{\xi}_k$, how probable would it generates $\mathbf{x}$ as its variant. The correct retrieval is therefore finding the pattern $\boldsymbol{\xi}_k$ that is most likely to produce $\mathbf{x}$ by varianting itself. This perspective yields an insight that optimal correct retrieval can be achieved by forcing the similarity measure to mimic the behavior of $p_{\mathcal{V}}(\mathbf{x}|\boldsymbol{\xi}_k)$.

However, it is not possible to derive the variant distribution $\mathcal{V}(\mathbf{x}_{1\cdots N})$ and the likelihood $p_{\mathcal{V}}(\mathbf{x}|\boldsymbol{\xi}_k)$ on most occasions. Thus, we need to reconstruct the unknown by mining deeply from what is observable: the query $\mathbf{x}$, stored patterns $\boldsymbol{\xi}_{1\cdots N}$, and samples matching the context that vaguely describe $\mathcal{V}$. Building on these motivations, we introduce an *adaptive similarity* framework that learns to approximate $p_{\mathcal{V}}(\mathbf{x}|\boldsymbol{\xi}_k)$ from samples observed from the variant distribution, without assuming the variant type is known a priori. Integrating this novel similarity measure into the Hopfield energy yields an adaptive Hopfield network that strives for correct retrieval by capturing the underlying variant distribution. Our key contributions are as follows:

- We introduce the **variant distribution** to model how queries emerge from stored patterns, and formalize **correct retrieval** as a robust and meaningful criterion for evaluating the theoretical accuracy of associative memories.

- We propose **adaptive similarity** derived from this framework and prove its optimality for three canonical and widely applicable types of memory variants: noisy, masked, and biased.

- We build a novel **adaptive Hopfield network** (`A-Hop`) that incorporates learnable adaptive similarity, achieving state-of-the-art performance among computational associative memories on tasks including memory retrieval, tabular classification, image classification, etc.

## 2 BACKGROUND

We consider an associative memory that stores $N$ memory patterns denoted by the memory matrix $\boldsymbol{\Xi} = [\boldsymbol{\xi}_1; \boldsymbol{\xi}_2; \cdots; \boldsymbol{\xi}_N] \in \mathbb{R}^{d \times N}$, where each column vector $\boldsymbol{\xi}_i \in \mathbb{R}^d$ represents a memory pattern. Given a memory variant (query) $\mathbf{x} \in \mathbb{R}^d$, the goal is to retrieve the stored memory that is most associated with it. For simplicity, we denote $[n] \triangleq \{k \in \mathbb{Z} \mid 1 \le k \le n\}$, and $\boldsymbol{\xi} \in \boldsymbol{\Xi}$ means $\boldsymbol{\xi}$ is one of the column vectors of the memory matrix $\boldsymbol{\Xi}$. Appendix A.1 contains a collection of notations.

### 2.1 HOPFIELD NETWORKS

Hopfield network is a line of associative memory that retrieves the most relevant stored memory through a similarity-based matching process. The original Hopfield network (Hopfield, 1982) uses $d$ binary neurons $\boldsymbol{\sigma} \in \{-1, +1\}^d$ to represent the states of the memory system that is limited to storage of binary values. For retrieving, the model sets the query as the initial state (i.e., $\boldsymbol{\sigma}^{(0)} = \mathbf{x}$),

Table 1: Summary of all Hopfield network by components; Hop (Hopfield, 1982), D-Hop (Krotov & Hopfield, 2016), E-Hop (Demircigil et al., 2017), M-Hop (Ramsauer et al., 2021), N-Hop (Millidge et al., 2022), S-Hop (Hu et al., 2023), U-Hop (Wu et al., 2024a), and A-Hop (Ours).

| Model | $\mathrm{sim}(\boldsymbol{\xi}, \mathbf{x})$ | $\mathrm{sep}(\mathbf{s})$ | $\mathrm{mod}(\boldsymbol{\xi})$ | $E(\mathbf{x})$ |
|---|---|---|---|---|
| Hop (Original) | $\boldsymbol{\xi}^\top \mathbf{x}$ | $\mathbf{s}$ | $\boldsymbol{\xi}$ | $-\frac{1}{2}\mathbf{x}^\top \boldsymbol{\Xi}\boldsymbol{\Xi}^\top \mathbf{x}$ |
| D-Hop (Dense) | $\boldsymbol{\xi}^\top \mathbf{x}$ | $\mathbf{s}^k$ | $\boldsymbol{\xi}$ | $-(\mathbf{1}^\top \boldsymbol{\Xi}^\top \mathbf{x})^{k+1}$ |
| E-Hop (Exponential) | $\boldsymbol{\xi}^\top \mathbf{x}$ | $\exp(\mathbf{s})$ | $\boldsymbol{\xi}$ | $-\exp(\mathbf{1}^\top \boldsymbol{\Xi}^\top \mathbf{x})$ |
| M-Hop (Modern) | $\boldsymbol{\xi}^\top \mathbf{x}$ | $\mathrm{softmax}(\mathbf{s})$ | $\boldsymbol{\xi}$ | $\frac{1}{2}\mathbf{x}^\top \mathbf{x} - \mathrm{lse}(\boldsymbol{\Xi}^\top \mathbf{x})$ |
| N-Hop (Universal) | $-\|\boldsymbol{\xi} - \mathbf{x}\|_1$ | $\arg\max(\mathbf{s})$ | $\boldsymbol{\xi}$ | $/$ |
| S-Hop (Sparse) | $\boldsymbol{\xi}^\top \mathbf{x}$ | $\mathrm{sparsemax}(\mathbf{s})$ | $\boldsymbol{\xi}$ | $\frac{1}{2}\mathbf{x}^\top \mathbf{x} - \Psi_2^\star(\beta \boldsymbol{\Xi}^\top \mathbf{x})$ |
| U-Hop (Uniform) | $\boldsymbol{\xi}^\top \mathbf{x}$ | $\alpha-\mathrm{entmax}(\mathbf{s})$ | $\boldsymbol{\Phi}^\top \boldsymbol{\Phi} \boldsymbol{\xi}$ | $\frac{1}{2}\mathbf{x}^\top \boldsymbol{\Phi}^\top \boldsymbol{\Phi} \mathbf{x} - \Psi_\alpha^\star(\beta \boldsymbol{\Xi}^\top \boldsymbol{\Phi}^\top \boldsymbol{\Phi} \mathbf{x})$ |
| A-Hop (Adaptive) | $\mathbf{w}^\top \mathrm{ftpt}(\boldsymbol{\xi}, \mathbf{x})$ | multiple | $\boldsymbol{\xi}$ | $-\mathrm{lse}(\mathbf{s}(\boldsymbol{\Xi}, \mathbf{x}))$ |

and updates one or more neuron(s) iteratively through the following dynamics until convergence:

$$\boldsymbol{\sigma}_i^{(t+1)} = \mathrm{sgn}\left(\sum_{j=1}^d \mathbf{T}_{i,j}\boldsymbol{\sigma}_j^{(t)}\right), \qquad \text{where } \mathbf{T}_{i,j} = \sum_{k=1}^N \boldsymbol{\xi}_{k,i} \cdot \boldsymbol{\xi}_{k,j}.$$

A vectorized retrieval dynamics exists when updating all neurons simultaneously in one iteration:

$$\boldsymbol{\sigma}^{(t+1)} = \mathrm{sgn}\left(\boldsymbol{\Xi}\boldsymbol{\Xi}^\top \boldsymbol{\sigma}^{(t)}\right).$$

Years later, Krotov & Hopfield (2016) improves the memory capacity of Hopfield network from $\mathcal{O}(d)$ to $\mathcal{O}(2^{d/2})$ when storing random samples. They adopted higher-order polynomial or exponential function (Demircigil et al., 2017) to distinguish each stored memory to alleviate the fuzzy memory problem: $\boldsymbol{\sigma}^{(t+1)} = \mathrm{sgn}(\boldsymbol{\Xi}(\boldsymbol{\Xi}^\top \boldsymbol{\sigma}^{(t)})^k)$ or $\boldsymbol{\sigma}^{(t+1)} = \mathrm{sgn}(\boldsymbol{\Xi}\exp(\boldsymbol{\Xi}^\top \boldsymbol{\sigma}^{(t)}))$.

This concept has evolved significantly and was extended to memories with continuous value. Modern Hopfield networks abstract retrieval as a one-iteration update (Ramsauer et al., 2021), and their retrieval dynamics $\mathcal{T}(\mathbf{x})$ can be unified under a three-step procedure (Millidge et al., 2022):

(1) **Similarity** [$\mathbf{s} = \mathrm{sim}(\boldsymbol{\Xi}, \mathbf{x})$]: The query $\mathbf{x}$ is compared against all stored patterns $\boldsymbol{\xi}_{1\cdots N}$ using the similarity function $\mathrm{sim}(\cdot, \cdot)$, obtaining a vector of similarity scores $\mathbf{s} \in \mathbb{R}^N$, where $\mathbf{s}_k = \mathrm{sim}(\boldsymbol{\xi}_k, \mathbf{x})$.

(2) **Separation** [$\mathbf{p} = \mathrm{sep}(\mathbf{s})$]: The similarity scores (logits) $\mathbf{s}$ are transformed by a separation function $\mathrm{sep}(\cdot)$ into a probability distribution $\mathbf{p}$. The separation function sharpens the scores and emphasizes patterns with high similarity.

(3) **Readout** [$\mathbf{y} = \boldsymbol{\Xi}\mathbf{p}$]: The final retrieved pattern $\mathbf{y}$ is computed as a weighted combination of stored memory patterns, using the weights $\mathbf{p}$ provided by the separation function.

Combining each step gives the unified retrieval dynamics:

$$\mathbf{y} = \mathcal{T}(\mathbf{x}) = \boldsymbol{\Xi}\,\mathrm{sep}(\mathrm{sim}(\boldsymbol{\Xi}, \mathbf{x})).$$

Concretely, Ramsauer et al. (2021) proposed using $\mathrm{softmax}(\cdot)$ function as the separation function, which further enlarges the memory capacity and draws a tight connection between associative memory and attention mechanism, with the retrieval dynamics being $\mathcal{T}(\mathbf{x}) = \boldsymbol{\Xi}\,\mathrm{softmax}\left(\boldsymbol{\Xi}^\top \mathbf{x}\right)$. Later, Hu et al. (2023) proposed sparse Hopfield network substituting $\mathrm{softmax}(\cdot)$ with $\mathrm{sparsemax}(\cdot)$, for inducing sparse selection while retaining differentiability. More recently, Wu et al. (2024a) attempted to store memory patterns in a kernel space with greater separation among patterns, giving rise to adding a new modulation step to the existing three-step unified framework. For clarity, we use the modulation function $\mathrm{mod}(\cdot)$ to describes how memory patterns are stored or pre-trained for better retrieval and larger capacity. So, it broadens the unified framework to:

$$\mathbf{y} = \boldsymbol{\Xi}\,\mathrm{sep}(\mathrm{sim}(\mathrm{mod}(\boldsymbol{\Xi}), \mathbf{x})).$$

The kernelized Hopfield network (Wu et al., 2024a) adopted $\mathrm{mod}(\boldsymbol{\Xi}) = \boldsymbol{\Phi}^\top \boldsymbol{\Phi} \boldsymbol{\Xi}$ for a learnable matrix $\boldsymbol{\Phi} \in \mathbb{R}^{D_\Phi \times d}$, that projects memory patterns into a kernel space with the retrieval dynamics being $\mathcal{T}(\mathbf{x}) = \boldsymbol{\Xi}\,\mathrm{sep}((\boldsymbol{\Phi}\boldsymbol{\Xi})^\top (\boldsymbol{\Phi}\mathbf{x})) = \boldsymbol{\Xi}\,\mathrm{sep}((\boldsymbol{\Phi}^\top \boldsymbol{\Phi}\boldsymbol{\Xi})^\top \mathbf{x})$. The kernel $\boldsymbol{\Phi}$ is trained to minimize a separation loss defined on $\boldsymbol{\Xi}$, so that the expected Euclidean distance between any two memory

patterns is maximized. A succeeding work (Hu et al., 2024) uses spherical codes to find the optimal kernel $\Xi$ that maximizes the capacity of the kernelized Hopfield network.

Furthermore, the energy-based view is a defining feature of Hopfield networks: memory retrieval can be viewed as descending on a *energy landscape* (a Lyapunov function) $E(\cdot)$ whose minima coincide with stored patterns (or their modulated version). Formally, the retrieval dynamics $\mathcal{T}(\mathbf{x})$ and the corresponding energy function $E(\mathbf{x})$ are jointly and carefully designed such that each update monotonically decreases the energy (i.e., $E(\mathcal{T}(\mathbf{x})) < E(\mathbf{x})$), and successful retrieval occurs when being sufficiently close to a generalized fixed point near a specific memory pattern $\boldsymbol{\xi}_k \in \Xi$ (i.e., $\|\mathcal{T}(\mathbf{x}) - \boldsymbol{\xi}_k\|_2 < \epsilon$). This principled linkage between dynamics and energy ensures convergence and provides a powerful interpretable model of memory retrieving for Hopfield networks. Connecting to the previous unified framework, the separation function decides the direction of the retrieval dynamics $\mathcal{T}(\cdot)$, and the modulation function reshapes the geometry of the energy landscape $E(\cdot)$, and we organize all existing Hopfield networks' components and energy function in Table 1.

However, across existing formulations, the energy and retrieval dynamics are anchored to a fixed, task-agnostic similarity measure (typically the dot product). Apart from that, the energy and dynamics are solely determined by stored memories $\Xi$ that overlook the nature of the subtle, nuanced, context-specific *association* between queries and memories required for "correct" retrieval. This fundamental gap motivates our work: to refine the energy and retrieval dynamics around a learnable, adaptive similarity measure while preserving the precious interpretability of Hopfield networks.

## 3 METHODS

In this section, we first establish a rigorous probabilistic framework to define *correct retrieval*, eliminating limitations of conventional proximity-based metrics (Section 3.1). To make this concept practical, we develop the similarity footprint (Section 3.2), a multi-scale descriptor measuring association between queries and memory patterns, and use it to learn an adaptive similarity integrated into Adaptive Hopfield Network (A-Hop) that achieves optimal correct retrieval for noisy, masked, and biased types of variants, and has a decreasing, convergent, and bounded energy (Section 3.3).

### 3.1 VARIANT DISTRIBUTION AND CORRECT RETRIEVAL

Conventional analyses of associative memory (Ramsauer et al., 2021; Hu et al., 2023; Wu et al., 2024a; Hu et al., 2024) mostly focus on $\epsilon$-retrieval:

> **Definition 1:** $\epsilon$-retrieval (Hu et al., 2023; Wu et al., 2024a; Hu et al., 2025)
>
> Given a query $\mathbf{x} \in \mathbb{R}^d$ and the retrieval result $\mathbf{y} \in \mathbb{R}^d$ given by the memory system, a memory pattern $\boldsymbol{\xi} \in \Xi$ is said to be $\epsilon$-retrieved if $\|\mathbf{y} - \boldsymbol{\xi}\|_2 \leq \epsilon$.

While $\epsilon$-retrieval ensures the retrieval result $\mathbf{y}$ lies near a certain stored memory pattern $\boldsymbol{\xi}_k \in \Xi$, it provides no guarantee that $\boldsymbol{\xi}_k$ is the most appropriate match for query $\mathbf{x}$. The query $\mathbf{x}$ may have stronger associations with a different pattern $\boldsymbol{\xi}_j$ ($j \neq k$), denoting that $\boldsymbol{\xi}_j$ could be the more appropriate match for $\mathbf{x}$. This identifies that proximity alone is an insufficient proxy for correctness.

To address this limitation, we use a probability distribution to model the generative process of the query $\mathbf{x}$. We posit that a query $\mathbf{x}$ is not an arbitrary vector but a *variant* of a specific stored memory pattern $\boldsymbol{\xi} \in \Xi$ generated by a context-dependent process. We formalize this via the *variant distribution*, which models the relation of memory patterns $\boldsymbol{\xi} \in \Xi$ and queries $\mathbf{x} \in \mathbb{R}^d$ as variants:

> **Definition 2:** Variant distribution
>
> A variant distribution $\mathcal{V}(\Xi)$ is a joint distribution over pair $(\boldsymbol{\xi}, \mathbf{x}) \in \Xi \times \mathbb{R}^d$ where $\boldsymbol{\xi} \in \Xi$ is one of the stored memory patterns and $\mathbf{x} \in \mathbb{R}^d$ is an arbitrary query.
> For $(\boldsymbol{\xi}, \mathbf{x}) \sim \mathcal{V}(\Xi)$, the probability density function $p_{\mathcal{V}(\Xi)}(\boldsymbol{\xi}, \mathbf{x})$ (or $p_{\mathcal{V}}(\boldsymbol{\xi}, \mathbf{x})$ when unambiguous) measures the likelihood of observing $\boldsymbol{\xi}$ and $\mathbf{x}$ at the same time.

Additionally, the posterior $p_{\mathcal{V}}(\boldsymbol{\xi}|\mathbf{x})$ represents the likelihood that query $\mathbf{x}$ originates from memory pattern $\mathbf{x}$, and the likelihood $p_{\mathcal{V}}(\mathbf{x}|\boldsymbol{\xi})$ models how probable that $\boldsymbol{\xi}$ generates $\mathbf{x}$. This leads to a rigorous definition of the context-dependent *correct retrieval*:

**Definition 3:** Correct retrieval

A query $\mathbf{x}$ is said to be correctly retrieved under $\mathcal{V}(\Xi)$, if the retrieval result $\mathbf{y}$ satisfies that:

$$\arg\min_{\boldsymbol{\xi}' \in \Xi} \left\{ \|\mathbf{y} - \boldsymbol{\xi}'\|_2 \right\} = \arg\max_{\boldsymbol{\xi}' \in \Xi} \left\{ p_{\mathcal{V}}(\boldsymbol{\xi}'|\mathbf{x}) \right\}. \tag{1}$$

In Eq. 1, the left-hand side identifies the closest memory pattern to the retrieval result $\mathbf{y}$ (given by the memory system), while the right-hand side is ground truth (the most probable origin of query $\mathbf{x}$ given by variant distribution $\mathcal{V}(\Xi)$). Thus, intuitively, correct retrieval requires that the closest memory pattern coincides with ground truth. With further derivation,

$$\arg\max_{\boldsymbol{\xi}' \in \Xi} \{p_{\mathcal{V}}(\boldsymbol{\xi}'|\mathbf{x})\} = \arg\max_{\boldsymbol{\xi}' \in \Xi} \left\{ p_{\mathcal{V}}(\mathbf{x}|\boldsymbol{\xi}') \cdot \frac{p_{\mathcal{V}}(\boldsymbol{\xi}')}{p_{\mathcal{V}}(\mathbf{x})} \right\} = \arg\max_{\boldsymbol{\xi}' \in \Xi} \{p_{\mathcal{V}}(\mathbf{x}|\boldsymbol{\xi}') \cdot p_{\mathcal{V}}(\boldsymbol{\xi}')\}. \tag{2}$$

This reformulation is necessary as modeling the likelihood $p_{\mathcal{V}}(\mathbf{x}|\boldsymbol{\xi})$ is more tractable than directly estimating the posterior $p_{\mathcal{V}}(\boldsymbol{\xi}|\mathbf{x})$. The likelihood $p_{\mathcal{V}}(\mathbf{x}|\boldsymbol{\xi})$ is conditioned on a single, finite, known memory $\boldsymbol{\xi}$, while the posterior $p_{\mathcal{V}}(\boldsymbol{\xi}|\mathbf{x})$ requires estimating a complex function that maps the entire query space $\mathbb{R}^d$ to a discrete distribution over $\Xi$. Given that the prior $p_{\mathcal{V}}(\boldsymbol{\xi})$ is typically uniform or can be easily estimated from samples, the central challenge of achieving correct retrieval reduces to accurately modeling $p_{\mathcal{V}}(\mathbf{x}|\boldsymbol{\xi})$, in other words, how probable does $\mathbf{x}$ generate $\boldsymbol{\xi}$ under $\mathcal{V}(\Xi)$? With this in hand, it is possible to model three canonical and common variant types rigorously:

**Definition 4:** Noisy variant

A query $\mathbf{x}$ is a noisy variant if it is generated by adding Gaussian noise to a certain memory pattern $\boldsymbol{\xi} \in \Xi$. Formally, $(\mathbf{x} - \boldsymbol{\xi}) \sim \mathcal{N}(\mathbf{0}, \mathrm{diag}(\boldsymbol{\sigma}))$ holds for $(\boldsymbol{\xi}, \mathbf{x}) \sim \mathcal{V}_{\text{noisy}}(\Xi)$, where $\mathrm{diag}(\mathbf{v})$ transform vector $bv$ to a diagonal matrix. The likelihood of noisy variant is:

$$p_{\mathcal{V}_{\text{noisy}}}(\mathbf{x}|\boldsymbol{\xi}) = \frac{1}{(2\pi)^{d/2}|\mathrm{diag}(\boldsymbol{\sigma})|^{1/2}} \exp\left( -\frac{1}{2}(\mathbf{x} - \boldsymbol{\xi})^\top \mathrm{diag}(\boldsymbol{\sigma})^{-1}(\mathbf{x} - \boldsymbol{\xi}) \right).$$

Noisy variants have been widely studied (Krotov & Hopfield, 2016; Hu et al., 2023; Wu et al., 2024a), and it occurs in scenarios such as sensor noise. Specially, under isotropy $\boldsymbol{\sigma} = \sigma\mathbf{1}$, the respective likelihood reduces to: $p_{\mathcal{V}_{\text{noisy}}}(\mathbf{x}|\boldsymbol{\xi}) = (2\pi\sigma)^{-d/2}\exp(-\|\mathbf{x} - \boldsymbol{\xi}\|_2^2 / 2\sigma)$.

**Definition 5:** Masked variant

A masked variant of a memory pattern $\boldsymbol{\xi} \in \Xi$ is obtained by changing values in each dimension with probability $p_{\text{masked}}$ to numbers generated by $\mathcal{G}$. The likelihood of masked variant is:

$$p_{\mathcal{V}_{\text{masked}}}(\mathbf{x}|\boldsymbol{\xi}) = \exp\left( \ln p_{\text{masked}} \cdot \sum_{i=1}^d [1 - \delta(\mathbf{x}_i - \boldsymbol{\xi}_i)] \right) \times \prod_{i=1}^d p_{\mathcal{G}}(\mathbf{x}_i)^{[1 - \delta(\mathbf{x}_i - \boldsymbol{\xi}_i)]}.$$

Here, $\delta(\cdot)$ denotes the Dirac's delta function, and the value of $\delta(\mathbf{x})$ is 1 iff $\mathbf{x} = \mathbf{0}$, and is 0 otherwise. Masked variants arise in real-world scenarios such as information loss during transmission, the same object appearing in different background, and more.

**Definition 6:** Biased variant

Adding a global bias to memory patterns gives the biased variant. Formally, $\mathbf{x} - \boldsymbol{\xi} = \mathbf{d}$ holds for $(\boldsymbol{\xi}, \mathbf{x}) \sim \mathcal{V}_{\text{biased}}(\Xi)$ and a constant vector $\mathbf{d} \in \mathbb{R}^d$. The likelihood of biased variant:

$$p_{\mathcal{V}_{\text{biased}}}(\mathbf{x}|\boldsymbol{\xi}) = \delta\left[ d - \sum_{i=1}^d \delta(\mathbf{x}_i - \boldsymbol{\xi}_i - \mathbf{d}_i) \right].$$

Biased variants occur as a systematic difference, such as changes in light conditions or use of filters.

We visualize the conditional probability density function $p_{\mathcal{V}}(\mathbf{x}|\boldsymbol{\xi})$ in Fig. 1, providing intuition akin to an *electron cloud*, with a memory pattern $\boldsymbol{\xi}$ as the atom nucleus and its variants as orbiting electrons. A direct observation is that $p_{\mathcal{V}}(\mathbf{x}|\boldsymbol{\xi})$ varies significantly across contexts, and may be analytically intractable. For instance, even though visualizing the noisy + masked variant (Fig. 1 (d)) is possible by composing these two operations, deriving its likelihood $p_{\mathcal{V}}(\mathbf{x}|\boldsymbol{\xi})$ is analytically cumbersome. Consequently, although $\mathcal{V}(\Xi)$ is a principled tool to link queries with memory patterns, it poses two challenges for correct retrieval: (1) the underlying variant type is generally unknown a priori; and (2) the resulting variant distribution $\mathcal{V}(\Xi)$ can be too complex to model explicitly.

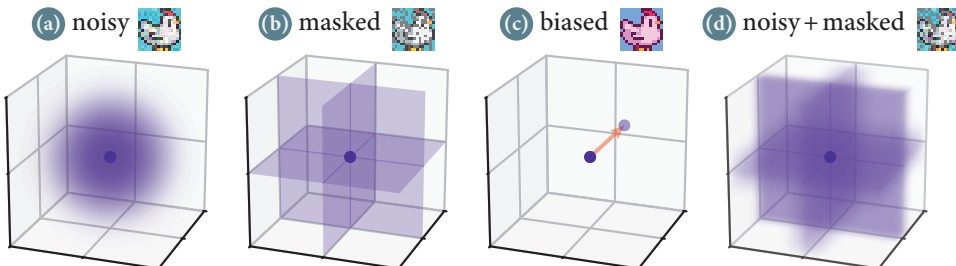

Figure 1: Visualization of probability density function $p_{\mathcal{V}}(\mathbf{x}|\boldsymbol{\xi})$ for noisy, masked, biased, and noisy + masked variants. Darker regions indicate larger $p_{\mathcal{V}}(\mathbf{x}|\boldsymbol{\xi})$, and the central dark point represents $\boldsymbol{\xi}$.

## 3.2 SIMILARITY FOOTPRINT

In the previous section, we established correct retrieval as selecting the pattern that maximizes $p_{\mathcal{V}}(\mathbf{x}|\boldsymbol{\xi})$. While $p_{\mathcal{V}}(\mathbf{x}|\boldsymbol{\xi})$ constitutes the ideal similarity metric (guiding the memory system to return $\arg\max_{\boldsymbol{\xi}\in\Xi}\{p_{\mathcal{V}}(\boldsymbol{\xi}|\mathbf{x})\}$ as per Eq. 2), it is often analytically intractable or unknown in real-world scenarios. To address this, we propose mining richer evidence from observable quantities to construct a tractable surrogate that mimics the selectivity of the true likelihood $p_{\mathcal{V}}(\mathbf{x}|\boldsymbol{\xi})$. We introduce *similarity footprint*, a multi-scale descriptor capturing the relation between query $\mathbf{x}$ and memory patterns $\Xi$ in multiple subspaces. By replacing proximity-based scalar similarity measure (e.g., $\boldsymbol{\xi}^\top \mathbf{x}$) with a structured descriptor (the footprint, a $d$-dimensional vector), the model could form a accurate and robust decision based on the comprehensive subspatial evidence.

We begin with a *base similarity* measure (e.g., dot product $\boldsymbol{\xi}^\top \mathbf{x}$ or negative squared Euclidean distance $-\|\boldsymbol{\xi} - \mathbf{x}\|_2^2$), and define the *k-optimal similarity* between $\boldsymbol{\xi}$ and $\mathbf{x}$:

$$\mathrm{sim}^{(k)}(\boldsymbol{\xi}, \mathbf{x}) \triangleq \max_{D\subseteq[d],|D|=k}\{\mathrm{sim}(\boldsymbol{\xi}_D, \mathbf{x}_D)\}, \qquad \text{where } \mathbf{v}_D \triangleq \left[\mathbf{v}_{D_1}, \mathbf{v}_{D_2}, \cdots, \mathbf{v}_{D_{|D|}}\right]^\top.$$

Here, $\mathbf{v}_D$ is a sub-vector of $\mathbf{v}$ containing only the elements corresponding to indices in $D$. Intuitively, $\mathrm{sim}_k(\boldsymbol{\xi}, \mathbf{x})$ quantifies the largest alignment between $\mathbf{x}$ and $\boldsymbol{\xi}$ within their most consistent $k$-dimensional subspace. This formulation inherently filters out corruption or pure noise by focusing on the most informative dimensions while ignoring outliers. Consequently, we define the *similarity footprint* as the vector aggregating all $k$-optimal similarity $\mathrm{sim}^{(k)}(\cdot, \cdot)$ across all dimensionalities:

$$\mathrm{ftpt}_{\mathrm{sim}}(\boldsymbol{\xi}, \mathbf{x}) \triangleq \left[\mathrm{sim}^{(1)}(\boldsymbol{\xi}, \mathbf{x}), \, \mathrm{sim}^{(2)}(\boldsymbol{\xi}, \mathbf{x}), \, \cdots, \, \mathrm{sim}^{(d)}(\boldsymbol{\xi}, \mathbf{x})\right]^\top$$

This vector serves as the rich multi-scale descriptor of the subspatial relation between $\boldsymbol{\xi}$ and $\mathbf{x}$, offering more evidence for measuring similarity. While a naïve computation of the footprint requires evaluating all $2^d - 1$ subspaces (impractical), an efficient computation is possible for *decomposable similarity* measures (Eq. 6). For such measures, whose result is the aggregation of similaritiy in each dimension, the footprint computation reduces to $\mathcal{O}(d\log d)$ via sorting, and the procedure is visualized in Fig. 2 bottom part. Let $\mathbf{q}$ be the dimension-wise similarity vector (Fig. 2 (1)), where $\mathbf{q}_i = \mathrm{sim}(\boldsymbol{\xi}_i, \mathbf{x}_i)$ for $i \in [d]$, and let $\tilde{\mathbf{q}}$ be the vector $\mathbf{q}$ sorted in decreasing order (Fig. 2 (2)). Then, the similarity footprint can be calculated as (Fig.2 (3)):

$$\mathrm{ftpt}_{\mathrm{sim}}(\boldsymbol{\xi}, \mathbf{x}) = \mathbf{U}\tilde{\mathbf{q}}. \tag{3}$$

where $\mathbf{U}$ is the lower-left triangular matrix of $\mathbf{1}_{d\times d}$, (i.e., $\mathbf{U}_{i,j} = 1$ if $1 \le j \le i \le d$, and $\mathbf{U}_{i,j} = 0$ otherwise). Such simplication is valid because $\mathrm{sim}^{(k)}(\boldsymbol{\xi}, \mathbf{x}) = \sum_{i=1}^{k} \tilde{\mathbf{q}}_i$ holds for decomposable similarity measures, and a rigorous proof is provided in Appendix A.2 Theorem 3.

## 3.3 ADAPTIVE SIMILARITY AND ADAPTIVE HOPFIELD NETWORK

The similarity footprint provides a structured, multi-scale descriptor of the association between a query $\mathbf{x}$ and a memory pattern $\boldsymbol{\xi}$. To exploit these subspatial evidences and construct a similarity measure that adapts to the underlying variant distribution $\mathcal{V}(\Xi)$, we introduce *adaptive similarity* as a learnable linear combination of the footprint elements: $s_{\mathrm{sim}}(\boldsymbol{\xi}, \mathbf{x}) = \mathbf{w}^\top \mathrm{ftpt}_{\mathrm{sim}}(\boldsymbol{\xi}, \mathbf{x}) = \mathbf{w}^\top \mathbf{U}\tilde{\mathbf{q}}$ (Fig. 2 (4)), for some weight vector $\mathbf{w} \in \mathbb{R}^d$. This formulation enables the model to learn the relative

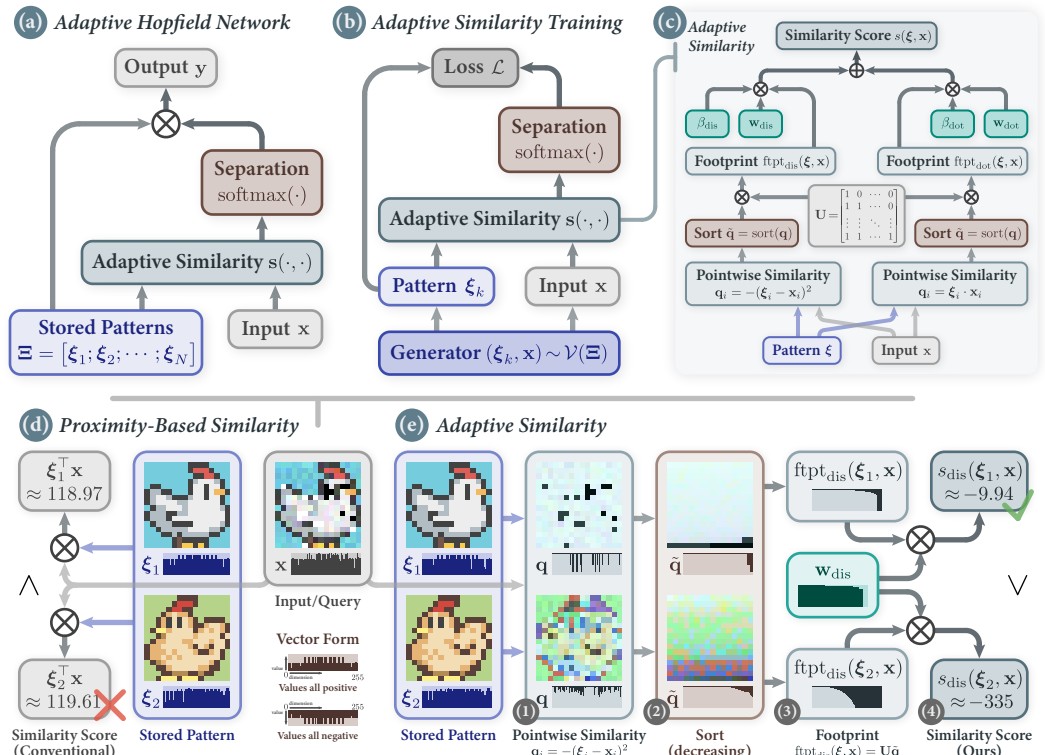

Figure 2: **Top**: The architecture of adaptive Hopfield network (a), the training procedure for adaptive similarity (b) and its design choice (c). **Bottom**: An illustrative example of memory retrieval procedure involving two $16 \times 16$ image patterns and one query. The conventional proximitiy-based similarity (d) fails to retrieve the *correct* pattern while the adaptive similarity (e) succeeds. The computation of proposed adaptive similarity consists of four steps: (1) the pointwise similarity $\mathbf{q} \in \mathbb{R}^d$ is computed as $\mathbf{q}_i = -(\boldsymbol{\xi}_i, \mathbf{x}_i)^2$, (2) elements in $\mathbf{q}$ are sorted in decreasing order to obtain $\tilde{\mathbf{q}}$, (3) the footprint is calculated as $\mathrm{ftpt}_{\mathrm{dis}}(\boldsymbol{\xi}, \mathbf{x}) = \mathbf{U}\tilde{\mathbf{q}}$, and (4) the adaptive similarity score is computed as $s_{\mathrm{dis}}(\boldsymbol{\xi}, \mathbf{x}) = \mathbf{w}_{\mathrm{dis}}^{\top}\mathrm{ftpt}_{\mathrm{dis}}(\boldsymbol{\xi}, \mathbf{x})$.

importance of subspaces with varing sizes and pay extra attention on more informative subspaces. As an example, for masked variants, the model could assign larger weights to the first $m$ terms in the footprint, effectively ignoring the "tail" of the footprint containing corrupted dimensions, whereas for noisy variants, it might distribute weights to larger subspaces for a global view.

To further enhance the model's expressiveness, we combine footprints of multiple base similarities (Fig. 2 (c)) and derive the final similarity function $s(\boldsymbol{\xi}, \mathbf{x})$ and its vectorized form $\mathbf{s}(\boldsymbol{\xi}, \mathbf{x})$:

$$s(\boldsymbol{\xi}, \mathbf{x}) = \sum_{k=1}^{B} \beta_k \cdot \mathbf{w}_k^{\top} \mathrm{ftpt}_{\mathrm{sim}_k}(\boldsymbol{\xi}, \mathbf{x}) \quad \text{and} \quad \mathbf{s}(\boldsymbol{\Xi}, \mathbf{x}) = [s(\boldsymbol{\xi}_1, \mathbf{x}), \; s(\boldsymbol{\xi}_2, \mathbf{x}), \; \cdots, \; s(\boldsymbol{\xi}_N, \mathbf{x})]^{\top},$$

where $\beta_{1\ldots B}$ are learnable scalars weighting $B$ different base similarities. In this work, we adopt two simple and common base similarities: $\mathrm{dis}(\boldsymbol{\xi}, \mathbf{x}) = -\|\boldsymbol{\xi} - \mathbf{x}\|_2^2$ and $\mathrm{dot}(\boldsymbol{\xi}, \mathbf{x}) = \boldsymbol{\xi}^{\top}\mathbf{x}$. Integrating this adaptive similarity into the unified associative memory framework (Table 1) using the $\mathrm{softmax}(\cdot)$ as the separation function yields *Adaptive Hopfield Network* (A-Hop, complete achitecure illustrated in Fig. 2 (a)(c)), and its retrieval dynamics can be formulated as:

$$\mathbf{y} = \mathcal{T}(\mathbf{x}) = \boldsymbol{\Xi}\,\mathrm{sep}(\mathbf{s}(\boldsymbol{\Xi}, \mathbf{x})) = \boldsymbol{\Xi}\,\mathrm{softmax}(\beta_{\mathrm{dis}} \cdot \mathbf{s}_{\mathrm{dis}}(\boldsymbol{\Xi}, \mathbf{x}) + \beta_{\mathrm{dot}} \cdot \mathbf{s}_{\mathrm{dot}}(\boldsymbol{\Xi}, \mathbf{x})). \quad (4)$$

The parameters $\mathbf{w}$'s and $\beta$'s are optimized to align the model's behavior with the underlying variant distribution. We construct a training set by drawing samples from the variant distribution $\mathcal{V}(\boldsymbol{\Xi})$ and minimize the discrepancy loss between the model's predicted likelihood $\tilde{p}_{\mathcal{V}}(\mathbf{x}|\boldsymbol{\xi}_k) \triangleq \mathrm{sep}(\mathbf{s}(\boldsymbol{\Xi}, \mathbf{x}))_k$ and the underlying ground-truth likelihood $p_{\mathcal{V}}(\mathbf{x}|\boldsymbol{\xi}_k)$ (see Fig. 2 (b) for training procedure):

$$\mathcal{L}(\boldsymbol{\Xi}, \mathcal{V}) = \mathbb{E}_{(\boldsymbol{\xi}_k, \mathbf{x}) \sim \mathcal{V}(\boldsymbol{\Xi})} \left[ -\log \tilde{p}_{\mathcal{V}}(\mathbf{x}|\boldsymbol{\xi}_k) \right]. \quad (5)$$

For an intuitive understanding of adaptive similarity, we demonstrates a comprehensive procedure of calculating the adaptive similarity score between $\boldsymbol{\xi}_1, \boldsymbol{\xi}_2$ (pixel art of two chicken) and the query $\mathbf{x}$ (a noisy + masked variant of $\boldsymbol{\xi}_1$) in the botton part of Fig. 2. While rigorously, Adaptive Hopfield Network satisfies the following theoretical properties.

> **Definition 7:** Optimal correct retrieval
>
> We say a retrieval dynamics $\mathcal{T}(\mathbf{x})$ achieves optimal correct retrieval under $\mathcal{V}(\boldsymbol{\Xi})$, if for any $(\boldsymbol{\xi}, \mathbf{x}) \sim \mathcal{V}(\boldsymbol{\Xi})$ it achieves correct retrieval (Def. 3) for query $\mathbf{x}$.

> **Theorem 1:** `A-Hop`'s retrieval dynamics for optimal correct retrieval
>
> The following retrieval dynamics adopted by `A-Hop` achieves optimal correct retrieval for noisy, masked, and biased variants, with a careful design of $\mathbf{s}(\boldsymbol{\Xi}, \mathbf{x})$:
>
> $$\mathbf{y} = \mathcal{T}(\mathbf{x}) = \boldsymbol{\Xi}\operatorname{sep}(\mathbf{s}(\boldsymbol{\Xi}, \mathbf{x}))$$

First, `A-Hop` is theoretically capable of achieving *optimal correct retrieval* (its retrieval is always *correct*, see Def. 7) for noisy, masked, and biased variants when weights $\mathbf{w}$'s and the adaptive similarity $\mathbf{s}(\boldsymbol{\Xi}, \mathbf{x})$ are ideally configured. A complete walk-through and proof to Theorem 1 is presented in Appendix A.2.1. While optimal correct retrieval is not always guaranteed for continuous and parameter-efficient adaptive similarity (e.g., $s(\boldsymbol{\xi}, \mathbf{x}) = \mathbf{w}^\top \operatorname{ftpt}(\boldsymbol{\xi}, \mathbf{x})$), we discuss the tradeoff between correctness and adaptivity in Appendix A.3.1, and show that learnable adaptive similarity can attain high retrieval accuracy with large probability. Together, this theoretical analysis and the empirical results in ablation study (Appendix A.4.6) validate the design choice illustrated in Fig 2.

> **Theorem 2:** `A-Hop`'s decreasing, convergent, and bounded energy function
>
> Energy $E(\mathbf{x})$ will be monotonically decreasing, convergent, bounded from below, for isotropic noisy, and biased variants, if the following energy is used:
>
> $$E(\mathbf{x}) = -\operatorname{lse}\left(\mathbf{s}(\boldsymbol{\Xi}, \mathbf{x})\right)$$

Second, consist with existing associative memories, `A-Hop` guarantees a monotonically decreasing, convergent energy that can be bounded from below by $-\ln n - \|\mathbf{b}\|_2^2/4$ during the iterative retrieval process. Notably, among the models compared in Table 1, this is the only energy function that can be bounded from below even as $\|\boldsymbol{\xi}\|_2 \to \infty$, which guarantees robust retrieval behavior for unbounded patterns. A detailed proof and explanation to Theorem 2 is provided in Appendix A.2.2.

## 4 EXPERIMENTS

We evaluate the effectiveness of `A-Hop` on tasks including memory retrieval, tabular classification, image classification, and multiple instance learning, demonstrating that `A-Hop` achieves state-of-the-art performance on these tasks. A further ablation study validates our design choice of adaptive similarity (Appendix A.4.6). Due to space constraints, full descriptions of baselines, metrics, datasets, and implementation details are moved to Appendix A.4.

### 4.1 MEMORY RETRIEVAL

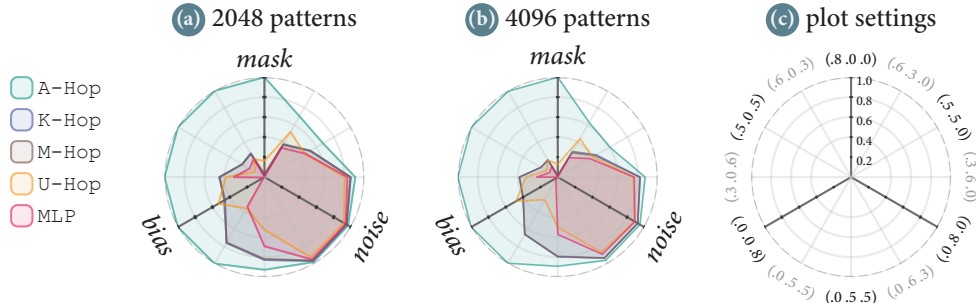

Figure 3: Retrieval accuracy (↑) of 64-dimensional synthetic memory patterns.

Prior work on Hopfield networks primarily assesses retrieval accuracy (Def. 9) under two settings: (1) masking half of the dimensions and (2) adding Gaussian noise. However, real-world data often exhibits compounded corruptions. To rigorously stress-test retrieval correctness, we introduce *mixed variant* parameterized by a triplet $(d_{\text{mask}}, d_{\text{noise}}, d_{\text{bias}}) \in [0,1]^3$ that controls the intensity (difficulty) of masking, noise, and bias, respectively (see Appendix A.4.3 for formal definitions).

We evaluate A-Hop against existing baselines on 12 mixed variant settings (listed in Fig. 3 (c)) with 64-dimensional ($d = 64$) random memory patterns at scales of $N = 2048$ (Fig. 3 (a)) and $N = 4096$ (Fig. 3 (b)). Furthermore, Table 2 details performance under high-intensity corruption using either $N = 2048, d = 64$ synthetic vectors or $N = 2048, d = 784$ MNIST digits as memory patterns.

As shown in Fig. 3, existing associative memory baselines perform poorly when masking or bias is involved as their retrieval accuracy is less than 20% on the *mask* axis and less than 60% on the *bias* axis while A-Hop achieve near-perfect accuracy on these settings. It is because fixed proximity-based similarity cannot distinguish between irrelevance and missing data (masking), and have no chance to learn the biased term added to memory pattern. In Table 2, baselines exhibit a sharp performance collapse as difficulty increases, while A-Hop maintains robust retrieval gaining a 6% to 20% accuracy increment. A-Hop's adaptive similarity creates a noticeable empirical gap by achieving higher retrieval accuracy and lower retrieval error in all settings. This highlights its robustness and impressive adaptability to align similarity to the underlying variant distribution through learning.

Table 2: Retrieval accuracy ($\uparrow$) and error ($\downarrow$) between models. Each cell contains the mean accuracy or error with standard deviation in a smaller font. Results of the best-performing model are **bolded**. Difficulty $d$ stands for mixed variant setting with $(d_{\text{mask}}, d_{\text{noise}}, d_{\text{bias}}) = (d, d, d)$.

| Dataset | Synthetic | | | | MNIST | | | |
|---|---|---|---|---|---|---|---|---|
| Difficulty | 0.4 | | 0.5 | | 0.6 | | 0.7 | |
| Metrics | Accuracy | Error | Accuracy | Error | Accuracy | Error | Accuracy | Error |
| M-Hop | $.520_{\pm.02}$ | $.176_{\pm.01}$ | $.195_{\pm.03}$ | $.300_{\pm.01}$ | $.875_{\pm.01}$ | $.013_{\pm.00}$ | $.661_{\pm.02}$ | $.068_{\pm.00}$ |
| N-Hop | $.260_{\pm.04}$ | $.417_{\pm.02}$ | $.059_{\pm.01}$ | $.554_{\pm.01}$ | $.540_{\pm.03}$ | $.143_{\pm.01}$ | $.176_{\pm.02}$ | $.347_{\pm.01}$ |
| U-Hop | $.487_{\pm.03}$ | $.295_{\pm.02}$ | $.177_{\pm.02}$ | $.764_{\pm.02}$ | $.764_{\pm.02}$ | $.064_{\pm.01}$ | $.526_{\pm.02}$ | $.164_{\pm.01}$ |
| $U_2$-Hop[1] | $.521_{\pm.02}$ | $.176_{\pm.01}$ | $.195_{\pm.02}$ | $.298_{\pm.01}$ | $.878_{\pm.01}$ | $.013_{\pm.00}$ | $.660_{\pm.01}$ | $.068_{\pm.01}$ |
| A-Hop | $\mathbf{.724}_{\pm.02}$ | $\mathbf{.106}_{\pm.01}$ | $\mathbf{.360}_{\pm.02}$ | $\mathbf{.227}_{\pm.01}$ | $\mathbf{.939}_{\pm.01}$ | $\mathbf{.005}_{\pm.00}$ | $\mathbf{.849}_{\pm.01}$ | $\mathbf{.015}_{\pm.01}$ |

## 4.2 TABULAR CLASSIFICATION

Table 3: Predictive performance ($\uparrow$) between models on tabular data. Each cell contains mean accuracy or AUC-ROC score with standard deviation in a smaller font. Results of the best-performing associative memory are **bolded**, and the best other model is underlined.

| Model | Adult | Bank | Vaccine | Purchase | Heart |
|---|---|---|---|---|---|
| M-Hop | $.8080_{\pm.001}$ | $.9085_{\pm.003}$ | $.7975_{\pm.001}$ | $.8822_{\pm.001}$ | $.6325_{\pm.002}$ |
| $U_2$-Hop | $.8172_{\pm.003}$ | $.9092_{\pm.002}$ | $.7971_{\pm.003}$ | $.8825_{\pm.002}$ | $.6473_{\pm.002}$ |
| A-Hop | $\mathbf{.8634}_{\pm.002}$ | $\mathbf{.9139}_{\pm.002}$ | $\mathbf{.8042}_{\pm.002}$ | $\mathbf{.9007}_{\pm.001}$ | $\mathbf{.7315}_{\pm.002}$ |
| Extra Trees | $.8595_{\pm.004}$ | $.9098_{\pm.003}$ | $.7932_{\pm.002}$ | $.8916_{\pm.002}$ | $.7175_{\pm.003}$ |
| Random Forest | $.8592_{\pm.002}$ | $.9132_{\pm.003}$ | $.7918_{\pm.003}$ | $.9002_{\pm.001}$ | $.7254_{\pm.002}$ |
| AdaBoost | $.8597_{\pm.003}$ | $.9094_{\pm.001}$ | $.8011_{\pm.002}$ | $.8865_{\pm.001}$ | $.7294_{\pm.001}$ |
| XGBoost | $\underline{.8640}_{\pm.002}$ | $\underline{.9152}_{\pm.003}$ | $\underline{.8034}_{\pm.002}$ | $\underline{.9032}_{\pm.003}$ | $\underline{.7370}_{\pm.003}$ |

We investigate the utility of A-Hop as a memory-based classifier for tabular data (setup details in Appendix A.4.4). Unlike image data, tabular data is often heterogeneous, containing a mix of continuous and categorical features with varying scales and sparsity.

A-Hop consistently outperforms all competing associative memories (M-Hop and $U_2$-Hop), demonstrating that adaptive similarity is crucial for handling the mixed feature types inherent in

---

[1]$U_2$-Hop is U-Hop whose kernel is optmized by Eq. 5, rather than the original separation loss.

tabular data. Furthermore, `A-Hop` surpasses standard deep learning baselines (Extra Trees, Random Forest, AdaBoost) in all datasets. While the gradient-boosting method XGBoost (Chen & Guestrin, 2016) retains a slight edge on 4 out of 5 datasets, `A-Hop` significantly narrows the gap compared to prior approaches, offering a differentiable alternative to tree ensembles. Significantly, the performance gap between `A-Hop` and other memory models is widest on the Adult (about 5%) and Heart (about 10%) datasets. We hypothesize that these datasets contain more subtle and complicated variant distribution where the "informativeness" of dimensions varies per query, but the results suggest that `A-Hop` can find the most informative subspace more accurately and can retrieve relevant prototypes even when the distance in the original feature space is misleading.

### 4.3 IMAGE CLASSIFICATION AND MULTIPLE INSTANCE LEARNING

Table 4: Classification accuracy (↑) of each model on images, and AUC-ROC score (↑) of each model in multiple instance learning task. Each cell contains accuracy or AUC-ROC score with standard deviation in a smaller font. Results of the best-performing associative memory are **bolded**.

| | *Image Classification* | | | | *Multiple Instance Learning* | | | |
|---|---|---|---|---|---|---|---|---|
| Dataset | CIFAR10 | CIFAR100 | Tiny ImageNet | Dataset | Tiger | Fox | Elephant | UCSB |
| `M-Hop` | $.5123_{\pm.003}$ | $.2464_{\pm.003}$ | $.1095_{\pm.002}$ | `M-Hop` | $.8924_{\pm.005}$ | $.6327_{\pm.013}$ | $.9344_{\pm.009}$ | $.8815_{\pm.022}$ |
| `U-Hop` | $.5489_{\pm.002}$ | $.2877_{\pm.002}$ | $.1164_{\pm.002}$ | `S-Hop` | $.8923_{\pm.006}$ | $.6433_{\pm.015}$ | $.9365_{\pm.002}$ | $.8794_{\pm.024}$ |
| `A-Hop` | $\mathbf{.5637}_{\pm.003}$ | $\mathbf{.2904}_{\pm.002}$ | $\mathbf{.1213}_{\pm.002}$ | `A-Hop` | $\mathbf{.9030}_{\pm.007}$ | $\mathbf{.6753}_{\pm.013}$ | $\mathbf{.9451}_{\pm.004}$ | $\mathbf{.8935}_{\pm.022}$ |

Following established protocols, we evaluate `A-Hop` on image classification (Wu et al., 2024a) and multiple instance learning (Ramsauer et al., 2021; Hu et al., 2023) by integrating it as a component within larger and more complicated deep neural network architectures (i.e., `HopfieldLayer`, `HopfieldPooling` (Ramsauer et al., 2021)).

As shown in Table 4, `A-Hop` consistently achieves the highest accuracy and AUC-ROC scores among all Hopfield variants. This demonstrates that the benefits of adaptive similarity extend to complex, high-dimensional data and can enhance the performance of sophisticated models like the Vision Transformer. While the absolute gains in these complex architectures are naturally smaller than in isolated retrieval tasks (as the associative memory is not the core mechanism), the consistent superiority of `A-Hop` validates its role as a general-purpose, robust associative memory layer. Nevertheless, the consistent improvement confirms that optimizing the similarity measure remains a valuable factor for enhancing performance in complex deep learning systems.

### 4.4 ABLATION STUDY

Due to page limit, ablation study are moved to Appendix A.4.6.

## 5 CONCLUSION

We reframe associative memory retrieval as a problem of *correct retrieval* under a task- and context-dependent variant distribution, motivating a similarity measure that approximates the likelihood that a stored pattern generated the query. Building on this principle, we propose adaptive similarity, prove its optimality for three canonical variant families (noisy, masked, biased), and instantiate it in a new adaptive Hopfield network, `A-Hop`. This perspective clarifies why fixed, pre-defined similarities are inherently limited: they cannot align to the prevailing variant distribution and thus struggle to guarantee correctness, whereas adaptivity enables the model to capture the underlying variant distribution through samples, shifting towards correctness.

Empirically, `A-Hop` establishes state-of-the-art performance among Hopfield networks across memory retrieval, tabular classification, image classification, and multiple instance learning. The gains are most pronounced under mixed variant settings where adaptive similarity maintains impressively high retrieval accuracy and low error. In downstream tasks, `A-Hop` consistently improves over prior Hopfield variants. Ultimately, adaptive similarity is a key principle for advancing associative memories, paving the way for more powerful and resilient memory systems.

## 6 ACKNOWLEDGEMENT

This work was partially supported by the CAS Project for Young Scientists in Basic Research (YSBR-116), National Natural Science Foundation of China (625B1029, 62325603, 62236009, U22A20103), Beijing Science and Technology Plan (Z241100004224011), and Shanghai NeuHelium Neuromorphic Technology Co., Ltd.

## 7 ETHICS STATEMENT

This work adheres to the ICLR Code of Ethics. In this study, no human subjects or animal experimentation was involved. All datasets used were sourced in compliance with relevant usage guidelines, ensuring no violation of privacy. We have taken care to avoid any biases or discriminatory outcomes in our research process. No personally identifiable information was used, and no experiments were conducted that could raise privacy or security concerns. We are committed to maintaining transparency and integrity throughout the research process.

## 8 REPRODUCIBILITY STATEMENT

**Code** We provide code to help understand this work, and is publicly available at: `https://anonymous.4open.science/r/Adaptive-Hopfield-Network-C137/`.

**Datasets** All datasets are either included in the repo, or a description for how to download and preprocess the dataset is provided. All datasets are public and raise no ethical concerns.

**Hyperparameters** All parameters of our proposed framework are in Appendix A.4.

**Environment** Details of our experimental setups are provided in Appendix A.4.

**Random Seed** we do not set a random seed specifically for all random behavior, with the random seed determined PyTorch.

## 9 LARGE LANGUAGE MODELS USAGE

Large Language Models (LLMs) were used to aid polishing of the manuscript. Specifically, we used an LLM to assist in refining the language, improving readability, and ensuring clarity in various sections of the paper. The model helped with tasks such as sentence rephrasing, grammar checking, and enhancing the overall flow of the text.

It is important to note that the LLM was not involved in the ideation, research methodology, or experimental design. All research concepts, ideas, and analyses were developed and conducted by the authors. The contributions of the LLM were solely focused on improving the linguistic quality of the paper, with no involvement in the scientific content or data analysis.

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

# A  APPENDIX

## APPENDIX CONTENTS

## A.1 NOTATIONS

Table 5: Notations and symbolds used in this work.

| Symbol | Description |
|---|---|
| $\boldsymbol{\xi}, \boldsymbol{\xi}_k$ | A specific memory pattern ($d \times 1$) (or memory, stored memory pattern, stored pattern). |
| $\boldsymbol{\Xi}$ | The $d \times N$ memory matrix, with each memory pattern being its column vector. |
| $d$ | The dimensionality of memory patterns. |
| $N$ | The number of stored memory patterns. |
| $\text{sim}(\boldsymbol{\xi}, \mathbf{x})$ | The similarity function that measures how strong the association are between the inputs (or similarity, similarity measure, measure, association). |
| $\text{sep}(\mathbf{s})$ | The separation function, turning the output of $\text{sim}(\cdot, \cdot)$ (logits) to a probability distribution. |
| $\text{mod}(\boldsymbol{\Xi})$ | The modulation function that governs how memory patterns are stored and learned. |
| $E(\mathbf{x})$ | The energy landscape, defined on the same vector space as memory patterns. |
| $\mathcal{T}(\mathbf{x})$ | The retrieval dynamics, defined on the same vector space as memory patterns. |
| $\mathbf{p}$ | The probability distribution vector produced by $\text{sep}(\cdot)$. |
| $\mathbf{s}$ | The similarity score vector produced by $\text{sim}(\cdot, \cdot)$. |
| $\mathbf{x}$ | The query vector. Also, the input to the associative memory |
| $\mathbf{y}$ | The retrieval result vector. Also, the output of the associative memory. |
| $\mathcal{V}(\boldsymbol{\Xi})$ | The variant distribution on memory matrix $\boldsymbol{\Xi}$, governs how queries are generated. Each query $\mathbf{x}$ is sampled from this distribution together with its origin memory pattern $\boldsymbol{\xi}$. |
| $p_{\mathcal{V}}(\boldsymbol{\xi}, \mathbf{x})$ | The joint probability density function that measures the likelihood that $\boldsymbol{\xi}$ and $\mathbf{x}$ are observed together. |
| $p_{\mathcal{V}}(\boldsymbol{\xi}|\mathbf{x})$ | The conditional probability density function (posterior) that measures the likelihood that $\mathbf{x}$ originates from $\mathbf{x}$ when observed $\mathbf{x}$. |
| $p_{\mathcal{V}}(\mathbf{x}|\boldsymbol{\xi})$ | The joint probability density function (likelihood) that measures the likelihood that $\boldsymbol{\xi}$ generates $\mathbf{x}$ when observed $\boldsymbol{\xi}$. |
| $\mathbf{q}$ | The dimension-wise similarity vector whose value of the $i$-th index measures the similarity between the value of $i$-th index in $\mathbf{x}$ and $\boldsymbol{\xi}$. |
| $\tilde{\mathbf{q}}$ | The sorted version of $\mathbf{q}$ (sorted in ascending order). |
| $\mathbf{U}$ | The upper right triangle matrix of ones. |
| $\text{dis}(\boldsymbol{\xi}, \mathbf{x})$ | The (negative and squared) Euclidean distance similarity $-\|\mathbf{x} - \boldsymbol{\xi}\|_2^2$. |
| $\text{dot}(\boldsymbol{\xi}, \mathbf{x})$ | The dot product similarity $\mathbf{x}^\top \boldsymbol{\xi}$. |
| $\text{sim}^{(k)}(\boldsymbol{\xi}, \mathbf{x})$ | The $k$-optimal similarity function that finds a $k$-dimensional subspace that maximizes the similarity $\text{sim}(\cdot, \cdot)$ of the inputs within that subspace. |
| $\text{ftpt}_{\text{sim}}(\boldsymbol{\xi}, \mathbf{x})$ | The similarity footprint function that generates the rich descriptor between $\boldsymbol{\xi}$ and $\mathbf{x}$ with $\text{sim}(\cdot, \cdot)$ being the base similarity (or footprint). |
| $s_{\text{sim}}(\boldsymbol{\xi}, \mathbf{x})$ | The adaptive similarity function adopting $\text{ftpt}_{\text{sim}}(\cdot, \cdot)$ with $\text{sim}(\cdot, \cdot)$ as the base similarity. |
| $\mathbf{s}_{\text{sim}}(\boldsymbol{\Xi}, \mathbf{x})$ | The vectorized form of the adaptive similarity function $s_{\text{sim}}(\cdot, \cdot)$, and returns a vector that measures the adaptive similarity between $\boldsymbol{\xi}_i$ and $\mathbf{x}$ for $i \in [N]$. |
| $s(\boldsymbol{\xi}, \mathbf{x})$ | The final adaptive similarity function that aggregate multiple $s_{\text{sim}_k}(\cdot, \cdot)$ for different base similarity $\text{sim}(\cdot, \cdot)$ / footprint. |
| $\mathbf{s}(\boldsymbol{\Xi}, \mathbf{x})$ | The vectorized form of the final adaptive similarity function $s(\cdot, \cdot)$, and returns a vector that measures the adaptive similarity between $\boldsymbol{\xi}_i$ and $\mathbf{x}$ for $i \in [N]$. |
| $\mathbf{w}$ | The weight vector that turns the footprint into a scalar, which is designed to extract information from the rich descriptor. |
| $\beta$ | Scalar used to aggregate different adaptive similarities $\mathbf{s}_{\text{sim}}(\cdot, \cdot)$. |
| $\mathcal{L}(\boldsymbol{\xi}, \mathcal{V})$ | The loss function used to optimize $\mathbf{w}$'s and $\beta$'s |
| $[n]$ | The set of integers less than or equal to $n$. |
| $\text{sgn}(x)$ | Return the sign ($-1$ or $+1$) of the input. |
| $\delta(\mathbf{x})$ | The Dirac delta that returns 1 when the input is $\mathbf{0}$ and returns 0 otherwise. |
| $\mathbf{v}^\top$ | Transpose of a vector / matrix. |
| $\text{diag}(\mathbf{v})$ | Transform vector $\mathbf{v}$ to a diagonal matrix. |
| $\mathbf{v}_D$ | A sub-vector of $\mathbf{v}$ containing only the elements corresponding to indices in $D$. |
| $\|\mathbf{v}\|_p$ | The $\ell_p$ norm. |
| $\text{lse}(\mathbf{v})$ | The log-sum-exp function. |

A.2 THEOREMS

We define the retrieval accuracy that estimates the retrieval performance of an associative memory under a certain variant distribution.

**Definition 8:** Retrieval accuracy

Retrieval accuracy for an associative memory with retrieval dynamics $\mathcal{T}(\cdot)$ is the probability that correct retrieval is met:

$$\mathbb{E}_{(\boldsymbol{\xi},\mathbf{x})\sim\mathcal{V}(\boldsymbol{\Xi})}\left[\delta\left(\arg\min_{\boldsymbol{\xi}'\in\boldsymbol{\Xi}}\{\|\mathcal{T}(\mathbf{x})-\boldsymbol{\xi}'\|_2\} - \arg\max_{\boldsymbol{\xi}'\in\boldsymbol{\Xi}}\{p_{\mathcal{V}}(\boldsymbol{\xi}'|\mathbf{x})\}\right)\right]$$

$$= \Pr_{(\boldsymbol{\xi},\mathbf{x})\sim\mathcal{V}(\boldsymbol{\Xi})}\left[\arg\min_{\boldsymbol{\xi}'\in\boldsymbol{\Xi}}\{\|\mathcal{T}(\mathbf{x})-\boldsymbol{\xi}'\|_2\} = \arg\max_{\boldsymbol{\xi}'\in\boldsymbol{\Xi}}\{p_{\mathcal{V}}(\boldsymbol{\xi}'|\mathbf{x})\}\right]$$

However, Def. 8 is usually intractable as $p_{\mathcal{V}}(\mathbf{x}|\boldsymbol{\xi})$ is unknown and complicated. Therefore, we define empirical retrieval accuracy based on samples drawn from $\mathcal{V}(\boldsymbol{\Xi})$, which is computable, and used in our experiments (Section 4).

**Definition 9:** Empirical retrieval accuracy

Empirical retrieval accuracy for an associative memory with retrieval dynamics $\mathcal{T}(\cdot)$ can be estimated by performing abundant retrieval tests:

$$\mathbb{E}_{(\boldsymbol{\xi},\mathbf{x})\sim\mathcal{V}(\boldsymbol{\Xi})}\left[\delta\left(\arg\min_{\boldsymbol{\xi}'\in\boldsymbol{\Xi}}\{\|\mathcal{T}(\mathbf{x})-\boldsymbol{\xi}'\|_2\} - \boldsymbol{\xi}\right)\right]$$

$$= \Pr_{(\boldsymbol{\xi},\mathbf{x})\sim\mathcal{V}(\boldsymbol{\Xi})}\left[\arg\min_{\boldsymbol{\xi}'\in\boldsymbol{\Xi}}\{\|\mathcal{T}(\mathbf{x})-\boldsymbol{\xi}'\|_2\} = \boldsymbol{\xi}\right]$$

**Theorem 3:** Equivalence form for decomposable base similarity

For decomposable similarity measure $\mathrm{sim}(\cdot,\cdot)$ satisfying:

$$\mathrm{sim}(\boldsymbol{\xi},\mathbf{x}) = \sum_{i=1}^{d}\mathrm{sim}(\boldsymbol{\xi}_i,\mathbf{x}_i) \tag{6}$$

Let $\mathbf{q}_i = \mathrm{sim}(\boldsymbol{\xi}_i,\mathbf{x}_i)$ $(1 \le i \le d)$, and $\tilde{\mathbf{q}}_j$ $(1 \le j \le d)$ be the $j$-th largest element in $\mathbf{q}$. We have:

$$\mathrm{sim}^{(k)}(\boldsymbol{\xi},\mathbf{x}) = \sum_{j=1}^{k}\tilde{\mathbf{q}}_j$$

*Proof.* Let $D^{(k)} \triangleq \arg\max_{D\subseteq[d],|D|=k}\{\mathrm{sim}(\boldsymbol{\xi}_D,\mathbf{x}_D)\}$, and define $Q^{(k)} \triangleq \{\tilde{\mathbf{q}}_i \mid i \in D^{(k)}\}$, then we know:

$$\mathrm{sim}^{(k)}(\boldsymbol{\xi},\mathbf{x}) = \sum_{i\in D^{(k)}}\mathbf{q}_i = \sum_{q\in Q^{(k)}}q.$$

Then, we try to prove by induction that $Q^{(k)} = Q^{(k-1)}\cup\{\tilde{\mathbf{q}}_k\}$ for $1 \le k \le d$, and we let $Q^{(0)} = \varnothing$. For, $k = 1$, we can see that $Q^{(1)} = \arg\max_{i\in[d]}\{\mathbf{q}_i\} = \tilde{\mathbf{q}}_i$ satisfying the induction hypothesis. Next, for $1 < k \le d$, we assume that the hypothesis holds for all $1 \le k' < k$, we can see that

$$Q^{(k-1)} = Q^{(k-2)}\cup\{\tilde{\mathbf{q}}_{k-1}\} = Q^{(k-3)}\cup\{\tilde{\mathbf{q}}_{k-1},\tilde{\mathbf{q}}_{k-2}\} = \bigcup_{j=1}^{k-1}\{\tilde{\mathbf{q}}_j\}$$

Now, suppose the followings:
(1) We can find $a \in \mathbf{q}_{1\cdots d}$ s.t. $a > \tilde{\mathbf{q}}_{k-1}$, $Q^{(k)} = Q^{(k-1)}\cup\{a\}$. This not possible because $\{\tilde{\mathbf{q}}_1,\cdots,\tilde{\mathbf{q}}_{k-2},a\}$ would be a better choice for $Q^{(k-1)}$, and this contradicts with the assumption.
(2) We can find $a \in \mathbf{q}_{1\cdots d}$ s.t. $\tilde{\mathbf{q}}_{k-1} \ge a > \tilde{\mathbf{q}}_k$, $Q^{(k)} = Q^{(k-1)}\cup\{a\}$. This is not possible because

there would be $k$ elements in $\mathbf{q}_{1\cdots d}$ that is larger than $\tilde{\mathbf{q}}_k$, so that $\tilde{\mathbf{q}}_k$ is the $(k+1)$-largest element, which contradicts with the definition of $\mathbf{q}_k$.
(3) We can find $a \in \mathbf{q}_{1\cdots d}$ s.t. $\tilde{\mathbf{q}}_k > a$, $Q^{(k)} = Q^{(k-1)} \cup \{a\}$. This is not possible because $\{\tilde{\mathbf{q}}_1, \cdots, \tilde{\mathbf{q}}_{k-1}, \tilde{\mathbf{q}}_k\}$ is a better choice for $Q^{(k)}$ compared to $\{\tilde{\mathbf{q}}_1, \cdots, \tilde{\mathbf{q}}_{k-1}, a\}$ because $\tilde{\mathbf{q}}_k > a$.

Therefore, by contradiction, we can see that $D^{(k)} = D^{(k-1)} \cup \{\tilde{\mathbf{q}}^{(k)}\}$, and the hypothesis holds due to induction. Consequently,

$$Q^{(k)} = \bigcup_{j=1}^{k} \tilde{\mathbf{q}}_j \quad \Longrightarrow \quad \mathrm{sim}^{(k)}(\boldsymbol{\xi}, \mathbf{x}) = \sum_{q \in Q^{(k)}} q = \sum_{j=1}^{k} \tilde{\mathbf{q}}_j.$$

$\square$

### A.2.1 OPTIMAL CORRECT RETRIEVAL

We now start to prove Theorem 1.

Let us begin with a simple variant — the isotropic noisy variant.

> **Lemma 1:** Optimal correct retrieval for isotropic noisy variant
>
> A-Hop achieves optimal correct retrieval (Def. 7) for isotrophic noisy variant $\mathcal{V}_{\mathrm{noisy}}(\boldsymbol{\Xi})$ (Def. 4, $(\mathbf{x}-\boldsymbol{\xi}) \sim \mathcal{N}(\mathbf{0}, \sigma\mathbf{I})$ for $(\boldsymbol{\xi}, \mathbf{x}) \sim \mathcal{V}_{\mathrm{noisy}}(\boldsymbol{\Xi})$ some $\sigma \in \mathbb{R}$) for arbitrary memory matrix $\boldsymbol{\Xi} \in \mathbb{R}^{d \times N}$.

*Proof.* We claim that the optimal correct retrieval is achieved when using $\mathrm{sep}(\cdot) = \arg\max(\cdot)$, and $\mathrm{ftpt}_{\mathrm{dis}}(\boldsymbol{\xi}, \mathbf{x})$ only (i.e., $\beta_1 = 1$ and $\beta_2 = 0$ in Eq. 4). That is, the retrieval dynamics should be:

$$\mathcal{T}(\mathbf{x}) = \arg\max_{\boldsymbol{\xi}' \in \boldsymbol{\Xi}} \left\{ \mathbf{w}^\top \mathrm{ftpt}_{\mathrm{dis}}(\boldsymbol{\xi}', \mathbf{x}) \right\}$$
$$= \arg\max_{\boldsymbol{\xi}' \in \boldsymbol{\Xi}} \left\{ -\|\boldsymbol{\xi}' - \mathbf{x}\|_2^2 \right\}$$

This step can be satisfied by setting $\mathbf{w}_1 = 1$, and $\mathbf{w}_i = 0$ for $2 \le i \le d$. Then, optimal retrieval is achieved only when Eq. 1 is met. We first estimate the right-hand side of Eq. 1:

$$\arg\max_{\boldsymbol{\xi}' \in \boldsymbol{\Xi}} \left\{ p_{\mathcal{V}_{\mathrm{noisy}}}(\boldsymbol{\xi}|\mathbf{x}) \right\} = \arg\max_{\boldsymbol{\xi}' \in \boldsymbol{\Xi}} \left\{ \ln p_{\mathcal{V}_{\mathrm{noisy}}}(\mathbf{x}|\boldsymbol{\xi}) \right\}$$
$$= \arg\max_{\boldsymbol{\xi}' \in \boldsymbol{\Xi}} \left\{ -\frac{d}{2}\ln 2\pi\sigma - \frac{1}{2\sigma}\|\mathbf{x} - \boldsymbol{\xi}'\|_2^2 \right\}$$
$$= \arg\max_{\boldsymbol{\xi}' \in \boldsymbol{\Xi}} \left\{ -\|\boldsymbol{\xi}' - \mathbf{x}\|_2^2 \right\}$$

The first step comes from Eq. 2, and assuming that the prior $p(\boldsymbol{\xi})$ is uniform (which is often the case for memory retrieval) or can be easily obtained from samples. And the second step comes from Def. 4. We can see that the derived results coincide with the retrieval dynamics derived before. Therefore, plugging the retrieval dynamics to the left-hand side of Eq. 1 gives:

$$\arg\min_{\boldsymbol{\xi}' \in \boldsymbol{\Xi}} \left\{ \|\mathcal{T}(\mathbf{x}) - \boldsymbol{\xi}'\|_2 \right\} = \arg\min_{\boldsymbol{\xi}' \in \boldsymbol{\Xi}} \left\{ \left\| \arg\max_{\boldsymbol{\xi}''} \left\{ -\|\boldsymbol{\xi}'' - \mathbf{x}\|_2^2 \right\} - \boldsymbol{\xi}' \right\|_2 \right\}$$
$$= \sum_{k=1}^{N} \delta\left( \left\| \arg\max_{\boldsymbol{\xi}''} \left\{ -\|\boldsymbol{\xi}'' - \mathbf{x}\|_2^2 \right\} - \boldsymbol{\xi}_k \right\|_2 \right) \cdot \boldsymbol{\xi}_k$$
$$= \sum_{k=1}^{N} \delta\left( \max_{\boldsymbol{\xi}''} \left\{ -\|\boldsymbol{\xi}'' - \mathbf{x}\|_2^2 \right\} - \left[ -\|\boldsymbol{\xi}_k - \mathbf{x}\|_2^2 \right] \right) \cdot \boldsymbol{\xi}_k$$
$$= \arg\max_{\boldsymbol{\xi}_k \in \boldsymbol{\Xi}} \{ -\|\boldsymbol{\xi}_k - \mathbf{x}\|_2^2 \}$$

The second step holds as there always exists a $\boldsymbol{\xi}' \in \boldsymbol{\Xi}$ that let $\| \arg\max_{\boldsymbol{\xi}''} \left\{ -\|\boldsymbol{\xi}'' - \mathbf{x}\|_2^2 \right\} - \boldsymbol{\xi}'\|_2 = 0$, since the resulting vector of the $\arg\max(\cdot) \in \boldsymbol{\Xi}$, and $\boldsymbol{\xi}'$ iterates every column vector of $\boldsymbol{\Xi}$, thus must have coincided with resulting vector, and the thrid step holds for a similar reason.

Therefore, we show that the left-hand side and right-hand side of Eq. 1 are the same $(\arg\max_{\boldsymbol{\xi}'\in\boldsymbol{\Xi}}\{-\|\boldsymbol{\xi}_k - \mathbf{x}\|_2^2\})$. Thus, the requirement for correct retrieval is met for all $(\boldsymbol{\xi}, \mathbf{x}) \sim \mathcal{V}(\boldsymbol{\Xi})$, yielding optimal correct retrieval. $\square$

If we adopt a footprint that does not sort the dimension-wise similarity vector $\mathbf{q}$ by substituting $\tilde{\mathbf{q}}$ in Eq. 3 to $\mathbf{q}$ and gives $\mathrm{ftpt}_{\mathrm{dis}'}(\boldsymbol{\xi}, \mathbf{x}) = \mathbf{U}\mathbf{q}$, we can prove the optimality for the standard noisy variant defined in Def. 4, which is more general than Lemma 1. However, the footprint $\mathrm{ftpt}_{\mathrm{dis}}(\boldsymbol{\xi}, \mathbf{x}) = \mathbf{U}\tilde{\mathbf{q}}$ achieves high empirical retrieval accuracy, but it is harder to estimate analytically.

> **Lemma 2:** Optimal retrieval for noisy variant
>
> A-Hop achieves optimal correct retrieval (Def. 7) for noisy variant $\mathcal{V}_{\mathrm{noisy}}(\boldsymbol{\Xi})$ (Def. 4 for arbitrary memory matrix $\boldsymbol{\Xi} \in \mathbb{R}^{d\times N}$.

*Proof.* Following the same spirit in the proof of Lemma 1. One can see that the right-hand side (RHS) of Eq. 1 is (similar to Lemma 1):

$$\arg\max_{\boldsymbol{\xi}'\in\boldsymbol{\Xi}} \left\{ p_{\mathcal{V}_{\mathrm{noise}}}(\boldsymbol{\xi}'|\mathbf{x}) \right\} = \arg\max_{\boldsymbol{\xi}'\in\boldsymbol{\Xi}} \left\{ -\frac{1}{2}\ln\left[(2\pi)^d|\mathrm{diag}(\boldsymbol{\sigma})|\right] - \frac{1}{2}(\mathbf{x}-\boldsymbol{\xi}')^\top \mathrm{diag}(\boldsymbol{\sigma})^{-1}(\mathbf{x}-\boldsymbol{\xi}') \right\}$$

$$= \arg\max_{\boldsymbol{\xi}'\in\boldsymbol{\Xi}} \left\{ -\sum_{i=1}^d \frac{(\boldsymbol{\xi}'_i - \mathbf{x})^2}{\boldsymbol{\sigma}_i} \right\}$$

While the left-hand side (LHS) of Eq. 1 is (step 1 follows how RHS is resolved in Lemma 1):

$$\arg\min_{\boldsymbol{\xi}'\in\boldsymbol{\Xi}} \left\{ \|\mathcal{T}(\mathbf{x}) - \boldsymbol{\xi}'\|_2 \right\} = \arg\max_{\boldsymbol{\xi}_k\in\boldsymbol{\Xi}} \left\{ \mathbf{w}^\top \mathbf{U}(\boldsymbol{\xi}_k - \mathbf{x})^2 \right\}$$

$$= \arg\max_{\boldsymbol{\xi}_k\in\boldsymbol{\Xi}} \left\{ \mathbf{u}^\top(\boldsymbol{\xi}_k - \mathbf{x})^2 \right\}$$

$$= \arg\max_{\boldsymbol{\xi}_k\in\boldsymbol{\Xi}} \left\{ \sum_{i=1}^d \mathbf{u}_i(\boldsymbol{\xi}_{k,i} - \mathbf{x}_i)^2 \right\}$$

Here, $\mathbf{v}^2 \in \mathbb{R}^d$ denotes a dimension-wise square operation over $\mathbf{v} \in \mathbb{R}^d$, so that $(\boldsymbol{\xi}_k - \mathbf{x})_i^2 = (\boldsymbol{\xi}_{k,i} - \mathbf{x}_i)^2$. Also, we let $\mathbf{u}^\top = \mathbf{w}^\top\mathbf{U}$ for simplicity. One can see that LHS equals RHS when:

$$\forall i,\, i \in [d],\, \mathbf{u}_i = -\frac{1}{\boldsymbol{\sigma}_i}$$

Since $\mathbf{U}$ is a full-rank matrix, so that $\mathbf{w}^\top = \mathbf{u}^\top\mathbf{U}^{-1}$ holds. When we set $\mathbf{w}$ in the following way:

$$\mathbf{w}_i = \begin{cases} -\boldsymbol{\sigma}_1^{-1} & i = 1 \\ \boldsymbol{\sigma}_{i-1}^{-1} - \boldsymbol{\sigma}_i^{-1} & 2 \le i \le d \end{cases}$$

LHS and RHS of Eq. 1 are the same, satisfying the requirement for correct retrieval. Furthermore, we can tell that Lemma 1 is a special case of this lemma. $\square$

> **Lemma 3:** Optimal retrieval for masked variant
>
> A-Hop achieves optimal correct retrieval (Def. 7) for masked variant $\mathcal{V}_{\mathrm{masked}}(\boldsymbol{\Xi})$ (Def. 5 for arbitrary memory matrix $\boldsymbol{\Xi} \in \mathbb{R}^{d\times N}$, and for a uniform generator $\mathcal{G}$ ($p_{\mathcal{G}}(\cdot)$ is a constant).

*Proof.* As in Lemma 1, we first reformulate the RHS of Eq. 1, and find the suitable choice for $\mathbf{w}$ to make LHS of Eq. 1 equal RHS. We let $\mathbf{q}$ be the dimension-wise similarity vector with $\mathbf{q}_i =$

$-(\boldsymbol{\xi}_i - \mathbf{x}_i)^2$, and $\tilde{\mathbf{q}}$ the vector that sort $\mathbf{q}$ in ascending order. The RHS can be expanded as:

$$\arg\max_{\boldsymbol{\xi}' \in \boldsymbol{\Xi}} \{p_{\mathcal{V}_{\text{masked}}}(\boldsymbol{\xi}'|\mathbf{x})\}$$

$$= \arg\max_{\boldsymbol{\xi}' \in \boldsymbol{\Xi}} \left\{ \ln p_{\text{masked}} \cdot \sum_{i=1}^{d} [1 - \delta(\mathbf{x}_i - \boldsymbol{\xi}'_i)] + \sum_{i=1}^{d} [1 - \delta(\mathbf{x}_i - \boldsymbol{\xi}_i)] \ln p_{\mathcal{G}}(\mathbf{x}'_i) \right\}$$

$$= \arg\max_{\boldsymbol{\xi}' \in \boldsymbol{\Xi}} \left\{ (\ln p_{\text{masked}} + \ln p_{\mathcal{G}}) \cdot \sum_{i=1}^{d} [1 - \delta(\mathbf{x}_i - \boldsymbol{\xi}'_i)] \right\}$$

$$= \arg\max_{\boldsymbol{\xi}' \in \boldsymbol{\Xi}} \left\{ -d + \sum_{i=1}^{d} \delta(\mathbf{x}_i - \boldsymbol{\xi}'_i) \right\}$$

$$= \arg\max_{\boldsymbol{\xi}' \in \boldsymbol{\Xi}} \left\{ \sum_{i=1}^{d} \delta(\tilde{\mathbf{q}}_i) \right\}$$

Here, the second step is valid as $p_{\mathcal{G}}$ is a constant, and we term this constant $p_{\mathcal{G}}$. As $p_{\mathcal{G}}$ is a constant, it must be less than or equal to 1 to make $p_{\mathcal{G}}(\cdot)$ a valid probability density function. Additionally, we know $0 \le p_{\text{masked}} < 1$ from definition, and therefore, $\ln p_{\text{masked}} + \ln p_{\mathcal{G}} < 0$, and this explains why a sign change occurs in step three. The derived RHS suggests designing a discrete adaptive similarity:

$$s_{\text{dis}}(\boldsymbol{\xi}, \mathbf{x}) = \mathbf{w}^\top \boldsymbol{\delta}(\tilde{\mathbf{q}}_i) = \sum_{i=1}^{d} \mathbf{w}_i \delta(\tilde{\mathbf{q}}_i)$$

Then, the LHS would be (first step following that of Lemma 1, and recall we use $\arg\max(\cdot)$ as separation function):

$$\arg\min_{\boldsymbol{\xi}' \in \boldsymbol{\Xi}} \{\|\mathcal{T}(\mathbf{x}) - \boldsymbol{\xi}'\|_2\} = \arg\max_{\boldsymbol{\xi}_k \in \boldsymbol{\Xi}} \{\mathbf{w}^\top \boldsymbol{\delta}(\tilde{\mathbf{q}}_i)\} = \arg\max_{\boldsymbol{\xi}_k \in \boldsymbol{\Xi}} \left\{ \sum_{i=1}^{d} \mathbf{w}_i \cdot \delta(\tilde{\mathbf{q}}_i) \right\}$$

Setting $\mathbf{w} = \mathbf{1}$ concludes that LHS equals RHS for all $(\boldsymbol{\xi}, \mathbf{x}) \sim \mathcal{V}(\boldsymbol{\Xi})$, and thus, the optimal correct retrieval is achieved for this concrete adaptive similarity $s_{\text{dis}}(\boldsymbol{\xi}, \mathbf{x})$. $\qquad\square$

It can be shown that it is impossible to find a continuous $s_{\text{dis}}(\boldsymbol{\xi}, \mathbf{x})$ for masked variant's optimal correct retrieval, unless more constraints on $\boldsymbol{\xi}$ and $\mathbf{x}$ are made. Typically, such a continuous function is possible if $\|\boldsymbol{\xi} - \mathbf{x}\|_2 \ge \varepsilon$ (can be bounded from below) for $\varepsilon > 0$.

> **Lemma 4:** Optimal retrieval for biased variant
>
> A-Hop achieves optimal correct retrieval (Def. 7) for biased variant $\mathcal{V}_{\text{biased}}(\boldsymbol{\Xi})$ (Def. 5 for arbitrary memory matrix $\boldsymbol{\Xi} \in \mathbb{R}^{d \times N}$, and an arbitrary difference vector $\mathbf{d} \in \mathbb{R}^d$.

*Proof.* Following the proof to Lemma 1, the LHS of Eq. 1 is:

$$\arg\max_{\boldsymbol{\xi}' \in \boldsymbol{\Xi}} \{p_{\mathcal{V}_{\text{biased}}}(\boldsymbol{\xi}'|\mathbf{x})\} = \arg\max_{\boldsymbol{\xi}' \in \boldsymbol{\Xi}} \left\{ \delta \left[ d - \sum_{i=1}^{d} \delta(\mathbf{x}_i - \boldsymbol{\xi}'_i - \mathbf{d}_i) \right] \right\}$$

$$= \arg\max_{\boldsymbol{\xi}' \in \boldsymbol{\Xi}} \left\{ -d + \sum_{i=1}^{d} \delta(\mathbf{x}_i - \boldsymbol{\xi}'_i - \mathbf{d}_i) \right\}$$

$$= \arg\max_{\boldsymbol{\xi}' \in \boldsymbol{\Xi}} \left\{ -\|\mathbf{x} - \boldsymbol{\xi}' - \mathbf{d}\|_2^2 \right\}$$

The last step follows that the maximum score is both 0 before and after the transform, and the goal is to assign a high score (0) when $\mathbf{x} - \boldsymbol{\xi} = \mathbf{d}$. Here, we use a similar continuous adaptive similarity defined in the main text:

$$s_{\text{dis}}(\boldsymbol{\xi}, \mathbf{x}) = \mathbf{w}^\top \mathbf{U}\mathbf{q} - \mathbf{q}^\top \mathbf{q}$$

with $\mathbf{q} = \mathbf{x} - \boldsymbol{\xi}$, and set $\mathbf{u}^\top = \mathbf{w}^\top \mathbf{U}$ with $\mathbf{u} = 2\mathbf{d}^\top$. Then, the RHS of Eq. 1 is:

$$\begin{aligned}
\underset{\boldsymbol{\xi}' \in \boldsymbol{\Xi}}{\arg\min} \left\{ \|\mathcal{T}(\mathbf{x}) - \boldsymbol{\xi}'\|_2 \right\} &= \underset{\boldsymbol{\xi}_k \in \boldsymbol{\Xi}}{\arg\max} \left\{ \mathbf{u}^\top \mathbf{q} - \mathbf{q}^\top \mathbf{q} \right\} \\
&= \underset{\boldsymbol{\xi}_k \in \boldsymbol{\Xi}}{\arg\max} \left\{ 2\mathbf{d}^\top \mathbf{q} - \mathbf{q}^\top \mathbf{q} - \mathbf{d}^\top \mathbf{d} \right\} \\
&= \underset{\boldsymbol{\xi}_k \in \boldsymbol{\Xi}}{\arg\max} \left\{ -(\mathbf{q} - \mathbf{d})^\top (\mathbf{q} - \mathbf{d}) \right\} \\
&= \underset{\boldsymbol{\xi}_k \in \boldsymbol{\Xi}}{\arg\max} \left\{ -\|\mathbf{q} - \mathbf{d}\|_2^2 \right\} \\
&= \underset{\boldsymbol{\xi}_k \in \boldsymbol{\Xi}}{\arg\max} \left\{ -\|\mathbf{x} - \boldsymbol{\xi}_k - \mathbf{d}\|_2^2 \right\}
\end{aligned}$$

This follows immediately that LHS equals RHS, and the optimal correct retrieval is achieved when:

$$\mathbf{w}_i = \begin{cases} 2\mathbf{d}_1 & i = 1 \\ 2\mathbf{d}_i - 2\mathbf{d}_{i-1} & 2 \leq i \leq d \end{cases}$$

$\square$

It finally comes down to Theorem 1:

> **Theorem 1:** `A-Hop`'s retrieval dynamics for optimal correct retrieval
>
> ret-dyn The following retrieval dynamics adopted by `A-Hop` achieves optimal correct retrieval for noisy, masked, and biased variants, with a careful design of $\mathbf{s}(\boldsymbol{\Xi}, \mathbf{x})$:
>
> $$\mathbf{y} = \mathcal{T}(\mathbf{x}) = \boldsymbol{\Xi}\operatorname{sep}(\mathbf{s}(\boldsymbol{\Xi}, \mathbf{x}))$$

*Proof.* First of all, $\operatorname{sep}(\cdot) = \arg\max(\cdot)$ is crucial for achieving optimal correct retrieval, as it transforms the left-hand side of Eq. 1 as (see Lemma 1):

$$\underset{\boldsymbol{\xi}' \in \boldsymbol{\Xi}}{\arg\min} \left\{ \|\mathcal{T}(\mathbf{x}) - \boldsymbol{\xi}'\|_2 \right\} = \underset{\boldsymbol{\xi}_k \in \boldsymbol{\Xi}}{\arg\max} \left\{ \mathbf{s}_{\mathrm{dis}}(\boldsymbol{\xi}_k, \mathbf{x}) \right\}$$

In Lemma 2, we see that using the following adaptive similarity achieves optimal correct retrieval:

$$s_{\mathrm{dis}}(\boldsymbol{\xi}, \mathbf{x}) = \mathbf{w}^\top \mathbf{U}\mathbf{q} \qquad \text{with } \mathbf{w}_i = \begin{cases} -\boldsymbol{\sigma}_1^{-1} & i = 1 \\ \boldsymbol{\sigma}_{i-1}^{-1} - \boldsymbol{\sigma}_i^{-1} & 2 \leq i \leq d \end{cases}$$

In Lemma 3, we see that using the following adaptive similarity achieves optimal correct retrieval:

$$s_{\mathrm{dis}}(\boldsymbol{\xi}, \mathbf{x}) = \mathbf{w}^\top \boldsymbol{\delta}(\tilde{\mathbf{q}}) \qquad \text{with } \mathbf{w}_i = 1 \text{ for } i \in [d]$$

In Lemma 4, we see that using the following adaptive similarity achieves optimal correct retrieval:

$$s_{\mathrm{dis}}(\boldsymbol{\xi}, \mathbf{x}) = \mathbf{w}^\top \mathbf{U}(\mathbf{x} - \boldsymbol{\xi}) - (\mathbf{x} - \boldsymbol{\xi})^\top (\mathbf{x} - \boldsymbol{\xi}) \qquad \text{with } \mathbf{w}_i = \begin{cases} 2\mathbf{d}_1 & i = 1 \\ 2\mathbf{d}_i - 2\mathbf{d}_{i-1} & 2 \leq i \leq d \end{cases}$$

$\square$

One can see that achieving optimal correct retrieval is not easy, and it requires the sacrifice of continunity. However, we can build a continuous adaptive similarity inspired from the proof of Theorem 1 that achieve high retrieval accuracy (at least, empirically). For more discussion on this topic, please read Appendix A.3.1.

### A.2.2 UNIFIED ADAPTIVE SIMILARITY AND ENERGY FUNCTION

We can find that the adaptive similarity in Lemma 1 has the form:

$$s(\boldsymbol{\xi}, \mathbf{x}) = -(\mathbf{x} - \boldsymbol{\xi})^\top (\mathbf{x} - \boldsymbol{\xi})$$

while that of Lemma 2 has the form:

$$s(\boldsymbol{\xi}, \mathbf{x}) = -(\mathbf{x} - \boldsymbol{\xi})^\top \operatorname{diag}(\mathbf{a})(\mathbf{x} - \boldsymbol{\xi})$$

for some diagonal matrix $\mathrm{diag}(\mathbf{a})$, and $\mathbf{a}_i > 0$ for all $i \in [d]$. Meanwhile, for Lemma 4 has the form:

$$s(\boldsymbol{\xi}, \mathbf{x}) = -(\mathbf{x} - \boldsymbol{\xi})^\top (\mathbf{x} - \boldsymbol{\xi}) + \mathbf{b}^\top (\mathbf{x} - \boldsymbol{\xi})$$

for some real vector $\mathbf{b}$. That is being said that we can unifies these three adaptive similarity by:

$$s_{\mathrm{unify}}(\boldsymbol{\xi}, \mathbf{x}) = -(\mathbf{x} - \boldsymbol{\xi})^\top \mathrm{diag}(\mathbf{a})(\mathbf{x} - \boldsymbol{\xi}) + \mathbf{b}^\top (\mathbf{x} - \boldsymbol{\xi})$$

However, this similarity is too tough, we can analysis a simpler one:

$$s_{\mathrm{unify}}(\boldsymbol{\xi}, \mathbf{x}) = -(\mathbf{x} - \boldsymbol{\xi})^\top (\mathbf{x} - \boldsymbol{\xi}) + \mathbf{b}^\top (\mathbf{x} - \boldsymbol{\xi})$$

If we use a $\mathrm{softmax}(\cdot)$ function as the separation function and $s_{\mathrm{unify}}(\boldsymbol{\xi}, \mathbf{x})$ as the similarity function, and construct an energy function (with $\mathbf{s}_{\mathrm{unify}}(\boldsymbol{\Xi}, \mathbf{x})$ being the vectorized form of $s_{\mathrm{unify}}(\boldsymbol{\xi}, \mathbf{x})$):

$$E(\mathbf{x}) = -\mathrm{lse}(\mathbf{s}_{\mathrm{unify}}(\boldsymbol{\Xi}, \mathbf{x})) \tag{7}$$

whose gradient is:

$$\nabla_{\mathbf{x}} E(\mathbf{x}) = -\mathrm{softmax}(\mathbf{s}_{\mathrm{unify}}(\boldsymbol{\Xi}, \mathbf{x}))^\top \nabla_{\mathbf{x}} \mathbf{s}_{\mathrm{unify}}$$

Letting $\mathbf{p}_i(\mathbf{x}) \triangleq \mathrm{softmax}(\mathbf{s}_{\mathrm{unify}}(\boldsymbol{\Xi}, \mathbf{x}))_i$:

$$\nabla_{\mathbf{x}} E(\mathbf{x}) = -\sum_{i=1}^{N} \mathbf{p}_i(\mathbf{x}) \nabla_{\mathbf{x}} s_{\mathrm{unify}}(\boldsymbol{\xi}_i, \mathbf{x})$$

$$= -\sum_{i=1}^{N} \mathbf{p}_i(\mathbf{x}) \cdot (-2\mathbf{x} + 2\boldsymbol{\xi}_i + \mathbf{b})$$

$$= 2\mathbf{x} - \mathbf{b} - 2\sum_{i=1}^{N} \mathbf{p}_i(\mathbf{x}) \cdot \boldsymbol{\xi}_i$$

Retrieval on the gradient flow gives:

$$\frac{\mathrm{d}\mathbf{x}}{\mathrm{d}t} = -\nabla_{\mathbf{x}} E(\mathbf{x}) = -2\mathbf{x} + \mathbf{b} + 2\sum_{i=1}^{N} \mathbf{p}_i(\mathbf{x}) \cdot \boldsymbol{\xi}_i$$

Then, consider using gradient descent with step $\eta$, where $\eta > 0$ for discrete-time retrieval:

$$\mathbf{x}^{(t+1)} = \mathbf{x}^{(t)} - \eta \nabla_{\mathbf{x}} E(\mathbf{x}^{(t)})$$

$$= (1 - 2\eta) \cdot \mathbf{x}^{(t)} + \eta \mathbf{b} + 2\eta \sum_{i=1}^{N} \mathbf{p}_i(\mathbf{x}^{(t)}) \cdot \boldsymbol{\xi}_i$$

By setting $\eta = \frac{1}{2}$ that would cancels the $\mathbf{x}$ term on the RHS and remove the coefficient before the summation, which is wonderful:

$$\mathbf{x}^{(t+1)} = \frac{1}{2}\mathbf{b} + \boldsymbol{\Xi}\, \mathrm{softmax}(\mathbf{s}_{\mathrm{unify}}(\boldsymbol{\Xi}, \mathbf{x}))$$

From Lemma 4, we know that the setting $\mathbf{b} = 2\mathbf{d}$ is optimal for noisy variant, plugging this in gives:

$$\mathbf{x}^{(t+1)} - \mathbf{d} = \mathcal{T}(\mathbf{x}^{(t)}) = \boldsymbol{\Xi}\, \mathrm{softmax}(\mathbf{s}_{\mathrm{unify}}(\boldsymbol{\Xi}, \mathbf{x})) \tag{8}$$

suggesting that we add a new de-bias term $-\mathbf{d}$ for biased variants, which coincidentally, remove the bias vector $\mathbf{d}$. However, when there is no bias, Eq. 8 reduce to the simple retrieval dynamics we are familiar with.

We then further analysis the behavior of the energy (Eq. 7) retrieval dynamics (Eq. 8):

> **Lemma 5:** Rewriting the energy
>
> Let the energy function be $E(\mathbf{x}) = -\mathrm{lse}(\mathbf{s}(\boldsymbol{\Xi}, \mathbf{x}))$, where $s(\boldsymbol{\xi}_i, \mathbf{x}) = -(\mathbf{x} - \boldsymbol{\xi}_i)^\top (\mathbf{x} - \boldsymbol{\xi}_i) + \mathbf{b}^\top (\mathbf{x} - \boldsymbol{\xi}_i)$. Then, the energy can be written as
>
> $$E(\mathbf{x}) = \|\mathbf{x}\|_2^2 - \mathrm{lse}(\mathbf{A}\mathbf{x} + \mathbf{c})$$
>
> for some matrix $\mathbf{A} \in \mathbb{R}^{N \times d}$ and vector $\mathbf{c} \in \mathbb{R}^N$.

*Proof.*

$$E(\mathbf{x}) = -\mathrm{lse}\left(\mathbf{s}(\boldsymbol{\Xi}, \mathbf{x})\right)$$

$$= -\ln \sum_{i=1}^{N} \exp\left[-\|\mathbf{x}\|_2^2 - \|\boldsymbol{\xi}_i\|_2^2 + (2\boldsymbol{\xi}_i + \mathbf{b})^\top \mathbf{x} + \boldsymbol{\xi}_i^\top \mathbf{b}\right]$$

$$= -\ln \sum_{i=1}^{n} \exp(-\|\mathbf{x}\|_2^2) + \exp\left[(2\boldsymbol{\xi}_i + \mathbf{b})^\top \mathbf{x} + \boldsymbol{\xi}_i^\top \mathbf{b} - \|\boldsymbol{\xi}_i\|_2^2\right]$$

$$= \ln n + \|\mathbf{x}\|_2^2 - \mathrm{lse}\left(\mathbf{A}\mathbf{x} + \mathbf{c}\right)$$

for $\mathbf{A}_i^\top = 2\boldsymbol{\xi}_i + \mathbf{b}$ and $\mathbf{c}_i = \boldsymbol{\xi}_i^\top \mathbf{b} - \|\boldsymbol{\xi}_i\|_2^2$. Also, we can omit the term $\ln n$ as it is a constant. $\qquad\square$

Therefore, we have decomposed the energy function $E(\mathbf{x})$ into a convex function $g(\mathbf{x}) = \|\mathbf{x}\|_2^2$, and a concave function $-h(\mathbf{x}) = -\mathrm{lse}(\mathbf{A}\mathbf{x} + \mathbf{c})$.

> **Lemma 6:** Decreasing energy
>
> Energy function $E(\mathbf{x})$ would be monotonically decreasing using the retrieval dynamics:
>
> $$\mathbf{x}^{(t+1)} - \mathbf{d} = \mathcal{T}(\mathbf{x}^{(t)}) = \boldsymbol{\Xi}\,\mathrm{softmax}(\mathbf{s}(\boldsymbol{\Xi}, \mathbf{x})) \quad \text{for } s(\boldsymbol{\xi}_i, \mathbf{x}) = -(\mathbf{x} - \boldsymbol{\xi}_i)^\top(\mathbf{x} - \boldsymbol{\xi}_i) + \mathbf{b}^\top(\mathbf{x} - \boldsymbol{\xi}_i)$$

*Proof.* Using the concave convex procedure (Lanckriet & Sriperumbudur, 2009), we construct a convex surrogate function $U_t(\mathbf{x})$ for each iteration $t$ by linearizing the concave function $-h(\mathbf{x})$ around the current $\mathbf{x}^{(t)}$:

$$U_t(\mathbf{x}) = g(\mathbf{x}) - \left[h(\mathbf{x}^{(t)}) + \nabla_{\mathbf{x}} h(\mathbf{x}^{(t)})^\top \left(\mathbf{x} - \mathbf{x}^{(t)}\right)\right]$$

$$= \|\mathbf{x}\|_2^2 - \nabla_{\mathbf{x}} h(\mathbf{x}^{(t)})^\top \mathbf{x} + \left(\nabla_{\mathbf{x}} h(\mathbf{x}^{(t)})^\top \mathbf{x}^{(t)} - h(\mathbf{x}^{(t)})\right)$$

The next $\mathbf{x}^{(t+1)}$ is the minimizer of $U_t(\mathbf{x})$, i.e., $\mathbf{x}^{(t+1)} = \arg\min_{\mathbf{x} \in \mathbb{R}^d} \{U_t(\mathbf{x})\}$, and we can find it by setting its gradient to zero:

$$\nabla_{\mathbf{x}} U_t(\mathbf{x}) = 2\mathbf{x} - \nabla_{\mathbf{x}} h(\mathbf{x}^{(t)}) = \mathbf{0} \implies \mathbf{x}^{(t+1)} = \frac{1}{2}\nabla_{\mathbf{x}} h(\mathbf{x}^{(t)}) = \frac{1}{2}\mathbf{A}^\top \mathrm{softmax}(\mathbf{A}\mathbf{x}^{(t)} + \mathbf{c})$$

We can add the term $-\|\mathbf{x}\|_2^2$ back to $\mathrm{softmax}(\cdot)$ as it is independent of index $i$. Thus, by denoting $\mathbf{p}_i(\mathbf{x}) = \mathrm{softmax}(\mathbf{A}\mathbf{x}^{(t)} + \mathbf{c})_i = \mathrm{softmax}(\mathbf{s}(\boldsymbol{\Xi}, \mathbf{x}))_i$ (follows Lemma 5), we have:

$$\mathbf{x}^{(t+1)} = \frac{1}{2}\sum_{i=1}^{N} \mathbf{A}_i \cdot \mathbf{p}_i(\mathbf{x}^{(t)})$$

$$= \sum_{i=1}^{N} \boldsymbol{\xi}_i \cdot \mathbf{p}_i(\mathbf{x}^{(t)}) - \frac{1}{2}\mathbf{b}\sum_{i=1}^{N} \mathbf{p}_i(\mathbf{x}^{(t)})$$

$$= \boldsymbol{\Xi}\,\mathrm{softmax}(\mathbf{s}(\boldsymbol{\Xi}, \mathbf{x}^{(t)})) - \frac{1}{2}\mathbf{b}$$

and this agrees with what we have derived before (Eq. 8).

Then, by convexity of $h(\mathbf{x})$ (recall $-h(\mathbf{x})$ is concave), we have the following inequality:

$$h(\mathbf{x}) \geq h(\mathbf{x}^{(t)}) + \nabla_{\mathbf{x}} h(\mathbf{x}^{(t)})^\top(\mathbf{x} - \mathbf{x}^{(t)})$$

$$\implies g(\mathbf{x}) - h(\mathbf{x}) \leq g(\mathbf{x}) - h(\mathbf{x}^{(t)}) + \nabla_{\mathbf{x}} h(\mathbf{x}^{(t)})^\top(\mathbf{x} - \mathbf{x}^{(t)}) = U_t(\mathbf{x})$$

with the equality holds iff $\mathbf{x} = \mathbf{x}^{(t)}$, and recall that $\mathbf{x}^{(t+1)}$ is the minimum value of $U_t(\mathbf{x})$, we have:

$$E(\mathbf{x}^{(t+1)}) \leq U_t(\mathbf{x}^{(t+1)}) \leq U_t(\mathbf{x}^{(t)}) = E(\mathbf{x}^{(t)}) \tag{9}$$

with equality holds when $\mathbf{x}^{(t)} = \mathbf{x}^{(t+1)}$. Thus, $E(\mathbf{x}^{(t+1)}) \leq E(\mathbf{x}^{(t)})$ completes the proof. $\qquad\square$

We can see that $E(\mathbf{x}) = \|\mathbf{x}\|_2^2 - \mathcal{O}(\|\mathbf{x}\|_2)$, so that $E(\mathbf{x}) \to +\infty$ as $\|\mathbf{x}\|_2 \to +\infty$, so $E(\mathbf{x})$ is coercive, meaning its level sets are compact. Additionally, $U_t(\mathbf{x})$ is 2-strongly convex as $\nabla_{\mathbf{x}}^2 U_t(\mathbf{x}) = 2\mathbf{I}$, therefore,

$$U_t(\mathbf{x}^{(t)}) - U_t(\mathbf{x}^{(t+1)}) \geq \|\mathbf{x}^{(t)} - \mathbf{x}^{(t+1)}\|_2^2$$

Along with Inequality 9:

$$E(\mathbf{x}^{(t)}) - E(\mathbf{x}^{(t+1)}) \geq U_t(\mathbf{x}^{(t)}) - U_t(\mathbf{x}^{(t+1)}) \geq \|\mathbf{x}^{(t)} - \mathbf{x}^{(t+1)}\|_2^2$$

This yields that

$$E(\mathbf{x}^{(0)}) - E(\mathbf{x}^{(t)}) \geq \sum_{\tau=0}^{t-1} \|\mathbf{x}^{(\tau+1)} - \mathbf{x}^{(\tau)}\|_2^2$$

This mean that $E(\mathbf{x}^{(t)})$ is bounded from above. We can see that the sequence of $\mathbf{x}$: $\{\mathbf{x}^{(t)}\}$ must remain within the level set $\{\mathbf{x} \mid E(\mathbf{x}) \leq E(\mathbf{x}^{(0)})\}$ as the sequence $E(\mathbf{x}^{(0)})$ is non-increasing. Therefore, $\{\mathbf{x}^{(t)}\}$ must remains in the compact set $\{\mathbf{x} \mid E(\mathbf{x}) \leq E(\mathbf{x}^{(0)})\}$, and be bounded.

> **Theorem 2:** `A-Hop`'s decreasing, convergent, and bounded energy function
>
> energy Energy $E(\mathbf{x})$ will be monotonically decreasing, convergent, bounded from below, for isotropic noisy, and biased variants, if the following energy is used:
>
> $$E(\mathbf{x}) = -\mathrm{lse}\left(\mathbf{s}(\mathbf{\Xi}, \mathbf{x})\right)$$

*Proof.* In Lemma 6, we have proven that the energy function will be monotonically decreasing.

Also, from above analysis we can see that $E(\mathbf{x}^{(t)})$ could be bounded by:

$$E(\mathbf{x}^{(t)}) \leq E(\mathbf{x}^{(0)}) - \sum_{\tau=0}^{t-1} \|\mathbf{x}^{(\tau+1)} - \mathbf{x}^{(\tau)}\|_2^2$$

On the other hand, we can try to bound $E(\mathbf{x}^{(t)})$ from below:

$$E(\mathbf{x}^{(t)}) = -\ln \sum_{i=1}^{N} \exp\left(-\|\mathbf{x} - \boldsymbol{\xi}_i\|_2^2 + \mathbf{b}^\top(\mathbf{x} - \boldsymbol{\xi}_i)\right)$$

$$\geq -\ln \sum_{i=1}^{N} \exp\left(-\|\mathbf{x} - \boldsymbol{\xi}_i\|_2^2 + \|\mathbf{b}\|_2 \cdot \|\mathbf{x} - \boldsymbol{\xi}_i\|_2\right)$$

$$\geq -\ln \sum_{i=1}^{N} \exp\left[-\left(\frac{\|\mathbf{b}\|_2}{2}\right)^2 + \|\mathbf{b}\|_2 \cdot \frac{\|\mathbf{b}\|_2}{2}\right]$$

$$= -\ln N - \frac{\|\mathbf{b}\|_2^2}{4}$$

As $\|\mathbf{b}\|_2$ is bounded, we can see that $E(\mathbf{x}^{(t)})$ is bounded from below. Since $E(\mathbf{x})$ is monotonically decreasing and bounded from below, we can simply put that it is convergent.

$\square$

We try to find the property for a very different energy landscape and a different retrieval dynamics, also these retrieval dynamics can guaratee optimal correct retrieval, for iostropic noisy, and biased variant, when the separation function is $\arg\max(\cdot)$ and has weight $\mathbf{b}$ choosen ideally. Theorem 2 tries to connect the noval correct retrieval and the traditional energy analysis.

## A.3 DISCUSSION

### A.3.1 ON GOOD DESIGN CHOICE OF ADAPTIVE SIMILARITY

Achieving optimal correct retrieval is costly, as it requires designing "weird" similarity function for corner cases or use discrete function that makes the model unlearnable. For instance, it is impossible

for a continuous similarity function to achieve optimal correct retrieval for masked variants, as they have a hard time distinguishing $|\boldsymbol{\xi}_i - \mathbf{x}_i| = 0$ and $|\boldsymbol{\xi}_i - \boldsymbol{\xi}_i| = \epsilon$ for some arbitrary small $\epsilon$. Also, using $\mathrm{softmax}(\cdot)$ as the separation function make it hard to prove whether optimal retrieval is achieved or not, as we cannot exclude the effect of other memory patherns from the one receive largest similarity score, while using $\mathrm{softmax}(\cdot)$ is crucial for learnable adpative similarity.

Therefore, achieving optimal correct retrieval for a certain variant distribution is not the only golden criteria for a similarity measure. More generally, we think good (adaptive) simliarty measures should be assessed from the following aspects:

**Adaptivity** Can the similarity measure adapt to a wide range of variant distribution? This criteria measures the *breadth* of the similarity.

**Correctness** How accurate can the similarity measure predicts the variant distribution? How often does the associative memory achieves correct retrieval? This criteria measures the *depth* of the similarity.

**Learnability** Can this similarity measure be learned easily when some samples from the variant distribution is provided? How many learnable parameters it requires? This criteria measures the *scale* of the similarity.

**Efficiency** What is the time complexity of calculating the similarity score? This criteria measures the *speed* of the similarity.

It is ideal but not likely that we can find a perfect adaptive similarity that meets all above mentioned criteria, because there are tradeoffs between these aspects. For example, for the adaptive similarity introduced in the proofs to Lemma 2, 3 and 4, they only achieves *optimality* (optimal correctness) in noisy, masked and biased variants but would have poor performance in other variants (i.e., poor adaptivity). This is because adaptivity is sacrified when every corner case (cases happen in a very low probability but need extra care) is handled carefully. On the other side, optimality would be limited for similarity with strong adaptivity because the outcome for some corner cases could be contradicting for different scenarios. Moreover, learnable adaptivity similarity might not often achieve optimality as the learning alogithm cannot always guarantee convergence of weights to the optimum state (e.g., minimum loss). Also, large-scale similarity measure with plenty of learnable parameters are not likely to have good efficiency, as their computation procedure could be complicated.

The design of the adaptive similarity $s(\boldsymbol{\xi}, \mathbf{x}) = \beta_{\mathrm{dis}}\mathrm{ftpt}_{\mathrm{dis}}(\boldsymbol{\xi}, \mathbf{x}) + \beta_{\mathrm{dot}}\mathrm{ftpt}_{\mathrm{dot}}(\boldsymbol{\xi}, \mathbf{x})$ in Eq. 4 (illustrated in Fig. 2) is a balanced result of considering all four aspects — adaptivity, correctness, learnability, and efficiency. We will explain the design choice of this adaptive similarity from the perspectives of these four criteria.

**The similarity footprint** The footprint is the core mechanism of adaptive similarity (Eq. 4 and Fig. 2). Sorting and cumulative sum is not the essential part of it, while the idea of exploiting subspaces is. The idea is that considering subspatial information will give extra benefits compared to considering the whole $\mathbb{R}^d$ itself. The footprint picks $d$ most *useful* subspaces from all $2^d - 1$ subspaces for much better efficiency (from computationally impossible to possible). However, sorting and cumulative sum happen to be the right algorithm for computing the similarity score efficiently for decomposable similarity measure (Eq. 6), it doesn't mean that the footprint is about sorting dimensions. Regard the footprint as mining the most valuable subspatial information from all $2^d - 1$ subspaces, and it provides useful evidence for guiding the model to make decision.

Recall that $\mathrm{ftpt}_{\mathrm{sim}}(\boldsymbol{\xi}, \mathbf{x})_k = \mathrm{sim}^{(k)}(\boldsymbol{\xi}, \mathbf{x})$, and $\mathrm{sim}^{(k)}(\boldsymbol{\xi}, \mathbf{x})$ is the $k$-optimal similarity between $\boldsymbol{\xi}$ and $\mathbf{x}$, which finds the optimal $k$-dimensional subspace such that $\boldsymbol{\xi}$ and $\mathbf{x}$ is most associated (closest) on this subspace. That is to say, the footprint finds a representative for each dimensionality $k \in [d]$ and regards the one that minimize the distance between $\boldsymbol{\xi}$ and $\mathbf{x}$ as the most useful $k$-dimensional subspace. But the positional information of each subspaces is not used (i.e., which dimensions contributes to $\mathrm{sim}^{(k)}(\boldsymbol{\xi}, \mathbf{x})$ is unkonwn) this would harm the correctness compared to a similarity measure that exploits all $2^d - 1$ subspaces. Although finding the positional information of each subspaces is possible (which would benefits correctness), it would need $\mathcal{O}(d^2)$ time for computation (which is not worthy).

However, compared to fixed proximity-based similarity (e.g., $\boldsymbol{\xi}^\top \mathbf{x}$), the footprint offers a multi-scale descriptor to the relation between $\boldsymbol{\xi}$ and $\mathbf{x}$ (the evidence for deciding whether $\boldsymbol{\xi}$ generates

$\mathbf{x}$). The $k$-optimal similarity $\mathrm{sim}^{(k)}(\cdot, \cdot)$ ignores $d - k$ non-informative dimensions, which make it inherently suitable for handling masked dimensions and heavily noisy dimensions (by simply ignore them). Therefore, the footprint enables better correctness against fixed proximity-based similarity as it offer richer evidence ($d$ v.s. 1) for models to make the right decision.

**Linear combination of footprint elements** The proposed adaptive similarity calculate the similarity score as the dot product of $\mathrm{ftpt}_{\mathrm{dis}}(\boldsymbol{\xi}, \mathbf{x})$ (a $d$-dimensional vector) and learnable weight $\mathbf{w}$ (also a $d$-dimensional vector). This allows the model to behave differently (by choosing different weight) when facing different variant distribution. Also, one can see that when $\mathbf{w}_d = 1$ and $\mathbf{w}_k = 0$ for $1 \leq k < d$, the adaptive similarity degenerates to its base similarity as $\mathbf{w}^\top \mathrm{ftpt}_{\mathrm{sim}}(\boldsymbol{\xi}, \mathbf{x}) = \mathrm{sim}(\boldsymbol{\xi}, \mathbf{x})$. That is to say, the adaptive similarity has better expressiveness (better adaptivity) and is at least as good as the fixed proximity-based similarity.

However, one can use a more complicated method (e.g., train a neural network to let $s(\boldsymbol{\xi}, \mathbf{x}) = \mathrm{MLP}(\boldsymbol{\xi}, \mathbf{x}, \mathrm{ftpt}_{\mathrm{sim}}(\boldsymbol{\xi}, \mathbf{x}))$) to extract information from footprints and gains better correctness and adaptivity. But this also brings more computation cost and may make the model harder to train. Using a linear map is the simplest and fastest way to exploit the footprint, and empirical results demonstrates that it can largely improve the performance of existing associative memory models in the task of memory retrieval.

**Aggregating among different similarities** The proposed adaptive similarity aggregate the similarity footprint with base similarity $\mathrm{dis}(\boldsymbol{\xi}, \mathbf{x}) = -\|\boldsymbol{\xi} - \mathbf{x}\|_2^2$ and $\mathrm{dot}(\boldsymbol{\xi}, \mathbf{x}) = \boldsymbol{\xi}^\top \mathbf{x}$. Aggregating adaptive similarities with different base similarity would improve model's adaptivity and correctness but reduce its efficiency (scale up the runtime by a constant). As an example, when we find a similarity $\mathbf{s}_1(\cdot, \cdot)$ that achieves optimal correct retrieval under variant distribution $\mathcal{V}_1$, and another similarity $\mathbf{s}_2(\cdot, \cdot)$ that achieves optimal correct retrieval under another variant distribution $\mathcal{V}_2$. Ideally, the aggregated model can set $\beta_1 = 1$ and $\beta_2 = 0$ under $\mathcal{V}_1$ and $\beta_1 = 0$ and $\beta_2 = 1$ under $\mathcal{V}_2$ to achieve optimality under both variant distribution.

Think of base similarity as base vectors in a vector space, and the aggregated model resembles the span of these basis. The learnable parameter $\beta_{1 \cdots B}$ controls how each base similarity is used in different scenarios (the weight of each basis). It is clear that $\mathrm{span}(\mathbf{v}_1, \mathbf{v}_2, \cdots, \mathbf{v}_B)$ covers more points in the vector space than $\mathrm{span}(\mathbf{v}_1)$ for $B > 1$, and covering more points means achieving better correctness in more occasions.

We can further improve the proposed adaptive similarity by letting it aware of dimension-specific information by setting $\mathbf{q}_i = \mathbf{a}_i \cdot \mathrm{sim}(\boldsymbol{\xi}_i, \mathbf{x}_i) + \mathbf{b}_i$ (for $1 \leq i \leq d$) with learnable weight $\mathbf{a}, \mathbf{b} \in \mathbb{R}^d$. This modification introduces $\mathcal{O}(d)$ extra parameter to the model, but its complexity does not change as we already have $\mathcal{O}(dB)$ learnable parameter (for $B = 2 = \mathcal{O}(1)$). From a different perspective, $\mathbf{q}$ is now the dimension-wise similarity vector after a affine transformation.

Next, we show that the proposed adaptive similarity $s(\boldsymbol{\xi}, \mathbf{x}) = \mathbf{w}^\top \mathbf{U} \tilde{\mathbf{q}}$ is suitable (has high correctness) for noisy + masked variant. Here, we assume that, with large probability, the affection brought by masking is much larger than that of noise. That is to say, for a certain dimension $1 \leq i \leq d$ and $(\boldsymbol{\xi}_k, \mathbf{x}) \sim \mathcal{V}(\boldsymbol{\Xi})$, let $\Delta_i^{\mathrm{masked}} = |\mathbf{x}_i - \boldsymbol{\xi}_{k,i}|$ when the $i$-th dimension is masked and $\Delta_i^{\mathrm{noisy}} = |\mathbf{x}_i - \boldsymbol{\xi}_{k,i}|$ when the $i$-th dimension is not masked, we will have $\Delta_i^{\mathrm{masked}} \gg \Delta_i^{\mathrm{noisy}}$ with large probability.

The intuition is that sorting (i.e., $\tilde{\mathbf{q}} = \mathrm{sort}(\mathbf{q})$, Fig. 2 (2)) can separates masked and un-masked dimensions. Since $\Delta_i^{\mathrm{masked}} \gg \Delta_i^{\mathrm{noisy}}$ with large probability, $\mathbf{q}_i$ (the dimension-wise similarity) would preserve this ordering for masked and un-masked dimensions. Suppose that $\mathbf{x}_i$ is masked (i.e. $\mathbf{x}_i = 0$) while $\mathbf{x}_j$ is not masked (i.e., $\mathbf{x}_j = \boldsymbol{\xi}_{k,j} + \boldsymbol{\epsilon}_j$ with $\boldsymbol{\epsilon}_j \sim \mathcal{N}(0, \sigma_j)$), then $\mathbf{q}_i \propto -(\Delta_i^{\mathrm{masked}})^2$ and $\mathbf{q}_j \propto -(\Delta_j^{\mathrm{noisy}})^2$, which results in $\mathbf{q}_i \ll \mathbf{q}_j$ with large probability. When there are exactly $m$ dimensions are masked, and by sorting $\mathbf{q}_{1 \cdots N}$ in decreasing order (to obtain $\tilde{\mathbf{q}}$), the values of $\tilde{\mathbf{q}}_{1 \cdots d-m}$ would be significantly larger than that of $\tilde{\mathbf{q}}_{d-m+1 \cdots d}$. It is because $\tilde{\mathbf{q}}_{1 \cdots d-m}$ are dimensions that are not masked, while $\tilde{\mathbf{q}}_{d-m+1 \cdots m}$ are dimensions that are masked. By setting $\mathbf{w}_{d-m} = 1$ and $\mathbf{w}_i = 0$ for $i \neq d - m$, the associative memory excludes all the masked dimensions and treats the remaining subspace as noisy variants. Let $s(\boldsymbol{\xi}, \mathbf{x} \,|\, m = k)$ denotes the adaptive similarity when assuming there are exactly $k$ masked dimensions. Then, we define the resulting adaptive similarity

as the expected similarity score on $m$:

$$
\begin{aligned}
s(\boldsymbol{\xi}, \mathbf{x}) &= \sum_{k=1}^{d} \Pr[\text{exactly } k \text{ dimensions are masked}] \cdot s(\boldsymbol{\xi}, \mathbf{x} \mid m = k) \\
&= \sum_{k=1}^{d} \binom{d}{k} \cdot p_{\text{masked}}^{k} \cdot (1 - p_{\text{masked}})^{d-k} \cdot \left[ \sum_{i=1}^{d} \delta(d - k - i)(\mathbf{U}\tilde{\mathbf{q}})_i \right] \\
&= \sum_{i=1}^{d} (\mathbf{U}\tilde{\mathbf{q}})_i \sum_{k=1}^{d} \delta(d - k - i) \binom{d}{k} \cdot p_{\text{masked}}^{k} \cdot (1 - p_{\text{masked}})^{d-k} \\
&= \sum_{i=1}^{d} \left[ \binom{d}{i} \cdot p_{\text{masked}}^{d-i} \cdot (1 - p_{\text{masked}})^{i} \right] (\mathbf{U}\tilde{\mathbf{q}})_i \\
&= \mathbf{w}^{\top} \mathbf{U}\tilde{\mathbf{q}}
\end{aligned}
\tag{10}
$$

This suggests that the proper choice of $\mathbf{w}$ is to let $\mathbf{w}_i = \binom{d}{i} p_{\text{masked}}^{d-i} (1 - p_{\text{masked}})^i$. Additionally, we can let $\mathbf{q}_i = -\frac{1}{\boldsymbol{\sigma}_i}(\boldsymbol{\xi}_i - \mathbf{x}_i)^2$ ($1 \leq i \leq d$) to turn noisy variants into isotropic noisy variants as suggested in the proof to Lemma 2, which is to say, the optimal choice for $\mathbf{a}$ and $\mathbf{b}$ is $\mathbf{a}_i = -\boldsymbol{\sigma}_i^{-1}$ and $\mathbf{b}_i = 0$ for $1 \leq i \leq d$. Furthermore, one can examine that the adaptive similarities that achieve optimal correct retrieval in noisy variants (Lemma 2) masked variants (Lemma 3) will not achieve optimality, or even cannot sustain a high retrieval accuracy. This is because the similarity used in Lemma 2 will confuse masked and un-masked dimensions, and similarity used in Lemma 3 will always returns 0 as adding noise make $\boldsymbol{\xi}_i - \mathbf{x}_i = 0$ not possible.

Then, we will show that the $\mathbf{w}^{\top} \mathbf{U}\tilde{\mathbf{q}}$ adaptive similarity achieve optimal correct retrieval for noisy and biased variants, and achieves correct retrieval with high probability for masked variants.

For noisy variants, using the $\mathbf{w}$ derived in Eq. 10 with $p_{\text{masked}} = 0$ guarantees optimal correct retrieval. With $p_{\text{masked}} = 0$, $\mathbf{w}$ turns into $\mathbf{w}_i = \delta(d - i)$ ($1 \leq i \leq d$) (i.e., $\mathbf{w}_d = 1$ and $\mathbf{w}_i = 0$ for $1 \leq i \leq d - 1$). This degenerates the model to the one defined in the proof to Lemma 2 (with $\mathbf{a}_i = -\boldsymbol{\sigma}_i^{-1}$) and enable the model to have global view of the entire $\mathbb{R}^d$ space, and electron cloud of said adaptive similarity would look like a shpere illustrated in Fig. 1 (a).

Next, for biased variants, similar to how we deal with the noisy variants, we let $p_{\text{masked}} = 0$ in the weight derived in Eq. 10, which is $\mathbf{w}_i = \delta(d - i)$ ($1 \leq i \leq d$). Then, we let $\mathbf{a}_i = 1$ and $\mathbf{b}_i = \mathbf{d}_i^2$ ($1 \leq i \leq d$) so that for $(\boldsymbol{\xi}, \mathbf{x}) \sim \mathcal{V}_{\text{biased}}(\boldsymbol{\Xi})$, we will always have $\mathbf{q}_i = -(\boldsymbol{\xi}_i - \mathbf{x}_i)^2 + \mathbf{b}_i = -\mathbf{d}_i^2 + \mathbf{b}_i = 0$. As $\mathbf{q} = \mathbf{0}$, then $\tilde{\mathbf{q}} = \mathbf{0}$ as well, and this would results in $s(\boldsymbol{\xi}, \mathbf{x}) = \mathbf{w}^{\top} \mathbf{U}\tilde{\mathbf{q}} = 0$ achieving the largest value of such adaptive similarity. In other words, the adaptive similarity $s(\boldsymbol{\xi}, \mathbf{x})$ is maximized when $(\boldsymbol{\xi}, \mathbf{x}) \sim \mathcal{V}_{\text{biased}}(\boldsymbol{\Xi})$ (i.e., $\mathbf{x} = \boldsymbol{\xi} + \mathbf{d}$).

Finally, for biased variant, achieving optimal correct retrieval is not possible for continuous adaptive similarity function, unless the difference between query and any stored pattern can be bounded. Formally, when there exists $0 < \epsilon < 1$, s.t. for all $(\boldsymbol{\xi}, \mathbf{x}) \sim \mathcal{V}(\boldsymbol{\Xi})$ and $i \in [d]$, we always have $\epsilon < (\boldsymbol{\xi}_i - \mathbf{x}_i)^2 = -\mathbf{q}_i < \frac{1}{\epsilon}$. Then, we set $(\mathbf{w}^{\top}\mathbf{U})_i = \epsilon^{-3(d-i)}$ and this leads to:

$$
s(\boldsymbol{\xi}, \mathbf{x}) = \mathbf{w}^{\top} \mathbf{U}\tilde{\mathbf{q}} = \sum_{i=1}^{d} \epsilon^{-3(d-i)} \cdot \tilde{\mathbf{q}}_i
$$

Let $s'(\boldsymbol{\xi}, \mathbf{x}) = \sum_{i=1}^{d} \delta(\tilde{\mathbf{q}}_i)$ (the adaptive similarity used in Lemma 3), we have:

$$
-\epsilon^{-3(d - s'(\boldsymbol{\xi},\mathbf{x})) - 2} < \sum_{i=1}^{d} \epsilon^{-3(d-i) - 1} < s(\boldsymbol{\xi}, \mathbf{x}) < -\epsilon^{-3(d - s'(\boldsymbol{\xi},\mathbf{x})) + 1}
$$

$$
\implies \quad \epsilon^{3d} \cdot s(\boldsymbol{\xi}, \mathbf{x}) \in \left( -\epsilon^{3s'(\boldsymbol{\xi},\mathbf{x}) - 2}, -\epsilon^{3s'(\boldsymbol{\xi},\mathbf{x}) + 1} \right)
\tag{11}
$$

As a result of Eq. 11, $s'(\boldsymbol{\xi}_1, \mathbf{x}) < s'(\boldsymbol{\xi}_2, \mathbf{x})$ iff $s(\boldsymbol{\xi}_1, \mathbf{x}) < s(\boldsymbol{\xi}_2, \mathbf{x})$ for $\boldsymbol{\xi}_1, \boldsymbol{\xi}_2 \sim \boldsymbol{\Xi}$ (i.e., $s(\boldsymbol{\xi}, \mathbf{x})$ preserves the ordering of $s'(\boldsymbol{\xi}, \mathbf{x})$). And since $s'(\boldsymbol{\xi}, \mathbf{x})$ achieves optimal correct retrieval (Lemma 3), $s(\boldsymbol{\xi}, \mathbf{x})$ also achieve optimal correct retrieval in this case.

However, when $\mathbf{q}_i$ cannot be bounded by such $\epsilon > 0$, we try to find the probability that $s(\boldsymbol{\xi}, \mathbf{x})$ does not achieve correct retrieval. Let $\mathcal{V}_{m\text{-masked}}(\boldsymbol{\Xi})$ denotes the variant distribution when exactly $m$ dimensions will be masked (i.e., $s'(\boldsymbol{\xi}, \mathbf{x}) = d - m$ for $(\boldsymbol{\xi}, \mathbf{x}) \sim \mathcal{V}_{m\text{-masked}}(\boldsymbol{\Xi})$). Let $p_{\text{err}}$ denotes the probability of $s(\boldsymbol{\xi}, \mathbf{x}) = \mathbf{w}^\top \mathbf{U}\tilde{\mathbf{q}}$ not achieving correct retrieval under masked variants. Formally, when $\boldsymbol{\Xi}$ and $\boldsymbol{\xi}$ are generated randomly,

$$
\begin{aligned}
p_{\text{err}} &\triangleq \Pr_{\substack{(\boldsymbol{\xi}, \mathbf{x}) \sim \mathcal{V}_{\text{masked}}(\boldsymbol{\Xi}) \\ \boldsymbol{\xi}' \in \boldsymbol{\Xi}, \boldsymbol{\xi}' \neq \boldsymbol{\xi}}} \left[ s(\boldsymbol{\xi}, \mathbf{x}) < s(\boldsymbol{\xi}', \mathbf{x}) \right] \\
&= \Pr_{\substack{(\boldsymbol{\xi}, \mathbf{x}) \sim \mathcal{V}_{\text{masked}}(\boldsymbol{\Xi}) \\ \boldsymbol{\xi}' \in \boldsymbol{\Xi}, \boldsymbol{\xi}' \neq \boldsymbol{\xi}}} \left[ \sum_{i=s'(\boldsymbol{\xi}, \mathbf{x})+1}^{d} \mathbf{w}_i (\mathbf{U}\tilde{\mathbf{q}})_i < \sum_{i=1}^{d} \mathbf{w}_i (\mathbf{U}\tilde{\mathbf{q}}')_i \right] \\
&= \sum_{m=1}^{d} \binom{d}{m} \cdot p_{\text{masked}}^m \cdot (1 - p_{\text{masked}})^m \Pr_{\substack{(\boldsymbol{\xi}, \mathbf{x}) \sim \mathcal{V}_{m\text{-masked}}(\boldsymbol{\Xi}) \\ \boldsymbol{\xi}' \in \boldsymbol{\Xi}, \boldsymbol{\xi}' \neq \boldsymbol{\xi}}} \left[ \sum_{i=d-m+1}^{d} \mathbf{w}_i (\mathbf{U}\tilde{\mathbf{q}})_i < \sum_{i=1}^{d} \mathbf{w}_i (\mathbf{U}\tilde{\mathbf{q}}')_i \right]
\end{aligned}
$$

This is difficult to estimate because it involves sortings and summation over random variables of different distribution. We computed the numerical error $\hat{p}_{\text{err}}$ for $\boldsymbol{\xi}, \mathbf{x} \in [-1, 1]^d$ through Monte Carlo estimation. With dimensionality $d = 64$, $p_{\text{err}} \approx 6.8 \times 10^{-14}$, $1.6 \times 10^{-9}$, $1.2 \times 10^{-6}$ and $0.001$ for $p_{\text{masked}} = 0.1$, $0.25$, $0.5$ and $0.75$, respectively, when using the $\mathbf{w}$ derived from Eq. 10. This result verify that $s(\boldsymbol{\xi}, \mathbf{x}) = \mathbf{w}^\top \mathbf{U}\tilde{\mathbf{q}}$ achieves correct retrieval with high probability.

The main point of this section is that achieving optimal correct retrieval is too strict so that it forces us to abandon other properties. However, optimal correct retrieval itself is a good property when the underlying variant distribution is known *a priori*, but this is not always the case. Therefore, a good design choice of adaptive similarity should consider not only *optimality*, but also *adaptivity*, *learnability*, *efficiency*.

### A.3.2 On More Complicated Variant Distribution

Variant distributions discussed so far are simple. In previous analysis, we assumes that all memory variants follows a similar distribution. For example, $(\boldsymbol{\xi}, \mathbf{x}) \sim \mathcal{V}_{\text{noisy}}(\boldsymbol{\Xi})$ ensures that all memory patterns generates a noisy variants, sharing a similar $p_{\mathcal{V}}(\mathbf{x}|\boldsymbol{\xi})$. We say a variant distribution *general* if knowing $p_{\mathcal{V}}(\mathbf{x}|\boldsymbol{\xi}_i)$ is equivalent to knowing $p_{\mathcal{V}}(\mathbf{x}|\boldsymbol{\xi}_j)$ for arbitrary $i, j \in [N]$ and $i \neq j$. In other words, we can obtain $p_{\mathcal{V}}(\mathbf{x}|\boldsymbol{\xi}_j)$ by substituting $\mathbf{x}_j$ with $\mathbf{x}_i$. However, there could be cases where each memory patterns generates variants quite differently. Intuitively, we say each pattern is generates on their own, meaning that $p_{\mathcal{V}}(\mathbf{x}|\boldsymbol{\xi})$ has completely different form for different memory patterns, and we call such memory variant *isolated*.

By looking closely to the adaptive similarity function:

$$
s(\boldsymbol{\xi}, \mathbf{x}) = \mathbf{w}^\top \mathbf{U}\tilde{\mathbf{q}}
$$

One can see that it uses a universal weight $\mathbf{w}$ for all pairs of $(\boldsymbol{\xi}, \mathbf{x}) \sim \mathcal{V}(\boldsymbol{\Xi})$, assuming that the variant distribution it is trying to model is general. However, such limiltaion can be easily broken by introducing more weights. For instance, we use separate weights for different memory patterns, i.e., for $N$ memory patterns, we spare weight $\mathbf{w}_k$ to memory pattern $\boldsymbol{\xi}_k$. Therefore, the adaptive similarity that can suit isolated variant distribution look like:

$$
s(\boldsymbol{\xi}_k, \mathbf{x}) = \mathbf{w}_k^\top \mathbf{U}\tilde{\mathbf{q}}
$$

This can fit each isolated likelihood $p_{\mathcal{V}}(\mathbf{x}|\boldsymbol{\xi}_k)$ as the weights are no longer shared. But this rises more problem: (1) it requires more samples, and it might be impossible in real-world scenarios. (2) it requires samples generated by each memory pattern $\boldsymbol{\xi}_k$, as each individual weight $\mathbf{w}_k$ is optimized by samples involving $\boldsymbol{\xi}_k$.

Additionally, the adaptive similarities are a family of similarity measures that can fit to the variant distribution through sampling, it is not solely Eq. 4. When proving the optimal correct retrieval for noisy, masked, and biased variants, we propose more adaptive similarity as theoretical tools. We wrote Eq. 4 in the main text simply because it is the most effective ones for retrieving under noisy, masked, biased, and mixed settings, and it requires minimum trainable weight, as examined in ablation study (Appendix A.4.6).

One more thing is that the priori $p_\mathcal{V}(\boldsymbol{\xi}|\mathbf{x})$ are often ignored in this work. Well, it should not be ignored in all cases. However, we can use a bias term $\mathbf{b} \in \mathbb{R}^N$ and use $\mathbf{b}_k$ capture the occurrance of pattern $\boldsymbol{\xi}_k$. By assuming similarity is a logits of the posteriori $p_\mathcal{V}(\boldsymbol{\xi}|\mathbf{x})$ (e.g., $\log p_\mathcal{V}(\boldsymbol{\xi}|\mathbf{x})$), then we can see that $\arg\max_{\boldsymbol{\xi}' \in \boldsymbol{\Xi}}\{\log p_\mathcal{V}(\boldsymbol{\xi}|\mathbf{x})\} = \arg\max_{\boldsymbol{\xi}' \in \boldsymbol{\Xi}}\{\log p_\mathcal{V}(\mathbf{x}|\boldsymbol{\xi}') + \log p_\mathcal{V}(\boldsymbol{\xi}')\}$, and we set pass $\mathbf{s}(\boldsymbol{\Xi}, \mathbf{x}) + \mathbf{b}$ to the separation function so that $\mathbf{b}$ handles the term $\log p_\mathcal{V}(\boldsymbol{\xi})$.

### A.4 EXPERIMENTS

#### A.4.1 BASELINES AND METRICS

For memory retrieval test, we compare our Adaptive Hopfield network A-Hop against: Modern Hopfield network M-Hop (Ramsauer et al., 2021); Universal Hopfield network N-Hop (Millidge et al., 2022) with $\mathrm{sim}(\boldsymbol{\xi}, \mathbf{x}) = -\|\boldsymbol{\xi} - \mathbf{x}\|_1$ and $\mathrm{sep}(\cdot) = \arg\max(\cdot)$ as they report a leading performance for such configuration when masking out half of the dimensions in memory patterns (similar to masked variant, Def. 5); Kernelized Hopfield network U-Hop (Wu et al., 2024a) with the kernel optimized by separation loss proposed by Wu et al. (2024a); Kernelized Hopfield network $U_2$-Hop with the kernel optimized by loss defined by variant distribution (Eq. 5) rather than the separation loss; and Multi-Layer Perceptrons MLP that has 4 layers and a input dimensionality $d$, output dimensionality $N$, trying to fit $p_\mathcal{V}(\boldsymbol{\xi}|\mathbf{x})$ but unsatisfactory. We estimate the empirical retrieval accuracy of each model (Def. 9), and the mean squared error $\mathbb{E}_{(\boldsymbol{\xi},\mathbf{x}) \sim \mathcal{V}(\boldsymbol{\Xi})}\left[(\mathcal{T}(\mathbf{x}) - \boldsymbol{\xi})^\top(\mathcal{T}(\mathbf{x}) - \boldsymbol{\xi})\right]$. We wrote a generator that can generate noisy, masked, biased, and mixed variants.

For tabular classification test, we compare the A-Hop with a memory-based classifier (Appendix A.4.4) with: M-Hop uses the same classifier framework but the unlearnable similarity function; $U_2$-Hop that uses the same classifier framework but a different similarity function, and its kernel function is optimized by the classification loss; Extremely Randomized Trees (Geurts et al., 2006), or Extra Trees, a tree-based classifier; Random Forest (Breiman, 2001), yes, the famous Random Forest classifier; AdaBoost (Freund & Schapire, 1997), a classic boosting classifier; and XGBoost (Chen & Guestrin, 2016), an enhanced Gradient Boosting Decision Tree. All models are tuned with a 5-fold cross-validation on the training set. We measure the test accuracy for the dataset with a positive sample rate $0.2 < \% \mathrm{pos} < 0.8$, and the ROC-AUC score otherwise, following the settings in Wang et al. (2025).

For image classification task, we follow the settings in Wu et al. (2024a), where they replaced the attention component with a HopfieldLayer (Ramsauer et al., 2021). We experiment by integrating A-Hop, M-Hop, and U-Hop into the HopfieldLayer. Similarly, we follow the settings in Ramsauer et al. (2021); Hu et al. (2023), where they use HopfieldPooling for multiple instance learning. We integrate A-Hop, M-Hop, and U-Hop to HopfieldPooling, and run a 5-fold cross-validation to report the mean ROC-AUC of all folds as the result.

For all experiments, we report the results with the mean and standard deviation of five runs.

#### A.4.2 DATASETS

We used a total of 12 datasets to assess the performance of A-Hop on tasks including tabular classification (Adult, Bank, Vaccine, Purchase, and Heart), and image classification (CIFAR 10, CIFAR 100, and Tiny ImageNet), and multiple instance learning (Tiger, Fox, Elephant, UCSB).

**Adult** (Becker & Kohavi, 1996) The prediction task for this dataset is to classify individuals' income levels as either above or below $50,000 annually. The data was extracted by Barry Becker from the 1994 Census database.

**Bank** (Moro et al., 2014) To predict the success of a term deposit subscription, this dataset records the outcomes of telemarketing campaigns from a banking institution in Portugal.

**Vaccine** (Bull et al., 2016) We use this dataset from a DrivenData competition to predict if a person received a seasonal flu vaccine. The data consists of 26,707 survey responses detailing 36 behavioral and personal attributes.

**Purchase** (Sakar & Kastro, 2018) The objective with this dataset is to forecast the online shopping intentions of visitors to an e-commerce website, determining whether a user will proceed with a purchase.

**Heart** (Kaggle, 2018) This dataset facilitates the prediction of cardiovascular disease presence. It contains health-related data from 70,000 patients, as provided by Ulianova.

**CIFAR 10** (Krizhevsky & Hinton, 2009) is a classic image recognition dataset consisting of 60,000 $32 \times 32$ color images in 10 classes, with 6,000 images per class.

**CIFAR 100** (Krizhevsky & Hinton, 2009) is a more challenging version of CIFAR 10, containing the same number of images but split into 100 fine-grained classes.

**Tiny ImageNet** (Deng et al., 2009) is a subset of the ImageNet dataset designed for educational purposes. It contains 100,000 images from 200 classes, downsized to $64 \times 64$ pixels.

**Tiger, Fox, Elephant** (Deng et al., 2009) These are specific class subsets extracted from the large-scale ImageNet database, often used for fine-grained image classification tasks.

**UCSB** (Gelasca et al., 2008) This dataset is composed of 58 breast cancer histopathology images stained with Hematoxylin and Eosin (H&E). The primary challenge it presents is the accurate segmentation of individual cells from the complex tissue background, which is a critical precursor to classifying cells as benign or malignant.

### A.4.3 MEMORY RETRIEVAL

We first introduce the intensity of variant setting. A triplet $(d_{\text{mask}}, d_{\text{noise}}, d_{\text{bias}}) \in [0,1]^3$ is used to describe a variant setting. This mean that we will first add a Gaussian noise $\mathbf{n} \sim \mathcal{N}(\mathbf{0}, d_{\text{noise}}\mathbf{I})$ to a certain memory pattern $\boldsymbol{\xi} \in \Xi$ obtaining $\mathbf{x} \leftarrow \boldsymbol{\xi} + \mathbf{n}$. Then, we will choose $d \cdot d_{\text{mask}}$ indices, and set these indices of $\mathbf{x}$ to random numbers choosen uniformly from $[-1, 1]$. Finally, we will add a bias $\mathbf{d}$ to $\mathbf{x} \leftarrow \mathbf{x} + \mathbf{d}$ with $\mathbf{d}_i = \mathbf{s}_i \cdot d_{\text{bias}}$, where $\mathbf{s}_i$ is a random sign sampled uniformly from $\{-1, +1\}$. Then, the generator return $(\boldsymbol{\xi}, \mathbf{x})$ as a sample of $\mathcal{V}(\Xi)$. Therefore, different triplet describe different variant settings, and thus, corresponds to different variant distribution $\mathcal{V}(\Xi)$.

In the memory retrieval task, we use the retrieval dynamics written in Eq. 4. To learn the weight $\mathbf{w}$'s and $\beta$'s we use optimizer `Adam` for 200 epoches, and use a learning rate 0.1 in all settings. However, we argue that number of epoches (`N_epoch`) and learning rate (`lr`) should be tuned when applying `A-Hop` to other memory retrieval settings.

For `U-Hop` and `U₂-Hop`, we tuned them carefully. For the original `U-Hop`, it optimize a separation loss, and we try to make it as small as possible. However, the retrieval accuracy of `U-Hop` is not satisfactory, and it is reasonable since it is not optimized for correct retrieval, but for $\epsilon$-retrieval. We found that the domain of the memory pattern in their experiments is $[0, 1]^d$, and it is harmful to dot product-based methods (think of using only $2^{-d}$ of the space) as dot product highly relies on signs. Therefore, for fair comparision, we sample the random vectors uniformly from $[-1, 1]^d$ and change the domain of pixels in MNIST images to $[-1, 1]$.

### A.4.4 TABULAR CLASSIFICATION

We develop a memory-based model for tabular classification that takes advantage of the excellent memory retrieval effectiveness of associative memories. The insight is that classification is hierarchical, and we divide instances into *cases*, and further classify cases into the final class. For instance, there are type I diabetes and type II diabetes, where each type here resembles the idea of cases. Another samples is that cats has plenty of breeds, and associative memory can capture specific idea of orange cat, or blue cat, while they have a hard time figuring out the generalize idea of cats. So, we let associative memory match instances into cases by choosing some instances in the training set as the representatives of cases, or use $K$-means cluster to produce such representatives, and pass the retrieval probability to a multi-layer perceptron (MLP) for classifying cases. That is, the model can be represented as $\mathbf{y} = \texttt{MLP}(\texttt{LayerNorm}(\text{sim}(\Xi, \mathbf{x})))$, and this can be trained on a conventional machine learning fashion by minimizing classification loss:

$$\mathcal{L}(\mathcal{D}) = \mathbb{E}_{(\mathbf{x},\mathbf{y}) \sim \mathcal{D}}[(\mathbf{y} - \hat{\mathbf{y}})^2] \tag{12}$$

We no longer tune weights in adaptive memories using loss defined in Eq. 5, as they can be tuned using the classification loss when participating into going forward in the network.

We tuned the hyperparameters by grid searching on the training data:

Table 6: Hyperparameters tuned in tabular classification

| Name | Domain |
|---|---|
| N_epoch | $\{50, 100, 150, 200, 250\}$ |
| batch_size | $\{100, 1000\}$ |
| lr | $\{2 \cdot 10^{-3}, 10^{-3}, 5 \cdot 10^{-4}, 2 \cdot 10^{-4}, 10^{-4}\}$ |
| init | Cluster, No cluster |

### A.4.5 IMAGE CLASSIFICATION AND MULTIPLE INSTANCE LEARNING

For image classification, we follow the settings in Wu et al. (2024a), and place the adaptive similarity to the Hopfield layer inside a image Transformer. For M-Hop and U-Hop, we use the hyperparameter suggested in Wu et al. (2024a), and find the optimal number of epoch and learning rate for A-Hop via grid search. We run five iterations of separate loss optimization for A-Hop before testing.

Table 7: Hyperparameters tuned in image classification

| Name | Domain |
|---|---|
| N_epoch | $\{25, 40, 50\}$ |
| lr | $\{10^{-3}, 10^{-4}\}$ |

For multiple instance learning, we follow the settings in Hu et al. (2023), and use HopfieldPooling as the backbone. Similarly, we place the adaptive similarity in the core Hopfield component replacing the fixed dot product. All experiments are run on a 5-fold validation, and the ROC-AUC scored is taken as the mean of all folds.

Table 8: Hyperparameters tuned in multiple instance learning

| Name | Domain |
|---|---|
| N_epoch | $\{10, 20, 40\}$ |
| lr | $\{10^{-3}, 10^{-4}, 10^{-5}\}$ |
| lr_decay | $\{0.9, 0.75\}$ |

### A.4.6 ABLATION STUDY

We conduct four different ablation studies to see the effective of different components.

In the first experiment (Table 9), we tested if sorting the dimension-wise similarity vector $\mathbf{q}$ and the upper-right triangle matrix $\mathbf{U}$ is needed. The results shows that both of them are necessay for high retrieval accuracy.

Table 9: Retrieval accuracy (↑) and error (↓) between unsorted and sorted $\mathbf{q}$. Each cell contains the mean accuracy or error with standard deviation in a smaller font. Results of the best-performing model are **bolded**.

| Conditions | | Synthetic ($d = 0.4$) | | Synthetic ($d = 0.5$) | |
|---|---|---|---|---|---|
| $\mathbf{q}$ sorted? | use $\mathbf{U}$? | Accuracy | Error | Accuracy | Error |
| ✗ | ✗ | $.5172_{\pm.034}$ | $.1900_{\pm.017}$ | $.2094_{\pm.022}$ | $.2658_{\pm.005}$ |
| ✗ | ✓ | $.5444_{\pm.007}$ | $.1665_{\pm.007}$ | $.1888_{\pm.015}$ | $.2738_{\pm.003}$ |
| ✓ | ✗ | $.6928_{\pm.034}$ | $.1173_{\pm.010}$ | $.3374_{\pm.025}$ | $.2324_{\pm.006}$ |
| ✓ | ✓ | $\mathbf{.7280}_{\pm.034}$ | $\mathbf{.1033}_{\pm.011}$ | $\mathbf{.3634}_{\pm.040}$ | $\mathbf{.2207}_{\pm.011}$ |

Next, we look for a better matrix $\mathbf{U}$ (Table 10), which is the core of similarity measure. Our results show that the upper-right triangle structure of $\mathbf{U}$ is optimal, while making $\mathbf{U}$ learnable and initialize

it with the upper-right triangle yields the best result, but we does not adopt learnable $\mathbf{U}$ in other experiments to keep minimum learnable parameters. Another finding is that a randomized matrix $\mathbf{U}$ is better than not having $\mathbf{U}$ (when $\mathbf{U} = \mathbf{I}$). Therefore, along with Table 9, we find that the footprint structure is essential to adaptive similarities.

Table 10: Retrieval accuracy ($\uparrow$) and error ($\downarrow$) between different configuration of $\mathbf{U}$. Each cell contains the mean accuracy or error with standard deviation in a smaller font. Results of the best-performing model are **bolded**, and the second are underlined.

| Conditions | | Synthetic ($d = 0.4$) | | Synthetic ($d = 0.5$) | |
|---|---|---|---|---|---|
| $\mathbf{U}$ learnable? | initialization? | Accuracy | Error | Accuracy | Error |
| ✗ | Random | $.6928_{\pm.015}$ | $.1147_{\pm.005}$ | $.3340_{\pm.019}$ | $.2305_{\pm.005}$ |
| ✗ | $\mathbf{I}$ | $.6802_{\pm.013}$ | $.1194_{\pm.004}$ | $.3458_{\pm.025}$ | $.2298_{\pm.008}$ |
| ✗ | $\mathbf{U}$ | $\underline{.7242}_{\pm.016}$ | $\underline{.1056}_{\pm.007}$ | $\underline{.3600}_{\pm.024}$ | $.2272_{\pm.005}$ |
| ✓ | Random | $.7176_{\pm.027}$ | $.1087_{\pm.007}$ | $.3414_{\pm.023}$ | $.2292_{\pm.006}$ |
| ✓ | $\mathbf{I}$ | $.7114_{\pm.019}$ | $.1096_{\pm.006}$ | $.3528_{\pm.021}$ | $\underline{.2270}_{\pm.005}$ |
| ✓ | $\mathbf{U}$ | $\mathbf{.7280}_{\pm.034}$ | $\mathbf{.1033}_{\pm.011}$ | $\mathbf{.3634}_{\pm.040}$ | $\mathbf{.2207}_{\pm.011}$ |

We also tested the effect of different footprint with different base similarity (see Table 11). Results shows that two footprints (dis and dot) is always better than one alone, and we found that $\mathrm{ftpt}_{\mathrm{dis}}$ performs better than $\mathrm{ftpt}_{\mathrm{dot}}$ in this setting. However, even though retrieval accuracy degrades by removing one of the footprint, using $\mathrm{ftpt}_{\mathrm{dis}}$ or $\mathrm{ftpt}_{\mathrm{dot}}$ only is still better than other Hopfield networks.

Table 11: Retrieval accuracy ($\uparrow$) and error ($\downarrow$) between different usage of footprint. Each cell contains the mean accuracy or error with standard deviation in a smaller font. Results of the best-performing model are **bolded**.

| Conditions | | Synthetic ($d = 0.4$) | | Synthetic ($d = 0.5$) | |
|---|---|---|---|---|---|
| use $\mathrm{ftpt}_{\mathrm{dis}}$? | use $\mathrm{ftpt}_{\mathrm{dot}}$? | Accuracy | Error | Accuracy | Error |
| ✗ | ✓ | $.5926_{\pm.016}$ | $.1517_{\pm.005}$ | $.2520_{\pm.027}$ | $.2515_{\pm.004}$ |
| ✓ | ✗ | $.6458_{\pm.042}$ | $.1317_{\pm.011}$ | $.3286_{\pm.023}$ | $.2326_{\pm.007}$ |
| ✓ | ✓ | $\mathbf{.7242}_{\pm.016}$ | $\mathbf{.1056}_{\pm.007}$ | $\mathbf{.3600}_{\pm.024}$ | $\mathbf{.2272}_{\pm.005}$ |

Finally, we tested the number of samples needed for adaptive similarity to mimic the variant distribution. It looks like training on only $512$ samples provides a good enough adaptive similarity for $2048$ $64$-dimension memory patterns.

Table 12: Retrieval accuracy ($\uparrow$) and error ($\downarrow$) between different number of samples provided for learning. Each cell contains the mean accuracy or error with standard deviation in a smaller font. Results of the best-performing model are **bolded**.

| Conditions | Synthetic ($d = 0.4$) | | Synthetic ($d = 0.5$) | |
|---|---|---|---|---|
| number of samples? | Accuracy | Error | Accuracy | Error |
| $512$ | $.7204_{\pm.029}$ | $.1077_{\pm.010}$ | $.3541_{\pm.028}$ | $.2311_{\pm.004}$ |
| $\infty$ | $\mathbf{.7242}_{\pm.016}$ | $\mathbf{.1056}_{\pm.007}$ | $\mathbf{.3600}_{\pm.024}$ | $\mathbf{.2272}_{\pm.005}$ |

### A.4.7 MULTIMODAL SETTINGS

We designed a multimodal memory retrieval task to further test the memory retrieval capacity of different associative memories. In this setting, we have $M$ different modalities, and $N$ different *concepts*. For each concept $1 \le i \le N$, it has $M$ different patterns $\boldsymbol{\xi}_{i,1}, \boldsymbol{\xi}_{i,2}, \cdots, \boldsymbol{\xi}_{i,M}$ as its $M$ different representations in different modalities. Furthermore, each modality $1 \le j \le M$ has a unique encoder $\mathbf{A}_j$, which encodes a pattern $\boldsymbol{\xi}_{i,j} \in \mathbb{R}^d$ in the $j$-th modality into a hidden state

Table 13: Retrieval accuracy ($\uparrow$) between models in the multimodal memory retrieval task. Each cell contains the mean accuracy with standard deviation in a smaller font. Results of the best-performing model are **bolded**.

| Model | | Number of concepts | | | |
|---|---|---|---|---|---|
| Phrase 1 | Phrase 2 | 128 | 256 | 512 | 1024 |
| M-Hop | M-Hop | $.173_{\pm.02}$ | $.092_{\pm.01}$ | $.044_{\pm.01}$ | $.021_{\pm.00}$ |
| U$_2$-Hop | U$_2$-Hop | $.574_{\pm.04}$ | $.349_{\pm.02}$ | $.146_{\pm.02}$ | $.081_{\pm.02}$ |
| U$_2$-Hop | A-Hop | $.636_{\pm.03}$ | $.465_{\pm.02}$ | $.276_{\pm.03}$ | $.183_{\pm.02}$ |
| A-Hop | U$_2$-Hop | $.904_{\pm.02}$ | $.709_{\pm.03}$ | $.492_{\pm.04}$ | $.328_{\pm.02}$ |
| A-Hop | A-Hop | $\mathbf{.997}_{\pm.00}$ | $\mathbf{.998}_{\pm.00}$ | $\mathbf{.993}_{\pm.00}$ | $\mathbf{.988}_{\pm.00}$ |

$\mathbf{h}_{i,j} \in \mathbb{R}^{d_h}$, by setting $\mathbf{h}_{i,j} = \mathbf{A}\boldsymbol{\xi}_{i,j}$. Here, $d$ is the dimensionality of each pattern, and $d_h$ is the dimensionality of each hidden state, and we can see that $\mathbf{A}_j \in \mathbb{R}^{d \times d_h}$ for all $1 \leq j \leq M$.

The objective of this task is that: given a hidden state variant of a unknown concept and unknown modality, find its corresponding patterns in all modalities. Formally, let $\mathbf{h}' = \mathbf{h}_{i,j} + \boldsymbol{\epsilon}$ for some unknown concept $1 \leq i \leq N$ unknown modality $1 \leq j \leq M$, and a noise vector $\boldsymbol{\epsilon} \in \mathbb{R}^{d_h}$ drawn from the normal distribution $\mathcal{N}(\mathbf{0}, \sigma\mathbf{I})$ (the isotorpic noisy variant, see Def. 4). Then, $\mathbf{h}'$ will be the query of the retrieval task, and $M$ patterns $\mathbf{y}_{1\cdots M}$ should be retruned for each modalities. Finally, the retrieval is correct is the following holds (i.e., correct retrieval is obtained in each modality):

$$\forall j, \ 1 \leq j \leq M, \ \text{s.t.} \ \underset{\boldsymbol{\xi}' \in \boldsymbol{\Xi}_j}{\arg\min} \left\{ \|\mathbf{y} - \boldsymbol{\xi}'\|_2 \right\} = \underset{\boldsymbol{\xi}' \in \boldsymbol{\Xi}_j}{\arg\max} \left\{ p_{\mathcal{V}}(\boldsymbol{\xi}'|\mathbf{x}) \right\} = \boldsymbol{\xi}_{i,j}$$

Here, $\boldsymbol{\Xi}_j \triangleq [\boldsymbol{\xi}_{1,j}; \boldsymbol{\xi}_{2,j}; \cdots ; \boldsymbol{\xi}_{N,j}]$ collects all concepts' pattern in the $j$-th modality.

To address this challenge, the retrieval process requires two distinct associative memories. **The first associative memory** stores $N$ patterns in form of $\text{concat}(\mathbf{h}_{i,1}, \mathbf{h}_{i,2}, \cdots, \mathbf{h}_{i,M})$ (i.e., the concatenation of $M$ $d_h$-dimensional vectors into one $M \times d_h$ dimensional vector), remembering all the hidden states of the same concepts. When a query $\mathbf{h}'$ comes in, a vector of form $\text{concat}(\mathbf{h}', \mathbf{h}', \cdots, \mathbf{h}')$ is given to the associative memory, it will find the corresponding concept $i$, such that one of the $\mathbf{h}_{i,1}, \mathbf{h}_{i,2}, \cdots, \mathbf{h}_{i,M}$ is close to $\mathbf{h}'$. **The second associative memory** stores $N \times M$ patterns in form of $\text{concat}(\mathbf{h}_{i,j}, \boldsymbol{\xi}_{i,j})$ for all $1 \leq i \leq N$ and $1 \leq j \leq M$, serves as the decoder that finds the original pattern $\boldsymbol{\xi}_{i,j}$ from a hidden state $\mathbf{h}_{i,j}$. When a query of form $\text{concat}(\mathbf{h}, \mathbf{v})$ is given to it ($\mathbf{v} \in \mathbb{R}^d$, the mask, can be any arbitrary $d$-dimensional vector), it will find the corresponding hidden state-pattern pair $\text{concat}(\mathbf{h}_{i,j}, \boldsymbol{\xi}_{i,j})$ so that $\mathbf{h}$ is close to $\mathbf{h}_{i,j}$ and the original pattern is restored as $\boldsymbol{\xi}_{i,j}$. Therefore, an additional $M$ queries to the second associative memory will retrieve all $M$ patterns of the $i$-th concept in all modalities, respectively.

For an intuitive example, let $\mathbf{h}'$ stands for the textual embedding (hidden state) of *inhaler*, which (ideally) is a variant of the textual embedding of *inhalator* (represented by $\mathbf{h}_{k,1}$). So, when passing $\text{concat}(\mathbf{h}', \mathbf{h}', \mathbf{h}')$ to the first associative memory, and it will retrieve the corresponding hidden states $\text{concat}(\mathbf{y}_1, \mathbf{y}_2, \mathbf{y}_3)$ of inhalators in different modalities, which should satisfy that $\text{concat}(\mathbf{y}_1, \mathbf{y}_2, \mathbf{y}_3) \approx \text{concat}(\mathbf{h}_{k,1}, \mathbf{h}_{k,2}, \mathbf{h}_{k,3})$. Here, $\mathbf{h}_{k,1}$ is the textual embedding of inhalator, $\mathbf{h}_{k,2}$ is the visual embedding of inhalator, and $\mathbf{h}_{k,3}$ is the acoustic embedding. Then, we can retrieve the textual, visual, and acoustic pattern (i.e., $\boldsymbol{\xi}_{k,1}$, $\boldsymbol{\xi}_{k,2}$ and $\boldsymbol{\xi}_{k,3}$) of *inhalator* by querying the second associative memory (the decoder) with $\text{concat}(\mathbf{y}_j, \mathbf{v})$ three times for all $1 \leq j \leq 3$.

In our experiment, we let $M = 3$ (i.e., three modalities), and $N \in \{128, 256, 512, 1024\}$ (varying number of patterns), $d = 32$ (each pattern is a 32-dimensional vector), and $d_h = 16$ (each hidden state is a 16-dimensional vector). Furthermore, each pattern $\boldsymbol{\xi}_{i,j}$ is generated randomly and independently (i.e., each dimension of $\boldsymbol{\xi}_{i,j}$ follows the uniform distribution on $[-1, 1]$), and the encoder matrix $\mathbf{A}_j$ is generated similarly. For each query, a certain hidden state $\mathbf{h}_{i,j}$ is randomly picked from all $N \times M$ hidden states, and a noise $\boldsymbol{\epsilon} \sim \mathcal{N}(\mathbf{0}, \sigma\mathbf{I})$ with $\sigma = 0.2$ is added to it (i.e., $\mathbf{h}' = \mathbf{h} + \boldsymbol{\epsilon}$).

We use different associative memories as the first (Phrase 1) and second (Phrase 2) associative memory mentioned above. In Table 13, we can observe that A-Hop greatly outperforms all other combinations in all settings. For all other combinations other than A-Hop + A-Hop (last row in Table 13), a sharp degeneration in retrieval accuracy can be observed as the number of concepts ($N$)

doubles. However, `A-Hop` + `A-Hop` retains a 98.8% accuracy even for $N = 1024$ concepts, while the runner-up (`A-Hop` + `U₂-Hop`) only scores 32.8%. By observing the last four rows in Table 13, one can tell that the retrieval accuracy increases as the involvement of `A-Hop` increases, which indicates that `A-Hop` is a better surrogate for the underlying variant distribution in both phrases.

### A.4.8 TIME COMPLEXITY

Table 14: Training time ($\downarrow$) and inference time ($\downarrow$) of `A-Hop` with and without sorting. Ratio denotes the quotient of the runtime of sorted version divided by the unsorted version.

| sort? | Scaling dimensionality $d$ | | | Scaling number of patterns $N$ | | |
|---|---|---|---|---|---|---|
| $d$ | 64 | 128 | 256 | 64 | 64 | 64 |
| $N$ | 2048 | 2048 | 2048 | 2048 | 4096 | 8192 |
| Training time (ms, $\downarrow$) | | | | | | |
| ✓ | 840 | 1573 | 3609 | 840 | 1531 | 2949 |
| ✗ | 2093 | 3059 | 8224 | 2093 | 4011 | 7983 |
| ratio | 2.49× | 1.95× | 2.27× | 2.49× | 2.62× | 2.71× |
| Inference time (ms, $\downarrow$) | | | | | | |
| ✓ | .355 | .368 | .414 | .355 | .446 | .570 |
| ✗ | .287 | .318 | .326 | .287 | .364 | .467 |
| ratio | 1.23× | 1.15× | 1.27× | 1.23× | 1.23× | 1.22× |

When conducting memory retrieval for a query $\mathbf{x}$, we need to find the adaptive similarity between $\mathbf{x}$ and all $N$ stored memory patterns $\boldsymbol{\xi}_{1\cdots N}$. Computation of the adaptive similarity requires sorting the dimemsion-wise similarity vector $\mathbf{q}_{1 \cdot d}$, which needs $\mathcal{O}(d \log d)$ time for comparison-based sorting algorithms, and is the time complexity bottleneck for the proposed adaptive similarity (Eq. 4). Therefore, the overall time complexity of memory retrieval is $\mathcal{O}(Nd \log d)$ for calling the adaptive similarity for $N$ times, which is higher than that of the Hopfield network with fixed proximity-based similarity ($\mathcal{O}(Nd)$).

We conducted a runtime comparison experiment of sorted and unsorted adaptive similarity, and the results are listed in Table 14. In the experiment, each model (sorted and unsorted) is trained for 200 epoches with the same batch size 256, and the same learning rate 0.01, and optimizer `AdamW`. For the inference phrase, each model performs 2000 memory retrieval tasks simultaneously (in one batch) with the same configuration (but different sorting operation). The experiment is run on a NVIDIA GeForce GTX 3090 GPU, and the time elapsed on each task is measured by `torch.cuda.Event.elapsed_time(end_event)`.

As listed in Table 14, training a sorted `A-Hop` is merely 2 times slower than a unsorted one, and it needs at most 30% more time during the inference phrase. This is smaller than what is expected (3× v.s. 8×, as $\log_2 256 = 8$), suggesting that `A-Hop` is light-weight. As written in the official documentation, `torch.sort()` is implemented via radix sort, which might be the reason why the adaptive similarity is faster than expected.

