# OpenReview forum: "Adaptive Hopfield Network: Rethinking Similarities in Associative Memory"
_ICLR.cc/2026/Conference — ICLR 2026 Poster_

### Official Review · Reviewer_ohC5 · 2025-10-29

**Soundness:** 3
**Presentation:** 3
**Contribution:** 2
**Rating:** 6
**Confidence:** 5

**Summary:**

This paper introduces an Adaptive Hopfield network (A-Hop) that learns a context-dependent similarity measure for content-addressable memory.
By reframing retrieval as a maximum a posteriori problem under a variant distribution, the authors derive an adaptive similarity function that approximates the likelihood of a stored pattern generating the query.
They prove that, with appropriate learned weights,
A-Hop achieves optimal correct retrieval for three common variant types (noisy, masked, biased).
Empirically, integrating this adaptive similarity into Hopfield networks yields state-of-the-art performance on synthetic memory retrieval tasks and on downstream benchmarks (tabular data, image classification, multiple-instance learning), outperforming prior Hopfield variants.

**Strengths:**

- Tackles a fundamental issue (retrieval correctness vs. proximity) with a clear probabilistic framework.

- Offers theoretical guarantees for optimal retrieval under well-defined generative scenarios (noise, masking, bias).  I skimmed through the proofs. Looks reasonable to me, but I didn't check them line-by-line.

- Extensive experiments across diverse tasks, showing consistent improvements over strong baselines. Code provided. I skimmed through them. Looks reasonable to me. I didn't check line-by-line nor run them at my end.

**Weaknesses:**

1. [major] Incremental novelty: Core idea is essentially learning a weighted distance (metric learning) for Hopfield retrieval similar to Wu & Hu et al 2024. It is a relatively straightforward extension of known concepts.

2. [minor] Theory vs. implementation gap: Proofs assume per-dimension weighting (unsorted footprint) for optimal retrieval, but the actual method uses sorted feature footprints without a formal optimality guarantee for that sorting. Please correct me if I am wrong.

3. [minor] Sorting overhead: Computing the similarity footprint requires sorting features ($O(d \log d)$ or $O(n \log n)$). This adds computational cost. The paper does not discuss runtime impact or scalability to very high-dimensional data.

4. [very minor] Reliance on supervision: The adaptive similarity weights are learned using ground-truth associations (either known memory-query pairs or class labels). The method may not apply directly in fully unsupervised retrieval settings without a way to obtain such training signals.

5. [very very minor, almost personal opinions] I kind of like this paper. It shows some solid efforts on theory and numerical validations. This might already be a good paper, but I really hope it is better, even great. But the clarity is not good enough in general. I feel it can be more precise and concise in many places. It will be great if the authors can polish more.

**Questions:**

1. where did you validate the design choices in the adaptive similarity? if there is no, please add

2. any runtime comparison (or complexity analysis)? please Report the computational overhead of A-Hop. should be at least provided to show how the $O(d \log d)$ footprint computation scales with the dimensionality and number of memories.

LLM disclaimer: I used LLM to polish my language.

---

> ### Author Response · Authors · 2025-11-25
> **[1/2] Reply to Reviewer ohC5**
>
> We appreciate the reviewer's effort to engage with our work in depth and their thoughtful questions, which have helped us stengthen our clarifications and contributions. Here are the responses to your concerns.
>
> > **1. Incremental novelty: Core idea is essentially learning a weighted distance for Hopfield retrieval similar to Wu & Hu et al. (2024).**
>
> We are not sure which Wu & Hu et al. (2024) paper you mentioned, and we assume it is *Uniform Memory Retrieval with Larger Capacity for Modern Hopfield Models* (Kernelized Hopfield network, or `K-Hop` in this work Table 1).
>
> We would like to share 4 noteworthy differences between `A-Hop` and `K-Hop`:
>
> * **Retrieval Objectives**
> `K-Hop` aims for $\epsilon$-retrieval (Def. 1), which cannot guarantees the correctness (Def. 3) of memory retrieval.
> In contrast, `A-Hop` use an adaptive similarity that aims for correct retrieval, so that its behavior will align with the variant distribution, satisfying the context's needs.
> Moreover, the core idea of `A-Hop` is to extract and exploit information from subspaces, and use $d$ informative subspaces as evidence to guide model make right decision on whether $\boldsymbol \xi$ generates $\mathbf x$ or not. While the core idea of `K-Hop` is to use a kernel map $\boldsymbol \Phi$ to make $\boldsymbol \Phi \boldsymbol \Xi$ distant apart, and be arraged in spherical codes in the most ideal case.
> Although they both have parameters, but the meaning and usage of the parameters are different.
>
> * **Learning Objectives**
> `K-Hop` has the first learnable similarity (to the best of our knowledge), and it optimizes a separation loss so that kernelized patterns are far apart from each other, which can reduce the fuzzy memory problem.
> The separation loss depends on the stored patterns $\boldsymbol \Xi$ only, and the model has no access to context information (i.e., the variant distribution $\mathcal V(\boldsymbol \Xi)$).
> As a result, `K-Hop` cannot adapt its similarity to mimic the behavior of the variant distribution, so it is still fixed similarity.
> In contrast, `A-Hop` is optimized from samples drawn from the variant distribution, so that the similarity behaves like $p_{\mathcal V}(\mathbf x | \boldsymbol \xi)$ ensuring correct retrieval.
> Although they are both learnable, their learning objectives and training procedure are different.
>
> * **Similarity Calculation**
> The similarity metrics used by `K-Hop` is still dot product, while the similarity in `A-Hop` consists of Euclidean distance, sortings, cumulative sum, dot product, etc.
> `A-Hop` has more components and it could has better expressiveness.
> Additionally, `A-Hop`'s adaptive similarity is designed to find a balance within adaptivity (generalization ability to various distributions), optimality (often achieve correct memory retrieval), learnability (easy to learn, and as few parameter as possible), efficiency (time complexity). Please read the public comment **Frequently Asked Questions - Part 1 (Design Choice)** for more details on how `A-Hop` finds a optimal design choice.
>
> * **Energy Bounds**
> `A-Hop`'s energy function can be bounded from below $E(\mathbf x) \ge -\ln N - \frac{\Vert \mathbf b \Vert_2^2}{4}$, so patterns with large norms ($\Vert \mathbf x \Vert \to \infty$) are guaranteed to converge, while the energy function of `K-Hop` cannot be bounded from below.
>
> > **2. Theory v.s. implementation gap: Proofs assume unsorted footprint for optimal retrieval, but the actual method uses sorted footprint without a formal optimality guarantee for sorting.**\
> Also in [Question 1]: **Where did you validate the design choice?**
>
> It is easier to prove that unsorted footprint achieve theoretical optimal correct retrieval, but it is hard for the sorted footprint because it would fail several corner cases.
> However, we can show that the sorted version could achieve optimal correct retrieval with high probability.
> Additionally, there is a tradeoff between model's adaptivity and optimality.
> For example, unsorted version enjoys clear optimal correct retrieval for a certain variant distribution, but its performance will greatly downgrades outside that distrbution (high optimality poor adaptivity).
> But sorted version can achieve good (near-optimal) retrieval accuracy in different variant distribution (at least in noisy, masked, biased and mixed situations) (good optimality with good adaptivity).
>
> The detailed justification on desgin choice is written in the public comment **Frequently Asked Questions - Part 1 (Design Choice)**, please read it there.
> Besides that, the empirical results in ablation study (Appendix A.4.6) also validates the importance of the sorted footprint.

---

> > ### Author Response · Authors · 2025-11-25
> > **[2/2] Reply to Reviewer ohC5**
> >
> > > **3. Sorting overhead: Computing the similarity footprint requires sorting features, which adds computation cost.**\
> > Also in [Question 2]: **Show how the $\mathcal O(d \log d)$ footprint computations scales with the dimensionality and number of memories**
> >
> > Computing the similarity footprint requires sorting $\mathcal O(d)$ features, which results in a $\mathcal O(d \log d)$ time for finding footprint.
> >
> > We did a scalability experiments for $d \in \\{ 64, 128, 256 \\}$ and $N \in \\{ 2048, 4096, 8192 \\}$, and finds that sortings requires 200\% more time for training and 20\% time for inference.
> > The detailed discussion and results are written in public comment **Frequently Asked Questions - Part 2 (Time Complexity)**, please read it there.
> >
> > > **4. Reliance on supervision: The adaptive similarity may not apply directly in fully unsupervised retrieval settings**
> >
> > This is an interesting topic.
> > Adaptive similarity relies on observing the variant distribution $\mathcal V(\boldsymbol \Xi)$, while such observation no longer exists in fully unsupervised scenarios.
> > It seems like one should rely heavily on a pre-defined similarity measure if we cannot gain any knowledge on $\mathcal V(\boldsymbol \Xi)$.
> > Therefore, in this case, the adaptive similarity can only rely on the limited a priori knowledge we have on the underlying variant distribution.
> > It would also be interesting to find an unsupervised adaptive similarity that could take full advantage of the limited a priori knowledge.
> >
> > > **5. The clarity is not good enough in general, and it will be great if the authors can polish more.**
> >
> > We have greatly polish the manuscript.
> > * We now added a half-paged figure (Fig. 2) to the main text.
> > The top part of it visualized (a) `A-Hop`'s architecture, (b) adaptive similarity's training procedure, and (c) design choice of adaptive similarity.
> > The bottom part of it demonstrates a comprehensive image retrieval procedure via visual examples.
> > Two 16 by 16 pixel art of chicken is used as stored patterns $\boldsymbol \xi_1$ and $\boldsymbol \xi_2$, and a noisy + masked variant of $\boldsymbol \xi_1$ is generated as the query $\mathbf x$.
> > It is clear to human eyes that the correct retrieval result of $\mathbf x$ is $\boldsymbol \xi_1$ (they are visually very similar), but conventional proximity-based similarity finds $\mathbf x$ more similar to the other pattern $\boldsymbol \xi_2$.
> > However, adaptive similarity successfully assign higher similarity score to $\boldsymbol \xi_1$, and the illustrated retrieval procedure make every step of retrieval very clear.
> > * We have completely rewrite and polish the Methods (section 3.2, 3.3), and the Experiments part (section 4.1, 4.2, 4.3), so that the readability is improved and ambiguous expressions are revised.
> > We added more clearly analysis of the tabular results, and briefly discuss their implications in section 4.1, 4.2, 4.3.
> > Two additional paragraphs are added to the end of section 3.3 discussing theoretical results and insights of `A-Hop`, and a lot more linkage between appendix and main text is also created there.

---

> > > ### Comment · Reviewer_ohC5 · 2025-11-25
> > > **Score Update and Remaining Notes**
> > >
> > > I thank the authors for the detailed rebuttal. All of my concerns have been addressed, except that I still do not see the value of the energy function being bounded from below as a clear advantage over UHop. It also feels somewhat unusual to rename prior methods simply to fit the presentation of the paper. Anyway, these are minor points.
> > >
> > > I have raised my score to reflect my satisfaction, and encourage the authors to continue polishing the paper (proofreading 10+ times if needed).

---

> > > > ### Comment · Reviewer_ohC5 · 2025-11-25
> > > >
> > > > To be precise, one area that would benefit from further revision is the justification of your design choices (as in [this global response](https://openreview.net/forum?id=HKSp4U69dy&noteId=rsPaGSZxmV)). I see the points you are trying to convey, and I appreciate the intention, but the wording needs more care to maintain rigor and precision. Several statements still need polishing to be fully accurate (you are about 80% there in the current draft). For example, when discussing optimality, there should be no tradeoff. otherwise, it is simply sub-optimal.

---

> > > > > ### Author Response · Authors · 2025-11-25
> > > > >
> > > > > Dear Reviewer ohC5,
> > > > >
> > > > > Thank you very much for your further and insightful engagement with our rebuttal.
> > > > >
> > > > > We take your point about precision and rigor in our justifications very seriously. You are absolutely right that our wording needs more care—particularly regarding statements about optimality, where, as you correctly note, it should either be optimal or not optimal. Therefore, we will consider using other noun for the "depth" criteria (e.g., accuracy or fidelity) to describe the how well the model can match the underlying distribution.
> > > > > We recognize the current explanation is rough and will dedicate significant effort to closing this gap in later versions.
> > > > >
> > > > > Thank you again for pushing us toward a more rigorous presentation. Your feedback has been invaluable.
> > > > >
> > > > > Warm regards,
> > > > >
> > > > > Authors of Submission13441

---

> > > > ### Author Response · Authors · 2025-11-25
> > > >
> > > > Dear Reviewer ohC5,
> > > >
> > > > Thank you very much for your thoughtful and timely engagement with our rebuttal and for raising your score. We greatly appreciate your constructive feedback throughout the review process.
> > > >
> > > > We acknowledge your point regarding the lower-bounded energy function. We would say that the energy landscapes behave differently, and such difference does not imply clear advantage for any of them.
> > > > Regarding the naming conventions, we appreciate your feedback. We will reconsider using `U-Hop` in our presentation to ensure it respects prior work while maintaining clarity.
> > > >
> > > > We will continuously and thoroughly polish the manuscript (including the Appendix), with multiple rounds of careful proofreading as you suggested. We will update it before the final submission (Dec. 3).
> > > >
> > > > Thank you again for your valuable time and insights, which have significantly improved our paper.
> > > >
> > > > Best regards,
> > > >
> > > > Authors of Submission13441

---

### Official Review · Reviewer_L7xN · 2025-10-30

**Soundness:** 2
**Presentation:** 3
**Contribution:** 2
**Rating:** 4
**Confidence:** 3

**Summary:**

This paper proposes an adaptive similarity metric to augment Hopfield Network-based storage and retrieval. It is a common observation that since the task is memory retrieval, if the query deviates from the stored patterns, then the retrieved pattern may not necessarily be 'semantically' close to the query. The authors analyze three particular query variants, namely, noisy variants of patterns, masked variants of patterns, and biased variants of patterns. The basic problem is to predict the most likely stored pattern given a query. The approach taken is to consider the query as a generative variant of the stored patterns and ask the question,  what is the maximum a posteriori probability that a given patttern is likely given the query. To estimate this, they propose an approach in which the similarity between query and stored pattern vectors are computed at the individual dimension level and the vectors are sorted. A combination of Euclidean and cosine distance are used and the softmax operator is applied to select the final stored pattern.

**Strengths:**

The beginning of the paper was on the right track in terms of addressing a problem since the current limitation of Hopfield networks was their ability to handle conceptual variants of the stored patterns. Modeling the query as a generative variant of the store patterns is also reasonable. Experimental results are indicating that the technique works under the modeled transformations of stored patterns. The proofs of the theorems are provided in the appendix which adds to some of the clarifications.

**Weaknesses:**

While the approach was well-motivated, the method presented was a bit under-whelming. Why would sorting along each dimensions help in finding similarity. It is also known that such operations can bring dis-similar vectors close together as well and create distractions. The tabular results are too abstract to interpret and not enough time is devoted in the paper. Having an illustration of the method through visual examples in case of image retrieval would strengthen the understanding of the  method.

The definitions and theorems are stated but not proved in the text which is fine but no reference to the appendix section is made.

**Questions:**

It would also be useful to address this problem in the context of cross-modal Hopfield retrieval, particularly using textual queries where it will be easier to show performance against truly conceptual variants. For example, if the stored model is a visual description is that of inhalator, queries such as inhaler should still be able to find them

---

> ### Author Response · Authors · 2025-11-25
> **[1/2] Reply to Reviewer L7xN**
>
> We appreciate the reviwers' constructive feedback and the opportunity to address their thoughtful questions, ensuring our work achieves its full potential.
> Here are the responses to thei unclear points and weaknesses you care about.
>
> > **1. Why would sorting along each dimensions help in finding similarity (as it may bring distractions)?**
>
> First of all, the sorting in adaptive similarity does not mess up indices.
> Notice that the sorting happens *after* a dimension-wise similarity is calculated, i.e., it has the form $\texttt{sort}(\mathbf g(\boldsymbol \xi, \mathbf x))$ for some $\mathbf g: \mathbb R^{d} \times \mathbb R^{d} \to \mathbb R^{d}$, while it is NOT in the form $\mathbf g(\texttt{sort}(\boldsymbol \xi), \texttt{sort}(\mathbf x))$.
> Therefore, for dimension $k \in [d]$, $\mathbf x_k$ is compared against $\boldsymbol \xi_k$, so it does not create distraction in dimension ordering.
> This procedure is illustrated in Fig. 2 (1) and (2).
>
> Second, sorting is not the core of adaptive similarity, and sorting with cumulative sum is needed only because it accelerates the calculation of the similarity footprint.
> We added Theorem 3. *Equivalence form for decomposable base similarity* (in Appendix A.2) which proves why the footprint of decomposable similarity can be computed in a first-sort-then-sum manner.
>
> Third, the core idea of adaptive similarity is to extract and exploit information from subspaces (i.e., space with dimensions restricted in $D \subseteq [d]$), and focus on the informative subspaces.
> It can be beneficial to extract information from all $2^d - 1$ subspaces, but it is impractical (computational intensive and hard to scale).
> So, footprint $\text{ftpt}\_{\text{sim}}(\boldsymbol \xi, \mathbf x)_{1 \cdots d}$ aims to find a representative subspace for each dimensionality (space size) $k \in [d]$, where the repsentative of $k$-dimensional subspace ensures that $\boldsymbol \xi$ and $\mathbf x$ is the closest (closeness defined by base similarity $\text{sim}(\cdot, \cdot)$) subspace.
> In other words, by focusing on the $k$-th element in the footprint (which is, $\text{sim}^{(k)}(\boldsymbol \xi, \mathbf x)$) the model can get rid of $d - k$ dimensions that $\boldsymbol \xi$ and $\mathbf x$ disagree on.
> This helps finding similarity as the model are inherently immuned to masked dimensions or heavily noisy dimensions.
>
> For detailed explanation, please read the public comment **Frequently Asked Questions - Part 1 (Design Choice)**, where we discuss how each component in the adaptive similarity (help model to find similarity, in general) affect model's adaptivity (generalization ability to various distributions), optimality (often achieve correct memory retrieval), learnability (easy to learn, and as few parameter as possible), efficiency (time complexity).
>
> > **2. The tabular results are too abstract to interpret and not enough time is devoted in this paper.**
>
> We have completely rewrite and polish the Methods (section 3.2, 3.3), and the Experiments part (section 4.1, 4.2, 4.3), so that the readability is improved and ambiguous expressions are revised.
> We added more clearly analysis of the tabular results, and briefly discuss their implications in section 4.1, 4.2, 4.3.
>
> > **3. Having an illustration of the method through visual examples in case of image retrieval would strengthen the understanding of the method.**
>
> We now added a half-paged figure (Fig. 2) to the main text.
> The top part of it visualized (a) `A-Hop`'s architecture, (b) adaptive similarity's training procedure, and (c) design choice of adaptive similarity.
> The bottom part of it demonstrates a comprehensive image retrieval procedure via visual examples.
> Two 16 by 16 pixel art of chicken is used as stored patterns $\boldsymbol \xi_1$ and $\boldsymbol \xi_2$, and a noisy + masked variant of $\boldsymbol \xi_1$ is generated as the query $\mathbf x$.
> It is clear to human eyes that the correct retrieval result of $\mathbf x$ is $\boldsymbol \xi_1$ (they are visually very similar), but conventional proximity-based similarity finds $\mathbf x$ more similar to the other pattern $\boldsymbol \xi_2$.
> However, adaptive similarity successfully assign higher similarity score to $\boldsymbol \xi_1$, and the illustrated retrieval procedure make every step of retrieval very clear.
>
> Additionally, we integrate Fig. 2 into the main text by referring to it when possible, so that it is possible to understand the formulas in an intuitive way.
>
> > **4. The definitions and theorems are stated but not proved in the text, but no reference to the appendix section is made.**
>
> The theorems and their corresponding proofs are now referred and briefly explained in the end of section 3.3, as we added two paragraphs discussing the theoretical results of `A-Hop`'s.

---

> > ### Author Response · Authors · 2025-11-25
> > **[2/2] Reply to Reviewer L7xN**
> >
> > > **5. Demnonstrate performance on cross-modal Hopfield retrieval.**
> >
> > As you suggested, we tried to desgin a multi-modal memory retrieval task.
> > In this settings, we have $m = 3$ different modalities, and there is $N = 1024$ concepts $c_{1 \cdots N}$.
> > For each concept $c_i$, it has 3 different patterns ($\boldsymbol \xi_{i, 1}$, $\boldsymbol \xi_{i, 2}$, $\boldsymbol \xi_{i, 3}$) for its 3 reprsentations in different modalities.
> > Each modality has a unique enocder $M_1$, $M_2$, and $M_3$, and a pattern's corresponding hidden state is defined as $\mathbf h_{i, j} = M_j \boldsymbol \xi_i$, for $1 \le i \le N$ and $1 \le j \le 3$.
> > The dimension of each pattern is $d = 32$, and the dimension of hidden state is $d_h = 16$, which is to say, $M_j \in \mathbb R^{32 \times 16}$ for all $1 \le j \le 3$.
> >
> > The objective of this task is that: given a hidden state variant of unknown concept and unknown modality, find its corresponding pattern of all 3 modalites.
> > For example, let $\mathbf h' = \mathbf h_{i, j} + \boldsymbol \varepsilon$ for some unknown $i, j$ and a noise $\boldsymbol \varepsilon \in \mathbb R^{d_h}$, the correct retrieval should be $\boldsymbol \xi_{i, 1}$, $\boldsymbol \xi_{i, 2}$, and $\boldsymbol \xi_{1, 3}$.
> >
> > To address this challenge, we need two associative memories.
> > The first one stores $N$ patterns in form of $\texttt{concat}(\mathbf h_{i, 1}, \mathbf h_{i, 2}, \mathbf h_{i, 3})$ for $1 \le i \le N$, store all the hidden states of the same concepts in different modality.
> > When a query of form $\texttt{concat}(\mathbf h', \mathbf h', \mathbf h')$ is given to it, the associative memory will find the corresponding concept $c_i$, such that one of $\mathbf h_{i, 1}$, $\mathbf h_{i, 2}$, and $\mathbf h_{i, 3}$ is close to $\mathbf h'$.
> > The second associative memory stores $3N$ patterns in form of $\texttt{concat}(\mathbf h_{i, j}, \boldsymbol \xi_{i, j})$ for all $1 \le i \le N$ and $1 \le j \le 3$, serves as the *decoder*.
> > When a query of form $\text{concat}(\mathbf h, \text{mask})$ (\text{mask} can be an arbitrary 32-dimensional vector) is given to it, the associative memory will find the corresponding hidden state-pattern pair $(\mathbf h_{i, j}, \boldsymbol \xi_{i, j})$ so that $\mathbf h$ is close $\mathbf h_{i, j}$, and the original pattern is restored as $\boldsymbol \xi_{i, j}$.
> >
> > For example, $\mathbf h'$ stands for the text embedding of *inhaler*, and $\mathbf h_{i, 1}$ stands for the text embedding of *inhalator*, and $\mathbf h_{i, 2}$ stands for the visual embedding of *inhalator*.
> > Then, for the first step $\texttt{concat}(\mathbf h', \mathbf h', \mathbf h')$ will be passed to the first associative memory, and it returns $\texttt{concat}(\mathbf h_{i, 1}, \mathbf h_{i, 2}, \mathbf h_{i, 3})$ where $c_i$ should (hopefully) be the concept of inhalator.
> > Then, $\texttt{concat}(\mathbf h_{i, 2}, \text{mask})$ is passed to the second associative memory, and it returns $\texttt{concat}(\mathbf h_{i, 2}, \boldsymbol \xi_{i, 2})$, where $\boldsymbol \xi_{i, 2}$ should be the visual description (image) of inhalator (if the retrieval is correct).
> > Then, one can pass $\texttt{concat}(\mathbf h_{i, 3}, \text{mask})$ to the second associative memory to retrieve patterns of inhalator in other modality.
> >
> > In this experiments, all $\boldsymbol \xi_{i, j}$ are generated randomly (each dimension follows the uniform distribution on $[-1, 1]$), and the enocder matrix is also randomly generated in the same manner.
> > Each query $\mathbf h'$ is gnerated by randomly picks a $\mathbf h_{i, j}$ and adds a Guassian noise to it.
> > The retrieval result is correct iff $\mathbf h_{i, 1}$, $\mathbf h_{i, 2}$, and $\mathbf h_{i, 3}$ are retrieved successfully (being assigned with highest similarity score, respectively).
> >
> > The results are listed in the following table. Phrase 1 denotes which model is used as the first associative memory, and Phrase 2 denotes the second.
> > We tested $N \in \{ 128, 256, 512, 1024 \}$ concepts, and report the retrieval accuracy.
> >
> > |Phrase 1|Phrase 2|128 concepts|256 concepts|512 concepts|1024 concepts|
> > |----------|----------|-----|-----|-----|------|
> > |`M-Hop`|`M-Hop`|.173±.02|.092±.01|.044±.01|.021±.00|
> > |`K₂-Hop`|`K₂-Hop`|.574±.04|.349±.02|.146±.02|.081±.02|
> > |`K₂-Hop`|`A-Hop`|.636±.03|.465±.02|.276±.03|.183±.02|
> > |`A-Hop`|`K₂-Hop`|.904±.02|.709±.03|.492±.04|.328±.02|
> > |**`A-Hop`**|**`A-Hop`**|**.997±.00**|**.998±.00**|**.993±.00**|**.988±.00**|
> >
> > We can observe that `A-Hop` greatly outperforms all other combinations.
> > A sharp degeneration in accuracy can be observed for all other combinations, while `A-Hop` + `A-Hop` retains a accuracy larger than $98\%$ even for $N = 1024$ concepts, while the runner-up only scores $32.8\%$.
> > By observing column 3-6, the accuracy increases as the involvement of `A-Hop` increases, which indicates that `A-Hop` is a better surrogate for the underlying variant distribution in both phrases (recall that `K₂-Hop` is optimized the `A-Hop`'s loss instead of the original separation loss).

---

### Official Review · Reviewer_67xP · 2025-10-31

**Soundness:** 3
**Presentation:** 3
**Contribution:** 3
**Rating:** 6
**Confidence:** 3

**Summary:**

This paper challenges the fixed, proximity-based similarity measures in traditional Hopfield networks, arguing they fail to capture context-dependent associations and cannot guarantee "correct retrieval" of a query's true origin. To solve this, the authors introduce the Adaptive Hopfield Network (A-Hop), which reframes retrieval as a probabilistic problem governed by a "variant distribution". A-Hop replaces the fixed metric with a learnable "adaptive similarity" that is built from a similarity footprint of sorted, dimension-wise similarities. The key advantage is that A-Hop's similarity measure is not fixed. It learns from samples to align with the task's specific context, such as noise or masking.

**Strengths:**

1. The paper's first strength lies in its novel reframing of the retrieval problem in associative memories through a probabilistic framework, and go beyond traditional epsilon-retrieval to a new concept term as "correct retrieval". The goal becomes finding the memory pattern that is the most probable origin of the query, based on a clear mathematical definition: maximizing the a posteriori probability

2. The authors demonstrate that A-Hop achieves state-of-the-art performance across a diverse set of four distinct tasks.

3. The paper is in general well-written.

**Weaknesses:**

1. My main concern is whether the proposed similarity footprint and adaptive similarity, $s(\xi,x) = w^{\top}U\tilde{q}$, can approximate the posterior distribution, about which the author spends a whole subsection (3.1) discussing the novelty of framing the retrieval problem in this probabilistic way. Maybe I miss some part of the content, but I feel like it lacks a strong theoretical justification for why this specific basis (a linear combination of cumulative sums of sorted dimension-wise similarities) is a universal approximator for an arbitrary likelihood function $p_{\nu}(x|\xi)$.

2. The authors write a detailed and insightful discussion section in the appendix (A.3), but this crucial context is never linked to or integrated into the main text. This discussion addresses fundamental questions (e.g., the trade-off between the provably optimal and the learnable formulation) that should be visible to the reader. It would be great if the authors could move the most critical parts of the discussion section into the main text, likely condensing and replacing some of the dense definition-heavy material in section 3.1.

**Questions:**

1. Does the author aim to release their code? The current link provided in the paper leads to a folder containing only a readme file.

2. Want to clarify the complexity of the proposed method compared to the modern Hopfield network. The proposed method requires a sort operation ($\mathcal{O}(d \log d)$) for each of the $N$ memory patterns during the similarity calculation. A modern Hopfield network (M-Hop) performs this in $\mathcal{O}(Nd)$ ($d$ for dot product and need to repeat for N memory patterns), but A-Hop increases this to at least $\mathcal{O}(Nd \log d)$, is my understanding correct?

---

> ### Author Response · Authors · 2025-11-24
> **Reply to Reviewer 67xP**
>
> We are grateful to the reviewers for their careful reading and valuable suggestions, which have significantly contributed to refining our research.
> Here are the responses to your points raised in the Weakness and Questions section.
>
> > **1. Whether $s(\boldsymbol \xi, \mathbf x) = \mathbf w^{\top} \mathbf U \tilde{\mathbf q}$ (sort then cumulative sum) can approximate arbitrary likelihood function $p_{\mathcal V}(\mathbf x | \boldsymbol \xi)$?**
>
> One word answer --- No.
> It is impossible for such simple form to approximate arbitrary memory variants.
>
> Please read **Frequently Asked Questions - Part 1 (Design Choice)** in the public comment, where we discuss why $s(\boldsymbol \xi, \mathbf x) = \mathbf w^{\top} \mathbf U \tilde{\mathbf q}$ serves as a good adaptive similarity.
>
> In short, we can say that the proposed adaptive similarity is a good approximator for common memory variants --- noisy, masked, biased, and noisy + masked.
> As theoretical results show that it achieve optimal correct retrieval with high probability, and empirical results (Table 2 and Fig. 3) shows that is has high retrieval accuracy on these variants.
>
> Additionally, approximating arbitrary likelihood function is like exploring the unknown with known concepts as building blocks.
> A direct metaphor is that space of consider arbitrary likelihood function is a vector space with infinity (or arbitrary large) dimensions, and each known concept (e.g., different similarity measure, different functions, anything mathematical tools that can be used to build a new similarity measure, etc.) is a base vector.
> And we say a likelihood function can be approximated if the corresponding point in the vector space can be represented by a linear combination of the basis (known concepts).
> Therefore, building similarities is like covering points in the vector space (of likelihoods) by spanning the basis vectors.
> While the known concept is finite, and the vector space's dimensionality seems infinite, so that the vector space cannot be fully covered.
> So, we may not be able to find a universal way to approximate any arbitrary variant distribution.
>
> > **2. The insightful discussion section is never linked or integrated into the main text.**
>
> Now it has been integrated to the end of Section 3.3, and we wrote two new paragraphs to breifly discuss the insights and theorems.
> Also, the link to corresponding proofs/discussion are added to the main text.
>
> Apart from these, we also polished our language in (rewrite) Methods and Experiments section, so that the readability is improved and ambiguous expressions are revised.
> Furthermore, added a new figure (Fig. 2) of A-Hop's and adaptive similarity's architecture, along with a complete visual example of how memory retrieval is proceed with associative memory.
>
> > **3. Does the author aim to release their code?**
>
> Yes, of course!
> We are currently cleaning up the working directories, and adding comments to the code, and it will be released within two days.
>
> > **4. Clarification on time complexity. Does `A-Hop` increase the time complexity from $\mathcal O(Nd)$ to $\mathcal O(N d \log d)$?**
>
> Your understanding is completely correct!
> We discussed the time complexity in public comment **Frequently Asked Questions - Part 2 (Time Complexity)**, please read it there.

---

> ### Comment · Reviewer_67xP · 2025-11-28
>
> Thanks for the quick and detailed response. Most of my concerns have been addressed: the added discussion in Section 3.3 is helpful, the anonymous code link now works, and the expanded complexity discussion and additional experiments look solid.
>
> I have one remaining clarification regarding correct retrieval. The main problem this work trying to solve is to perform "correct retrieval" defined in definition 3 that $\epsilon$-retrieval don't gurantee. In the end of appendix a.3.1, and also the FAQ in the rebuttal, the author state
>
> > *The main point of this section is that the optimality of correct retrieval is too strict so that achieving
> so force us to abandon good properties. Also, in most cases, some very corner cases set very high
> difficulty for achieve optimal. However, correct retrieval itself is a good property, but making every
> retrieval correct is too strict. Therefore, an open question is that how to define a sub-optimal correct
> retrieval standard so that it guarantees great memory retreival performance and leave us freedom.*
>
> , and also mentioned the proposed adaptive similarity is described as balancing four aspects (one of which corresponds to correct retrieval). Does this mean the proposed method still doesn't guarantee correct retrieval? But in theorem 1 it looks like the author do prove the the optimal correct retrieval for three types of memory pattern. I am a little confused on the proposed adaptive similarity (Eq.4) and its ability to solve the correct retrieval issue after reading the discussion. Relatedly, in line 1246-1249
> > *Therefore, achieving optimal correct retrieval for a certain variant distribution is not the only golden criteria for a similarity measure. More generally, we think good (adaptive) simliarty measures should be assessed from the following aspects:...*
>
> After this, it is a bit unclear to me what the primary target of the paper is.
> Is the main goal still to approximate correct retrieval as closely as possible? Or is the main contribution more about proposing a variant distribution based framework and introducing adaptive similarity as one design that perform well on the four criteria?
>
> ---
>
> Minor Suggestion:
>
> * In definition 7 when mentioning "correct retrieval" it might be helpful to link to definition 3.
>
> * In line 1325 there looks like a unfinished TODO that was not removed.
>
> * I think figure 2, part e would benefit by adding more caption to describe step 1-4, or at least link the corresponding main text to these four steps, so reader can by looking at the figure and find the corresponding explanation easily.
>
> * In definition 5 the dirac delta function is used before being defined
>
> * line 1320 -1322 state
> >  *Next, we show that the proposed adaptive similarity (Eq. 4) is suitable (has high optimality) for
> noisy + masked variant.*
> But I did not find the corresponding discussion related directly to Eq. 4 in the paragraphs that follow.

---

> ### Author Response · Authors · 2025-12-04
>
> Dear Reviewer 67xP,
>
> Thank you for your response and we appreciate that you looked very closely to every part of our manuscript.
>
> **The clarification regarding *correct retrieval* is that:**
>
> **(1)** correct retrieval is defined for examining whether or not the associative memory finds the memory pattern that is most associated to the query.
>
> **(2)** Adaptive similarity can have different forms, it can be very complicated or very simple.
> $s(\boldsymbol \xi, \mathbf x) = \mathbf w^{\top} \mathbf U \tilde{\mathbf q}$ (Eq. 4) is one of the design but not the only design of adaptive similarity.
> As an intuitive example:
> Some adaptive similarity can score 100 on variant distribution $\mathcal V_1$ but score 20 on variant distribution $\mathcal V_2$.
> A different adaptive similarity can score 100 on $\mathcal V_2$ but $20$ on $\mathcal V_1$.
> We want to find something that score 99 on both $\mathcal V_1$ and $\mathcal V_2$.
> However, $s(\boldsymbol \xi, \mathbf x) = \mathbf w^{\top} \mathbf U \tilde{\mathbf q}$ (Eq. 4) surpass every existing associative memory in every tested variant distribution (Section 4.1, Fig. 3 & Table 2)
>
> **(3)** Our design $s(\boldsymbol \xi, \mathbf x) = \mathbf w^{\top} \mathbf U \tilde{\mathbf q}$ can achieve optimal correct retrieval for noisy variant, biased variant, and masked variant with $\mathbf q_i$ bounded by $[\epsilon, \epsilon^{-1}]$, see **Appendix A.3.1 On Good Design Choice of Adaptive Similarity** `line 1376-1428`. It cannot achieve optimal correct retrieval for masked variant with unbounded $\mathbf q_i$, which we believe is impossible for continuous adaptive similarity functions, but we show that Eq. 4 will achieve high retrieval accuracy with high probability for unbounded masked variants.
> The proof to achieving optimal correct retrieval to general masked variants used a different (discrete) adaptive similarity so the optimality can hold.
>
> **(4)** The primary target of this paper is to (1) define *correct retrieval* and (2) find a similarity measure that has better memory retrieval accuracy (achieves correct retrieval more often compared to existing methods) and it can self-adapt to different underlying variant distribution $\mathcal V$ (so it achieves correct retrieval under different variant distributions, unlike existing models only performs well in noisy variants) by providing some samples drawn from $\mathcal V$.
>
> It is about not approximating *certain* variant distribution as close as possible.
> It is about having the capability to approximate *several* (ideally, *every*) variant distribution as close as possible (without knowing the distribution as *a priori*), while using as few learnable parameters as possible and having relatively small time complexity (as you said, performs well on the four criteria).
>
> **Regarding the minor suggestions you raised:**
> * In Def. 7, a link to the definition of *correct retrieval* (Def. 3) is added (`line 385`).
> * We have remove the [TODO] tag. Thank you for pointing it out!
> * We added a description of Fig. 2 (1) (2) (3) (4) to its caption (`line 352-355`), and linked it with the corresponding main text (`line 309-311, 323`).
> * The Dirac delta function is defined in Def. 5 (`line 252`).
> * The discussion on why the proposed adaptive similarity $s(\boldsymbol \xi, \mathbf x) = \mathbf w^{\top} \mathbf U \tilde{\mathbf q}$ is suitable for noisy + masked variant (because it can separate masked and un-masked elements, and deal with them differently) is added see **Appendix A.3.1 On Good Design Choice of Adaptive Similarity** `line 1340-1373`, along with why $s(\boldsymbol \xi, \mathbf x) = \mathbf w^{\top} \mathbf U \tilde{\mathbf q}$ also is suitable for noisy, masked, or biased variants.
>
> Warm regards,
>
> Authors of Submission13441

---

### Author Response · Authors · 2025-11-24
**Frequently Asked Questions - Part 1 (Design Choice)**

> **1. On good design choice of adaptive similarity (why a universal approximater for arbitary likelihood? why sorting? theory v.s. implementation gap)**

Achieving optimal correct retrieval is costly, as it requires designing weird similarity function for corner cases or use discrete function that makes the model unlearnable.
For instance, it is impossible for a continuous similarity function to achieve optimal correct retrieval for masked variants.
Therefore, achieving optimal correct retrieval for a certain variant distribution is not the only golden criteria for a similarity measure. More generally, we think good (adaptive) simliarty measures should follow these aspects:
* **Adaptivity** (breadth) Can the similarity measure adapt to a wide range of variant distribution?
* **Optimality** (depth) How accurate can the similarity measure predicts the variant distribution?
* **Learnability** (scale) Can this similarity measure be learned easily when some samples from the variant distribution is provided? How many learnable parameters it requires?
* **Efficiency** (speed)  What is the time complexity of calculating the similarity score?

Maximizing all four criteria is ideal but not likely.
Good adaptivity may imply poor optimality and vice versa; good adaptivity and good optimality may induce poor learnability and poor efficiency.
Also, adaptive similarities that achieves noisy variants (discussed in Lemma 2) has poor performance on masked or noisy variants, the similar goes for similarites discussed in Lemma 3 (masked) and Lemma 4 (noisy); complete optimality limits their generalization ability (adaptivity) to other distribution.

Adaptive similarity is not limited to the one proposed in this paper (Eq. 4 and Fig. 2), one can design different adaptive similarities for different tradeoffs in the 4 criteria listed above.
However, the proposed adaptive similarity $\beta_{\text{dis}} \mathbf w_{\text{dis}} \text{ftpt}\_{\text{dis}}(\boldsymbol \Xi, \mathbf x)$ $+ \beta_{\text{dot}} \mathbf w_{\text{dot}} \text{ftpt}\_{\text{dot}}(\boldsymbol \Xi, \mathbf x)$ is under full consideration of all 4 criteria (adaptivity, optimality, learnability, and efficiency) and aims for the ``Pareto-optimal balance''.
We will explain the design choice of this adaptive similarity from the perspectives of these 4 criteria.

**1.A. The similarity footprint**
[TLDR] *better adaptivity, better optimality, worse efficiency compared to fixed similarity; worse optimality, but way better efficiency compared to adaptive similarity considering more subspaces.*

Sorting and cumulative sum is not the core of footprint, while the idea of exploiting subspaces is.
The footprint picks $d$ most informative subspaces from all $2^{d} - 1$ subspaces, and gathering these subspatial information is helpful to optimality.
However, sorting and cumulative sum happen to be the right algorithm for computing the similarity score efficiently for decomposable similarity measure, it doesn't mean that the footprint is all about sorting dimensions.
Regard the footprint as mining the most valuable subspatial information from all $2^d - 1$ subspaces, and it provides useful evidence for guiding the model to make decision on whether $\boldsymbol \xi$ generates $\mathbf x$ or not.

Recall that $\text{ftpt}_{\text{sim}}(\boldsymbol \xi, \mathbf x)_k = \text{sim}^{(k)}(\boldsymbol \xi, \mathbf x)$, and $\text{sim}^{(k)}(\boldsymbol \xi, \mathbf x)$ finds the optimal $k$-dimensional subspace such that $\boldsymbol \xi$ and $\mathbf x$ is most associated (closest) on this subspace.
That is to say, the footprint finds a representative for each dimensionality $k \in [d]$ and regards the one that minimize the distance between $\boldsymbol \xi$ and $\mathbf x$ as the most informative $k$-dimensional subspace.
Although passing out non-informative subspaces gains efficiency, it also reduces optimality and adaptivity for losing information.

However, compared to fixed proximity-based similarity (e.g., $\boldsymbol \xi^\top \mathbf x$), the footprint offers a multi-scale descriptor to the relation between $\boldsymbol \xi$ and $\mathbf x$.
The $k$-optimal similarity $\text{sim}^{(k)}(\cdot, \cdot)$ ignores $d - k$ non-informative dimensions, which make it inherently suitable for handling masked dimensions and heavily noisy dimensions.
Therefore, the footprint enables better optimality against fixed proximity-based similarity as it offer richer evidence ($d$ v.s. $1$) for models to make the right decision.

---

> ### Author Response · Authors · 2025-11-24
> **Frequently Asked Questions - Part 1 (Continued)**
>
> **1.B. Linear Combination of footprint elements**
> [TLDR] *better adaptivity, better optimality compared to fixed similarity; worse adaptivity, worse optimality, but better learnability and better efficiency compared to more complicated models.*
>
> The weight $\mathbf w$ allows the model to behave differently when facing different variant distribution.
> Also, one can see that when $\mathbf w_{d} = 1$ and $\mathbf w_{k} = 0$ for $1 \le k < d$, the adaptive similarity degenerates to its base similarity.
> That is to say, the adaptive similarity has better adaptivity and the optimality is at least as good as the fixed proximity-based similarity.
>
> However, one can use a more complicated method (e.g., train a deep neural network) to extract information from footprints and gains better optimality and adaptivity.
> But this also brings more computation cost and may make the model harder to train.
> Using a linear map is the simplest fastest way to exploit the footprint, and empirical results demonstrates that it can largely improve the optimality (Table 2) and adaptivity (Fig. 3) compared to existing associative memory models.
>
> **1.C. Aggregating among different base similarities**
> [TLDR] *better adaptivity, better optimality, worse efficiency compared to fixed similarity*
>
> Aggregating adaptive similarities with different base similarity would improve model's adaptivity and optimality but reduce its efficiency (scale up the runtime by a constant).
> Think of base similarity as base vectors in a linear space, and the aggregated model resembles the span of these basis, so that more space in this linear space are covered.
> The learnable parameter $\beta_{1 \cdots B}$ controls how each base similarity is used in different scenarios, and empirical results demonstrates that it can futher improve optimality (Appendix A.4.6 Table 11).
>
> **1.D. Near optimal correct retrieval behavior**
>
> A full discussion on adaptivity, optimality, learnability and efficiency tradeoffs and design choice for proposed adaptive similarity is presented in Appendix 4.3.1, which may also include the following discussion on optimality of proposed adaptive similarity on noisy, masked, biased, and mixed variants.
>
> Although the proposed adaptive similarity $\beta_{\text{dis}} \mathbf w_{\text{dis}} \text{ftpt}\_{\text{dis}}(\boldsymbol \Xi, \mathbf x)$ $+ \beta_{\text{dot}} \mathbf w_{\text{dot}} \text{ftpt}\_{\text{dot}}(\boldsymbol \Xi, \mathbf x)$ is not optimal in masked and biased settings, we believe that it would achieve near optimal accuracy for high probability (Appendix A.4.6).
>
> Furthermore, the proposed adaptive similarity is suitable for noisy + masked variants (assume that the affect of noise is less than masking).
> For example, if we know that $m$ dimensions will be masked (but we don't know which), the footprint will let the model focus on the uncorrupted $d - m$ dimensions as $\text{sim}^{(k)}(\boldsymbol \xi, \mathbf x)$ ($k \le d - m$) is only affected by noise, while $\text{sim}^{(k)}(\boldsymbol \xi, \mathbf x)$ ($k > d - m$) is affected mostly by mask.
> So that the footprint separates the noisy and masked dimensions and make it easier for the model (weights $\mathbf w$) to make predictions.
>
> **1.E. Empirical validations**
>
> In Appendix A.6. we reported an ablation study on varying components in propsed adaptive similarity.
> The conclusion is that the footprint structure (looking for the most informative subspaces) is crucial for optimality (both sorting and cumulative sum is necessary).

---

### Author Response · Authors · 2025-11-24
**Frequently Asked Questions - Part 2 (Time Complexity)**

> **2. On time complexity and runtime of adaptive similarity**

When conducting memory retrieval for a query $\mathbf x$, we need to find the adaptive similarity between $\mathbf x$ and all $N$ stored memory patterns $\boldsymbol \xi_{1 \cdots N}$.
The adaptive similarity need to sort dimension-wise similarity vector $\mathbf q_{1 \cdots d}$, which needs $\mathcal O(d \log d)$ time and is the bottleneck.
Therefore, the overall time complexity of memory retrieval is $\mathcal O(N d \log d)$, which is higher than that of the Hopfield network with fixed proximity-based similarity ($\mathcal O(Nd)$).

However, by scaling the dimensionality $d$ and number of patterns $N$, the runtime of adaptive similarity is 2 times slower than the unsorted similarity.
The experiments is run on a NVIDIA GeForce GTX 3090 GPU, and the time elapsed is obtained by `torch.cuda.Event.elapsed_time(end_event)`.
As written in the documentation, `torch.sort()` is implemented via radix sort, which might be the reason why the adaptive similarity is faster than expected.

The result is listed in the following table (all elapsed time is measured in milliseconds (ms)):

| sort? |  |  |  |  |  |  |
|-------|------|------|------|------|------|------|
| $d$ | 64 | 128 | 256 | 64 | 64 | 64 |
| $N$ | 2048 | 2048 | 2048 | 2048 | 4096 | 8192 |
| **Training time (ms, ↓)** |  |  |  |  |  |  |
| **not** sorted | 840 | 1573 | 3609 | 840 | 1531 | 2949 |
| **sorted** | 2093 | 3059 | 8224 | 2093 | 4011 | 7983 |
| ratio | **2.49×** | **1.95×** | **2.27×** | **2.49×** | **2.62×** | **2.71×** |
| **Inference time (ms, ↓)** |  |  |  |  |  |  |
| **not** sorted | .355 | .368 | .414 | .355 | .446 | .570 |
| **sorted** | .287 | .318 | .326 | .287 | .364 | .467 |
| ratio | **1.23×** | **1.15×** | **1.27×** | **1.23×** | **1.23×** | **1.22×** |

For training, each model is trained for 200 epoches with the same batch size 256, learning rate 0.01, and optimizer `AdamW`.
For inference, each model performs 2000 memory retrieval tasks at once.

From the table, we can see that using `torch.sort()` requires about 3 times of the time without sorting, while it only needs 20% more time during the inference phrase, suggesting that adaptive similarity is light-weight.

---

### Author Response · Authors · 2025-12-04
**Summary of Rebuttal**

**Dear Area Chairs, Senior Area Chair, and Program Chairs,**

We sincerely thank you and all reviewers for the time and effort dedicated to evaluating and improving our paper.
A highly effective rebuttal is carried out during the past three weeks.
2 of 3 reviewers (Reviewer `67xP`, `ohC5`) replied, and **one of them (Reviewer `ohC5`) has raised its score from `6` to `8`** (in Dec. 25, before the incident).

* ### **Overview of Revisions**
The authors engaged with three reviewers (`67xP`, `L7xN`, `ohC5`), resulting in a significantly improved manuscript.
Key updates include the addition of **Figure 2** (illustrating architecture, training procedure, and a visual example), the integration of theoretical proofs and discussions from the appendix into the main text (Section 3.3), and a polished version of the Methods and Experiments sections for clarity.
Additionally, a detailed discussion regrading the good design choice for adaptive similarity is added to Appendix A.3.1.

* ### **Key Concerns Addressed**

**1. Design Choice Validation (Reviewers `67xP`, `L7xN`, `ohC5`)**

We clarified that the adaptive similarity is designed to adapt itself to different variant distributions (good adaptivity) and achieve high retrieval accuracy (better correctness), with small amount of learnable parameter (small scale), and relatively small time complexity (efficient).
We wrote the explanation in public comment **Frequently Asked Questions - Part 1 (Design Choice)**, and added a detailed and improved version in **Appendix A.3.1 On Good Design Choice of Adaptive Similarity**.

**2. Time Complexity and Runtime Overhead (Reviewers 67xP, ohC5)**

We clarified that the time complexity of computing the proposed adaptive similarity (Eq. 4) is $\mathcal O(d \log d)$, and added a experiment estimating the training and inference time of adaptive similarity with and without sorting.

**3. Novelty and Distinction from Existing Models (Reviewer ohC5)**

We differentiated our **A-Hop** from Wu & Hu et al. (2024) **U-Hop** in three different aspects. A-Hop focuses on "correct retrieval" using **adaptive similarity** that adapts itself to align with variant distributions (e.g., noisy, masked, biased), whereas K-Hop uses fixed similarity measures for epsilon-retrieval.
Reviewer `ohC5` is satisfied with our response.

**4. New Multi-Modal Retrieval Experiments (Reviewer `L7xN`)**

A new multi-modal memory retrieval task was introduced as suggested by Reviewer `L7xN`.
Results demonstrate that **A-Hop significantly outperforms** other methods in complex scenarios.

**5. Polishing and Improving the Manuscript (Reviewer `L7xN`, `ohC5`)**

We polished and improved the Methods (section 3.2, 3.3), and the Experiments part (section 4.1, 4.2, 4.3), so that the readability is improved and ambiguous expressions are revised. We added more clearly analysis of the tabular results, and briefly discuss their implications as required by Reviewer `L7xN`.
We also added Figure 2 to illustratively demonstrate the structure and training procedure of `A-Hop` and adaptive similarity, and a visual example of how adaptive similarity find the similarity score is added as required by Reviewer `L7xN`.

* ### **Reviewer's Reply**

**Reviewer `67xP`** acknowledged that we have addressed most of its concerns, and it need one more clarification on *correct retrieval*.
We added a detailed reply for clarifications, and we also improved the manuscript by following all of its minor suggestions.

**Reviewer `L7xN`** did not reply.
We completed all of its requirements (1. Clarifying the proposed method, 2. Polishing the manuscript and adding illustrative examples, 3. Adding experiments on multi-modal settings).

**Reviewer `ohC5`** replied and acknowledge that we have addressed all of its concerns (**and raised its score from `6` to `8`**), and suggested us to (1) change *Optimality* to some other word, (2) Rename `K-Hop` as it is named `U-Hop` in the original work, (3) Continuing polishing the manuscript especially the discussion added to the appendix.
We then (1) use *Correctness*  as the and use *Optimality* for absolute correct only, (2) Rename `K-Hop` to `U-Hop`, (3) Polished our manuscript further.

**Best regards,**

**Authors of Submission13441**

---

### Meta-Review · Area_Chair_Rjsb · 2026-01-05

**Summary:**

This paper introduces a principled and compelling reformulation of associative memory retrieval by grounding correctness in a generative, MAP-based perspective rather than heuristic proximity. The notion of a variant distribution provides a clear conceptual advance, explaining why fixed similarity measures are fundamentally insufficient and motivating the need for adaptive similarity learned from context.

The proposed adaptive similarity mechanism is theoretically well-founded, with rigorous guarantees under broad and canonical variant models, and is instantiated in a novel adaptive Hopfield network. Extensive experiments across diverse tasks consistently demonstrate strong empirical gains, reinforcing both the generality and practical impact of the approach. Overall, this work is technically solid, conceptually insightful, and represents a meaningful contribution to associative memory and interpretable neural architectures.

**Reviewer Scores:**

can't predict reliably

---

### Decision · Program_Chairs · 2026-01-26

Accept (Poster)